# Sub-optimal Experts mitigate Ambiguity in Inverse Reinforcement Learning

**Riccardo Poiani**
DEIB, Politecnico di Milano
riccardo.poiani@polimi.it

**Gabriele Curti**
DEIB, Politecnico di Milano
gabriele.curti@mail.polimi.it

**Alberto Maria Metelli**
DEIB, Politecnico di Milano
albertomaria.metelli@polimi.it

**Marcello Restelli**
DEIB, Politecnico di Milano
marcello.restelli@polimi.it

## Abstract

Inverse Reinforcement Learning (IRL) deals with the problem of deducing a reward function that explains the behavior of an expert agent who is assumed to act *optimally* in an underlying unknown task. Recent works have studied the IRL problem from the perspective of recovering the *feasible reward set*, i.e., the class of reward functions that are compatible with a unique optimal expert. However, in several problems of interest it is possible to observe the behavior of multiple experts with different degree of optimality (e.g., racing drivers whose skills ranges from amateurs to professionals). For this reason, in this work, we focus on the reconstruction of the feasible reward set when, in addition to demonstrations from the optimal expert, we observe the behavior of multiple *sub-optimal experts*. Given this problem, we first study the theoretical properties showing that the presence of multiple sub-optimal experts, in addition to the optimal one, can significantly shrink the set of compatible rewards, ultimately mitigating the inherent ambiguity of IRL. Furthermore, we study the statistical complexity of estimating the feasible reward set with a generative model and analyze a uniform sampling algorithm that turns out to be minimax optimal whenever the sub-optimal experts' performance level is sufficiently close to that of the optimal expert.

## 1 Introduction

*Inverse Reinforcement Learning* [IRL, 26] deals with the problem of recovering a reward function that explains the behavior of an expert agent who is assumed to act optimally in an underlying unknown task. Over the years, the IRL problem has consistently captured the attention of the research community (see, for instance, [4] and [1] for in-depth surveys). Indeed, this general scenario, where the reward function needs to be learned, emerges in numerous real-world applications. A example arises from human-in-the-loop settings [25], where the expert is a human solving a task, and an explicit specification of the human's goal in the form of a reward function is often unavailable. Notably, humans encounter difficulty in expressing their intentions in a numerical form, preferring instead to demonstrate what they perceive as the correct behavior. Once we retrieve a reward function, (i) we obtain explicit information for understanding the expert's choices, and, furthermore, (ii) we can utilize it to train reinforcement learning agents, even under shifts in the underlying system.

38th Conference on Neural Information Processing Systems (NeurIPS 2024).

Since the seminal work [26], IRL has emerged as a significantly complex task. One of its primary challenges lies in the intrinsic *ill-posed* nature of the problem, as multiple reward functions compatible with the expert's behavior exist. Recently, a promising avenue of research [24, 19, 22] has tackled this ambiguity issue from an intriguing perspective. Specifically, this strand of works focuses on estimating *all* the reward functions that are compatible with the observed demonstration, thereby postponing the selection of the reward and directing their focus solely on the expert's intentions.

Nevertheless, these approaches based on recovering the feasible set [24, 19, 22] fall short in modeling more articulated situations that arise in the real world. In several problems of interest, it is possible to observe the behavior of *multiple agents with different degrees of expertise*. As an illustrative example, consider the human-in-the-loop settings, mentioned above, in which we are interested in recovering reward functions that explain the intent behind racing drivers. In this scenario, racing car companies typically have access to a variety of drivers with diverse skills, including professionals, test drivers, and emerging talents from developmental programs. In this context, while the focus is typically on the reward function of professional drivers, we expect a proficient IRL method to effectively leverage demonstrations provided by drivers with lower expertise. From an intuitive perspective, if we have some information on the degree of expertise of other drivers, we expect that, by exploiting their demonstrations, we can reduce the ambiguity of IRL problem.

With these motivations, in this work, we extend the IRL formulation as the reconstruction of the feasible reward set to settings where, in addition to demonstrations from an optimal expert, we observe the behavior of multiple sub-optimal experts, of which we know a *bound on their sub-optimality*. More specifically, we will primarily focus in answering the following theoretical questions:

(Q1) *How does the presence of sub-optimal experts affect the set of reward functions that are compatible with the observed behaviors? Can they mitigate the intrinsic ambiguity of IRL?*

(Q2) *What is the statistical complexity of estimating the set of reward functions that are compatible with the given experts? How does it compare against the one of single-experts IRL problems?*

**Contributions and Outline.** After providing the necessary background, we introduce the novel problem of Inverse Reinforcement Learning with multiple and sub-optimal experts (IRL-SE, Section 2). We then proceed by studying the *theoretical properties* of the class of reward functions that are compatible with a given set of experts under the assumption that an upper bound on the performance between a sub-optimal expert and the optimal expert is available to the designer of the IRL system (Section 3). Our findings indicate that having multiple sub-optimal experts can significantly shrink the set of compatible rewards, thereby *mitigating* the ambiguity issue that affects IRL. Leveraging our previous results, we continue by studying the *statistical complexity* of estimating the feasible reward set with a generative model (Section 4). To this end, after formally introducing a Probabilistic Approximately Correct [PAC, 9] framework, we derive a novel lower bound on the number of samples that are required to obtain an accurate estimate of the feasible reward set. Then, we present a uniform sampling algorithm and analyze its theoretical guarantees. Our results show that (i) the IRL problem with sub-optimal experts is statistically harder than the single expert IRL setting, and (ii) that the uniform sampling algorithm is minimax optimal whenever the sub-optimal experts' performance level is sufficiently close to the one of the optimal expert. Finally, we conclude with a discussion on existing works (Section 5) and by highlighting potential avenues for future research (Section 6).

## 2 Preliminaries

In this section, we provide the notation and essential concepts employed throughout this document. Appendix A contains tables of symbols and a summary of the notation.

**Notation.** Let $\mathcal{X}$ be a finite set, we denote with $\Delta^{\mathcal{X}}$ the set of probability measures over $\mathcal{X}$. Let $\mathcal{Y}$ be a set, we denote with $\Delta^{\mathcal{X}}_{\mathcal{Y}}$ the set of functions $f : \mathcal{Y} \to \Delta^{\mathcal{X}}$. Given $f \in \mathbb{R}^n$, we denote with $\|f\|_{\infty}$ the infinite norm of $f$. Let $\mathcal{X}$ and $\mathcal{X}'$ be two non-empty subsets of a metric space $(\mathcal{Y}, d)$, we define the Hausdorff distance [31] between $\mathcal{X}$ and $\mathcal{X}'$ as $H_d(\mathcal{X}, \mathcal{X}') = \max\{\sup_{x \in \mathcal{X}} \inf_{x' \in \mathcal{X}'} d(x, x'), \sup_{x' \in \mathcal{X}'} \inf_{x \in \mathcal{X}} d(x, x')\}$. The Hausdorff distance directly depends on the metric $d$. We denote with $\mathbf{1}_n$ the $n$-dimensional vector given by $(1, \dots, 1)^{\top}$.

**Markov Decision Processes.** A Markov Decision Process *without a reward function* (MDP\R) is a tuple $\mathcal{M} = (\mathcal{S}, \mathcal{A}, p, \gamma)$, where $\mathcal{S}$ is the set of states, $\mathcal{A}$ is the set of actions, $p \in \Delta^{\mathcal{S}}_{\mathcal{S} \times \mathcal{A}}$ is

the transition probability kernel, and $\gamma \in [0, 1)$ is the discount factor. We consider finite state and action spaces, namely $|\mathcal{S}| = S$ and $|\mathcal{A}| = A$. A Markov Decision Process [MDP, 27] is obtained by combining an MDP\R $\mathcal{M}$ with a reward function $r \in \mathbb{R}^{\mathcal{S} \times \mathcal{A}}$. Without loss of generality, we assume reward functions bounded in $[0, 1]$. We denote with $\mathcal{M} \cup r$ the resulting MDP. The behavior of an agent is described by a policy $\pi \in \Delta_{\mathcal{S}}^{\mathcal{A}}$, that, for each state, prescribes a distribution over actions.

**Operators.** Let $f \in \mathbb{R}^{\mathcal{S}}$ and $g \in \mathbb{R}^{\mathcal{S} \times \mathcal{A}}$. We denote with $P$ and $\pi$ the operators induced by the transition model $p$ and the policy $\pi$ respectively.[1] Specifically, $Pf(s, a) = \sum_{s' \in \mathcal{S}} p(s'|s, a) f(s')$, and $\pi g(s) = \sum_{a \in \mathcal{A}} \pi(a|s) g(s, a)$. Moreover, we introduce the operators $E$ and $\bar{B}^\pi$ defined as: $Ef(s, a) = f(s)$ and $\left( \bar{B}^\pi g \right)(s, a) = \mathbb{1} \{\pi(a|s) = 0\} g(s, a)$. Finally, we define $d^\pi f$ as the expectation of $f$ under the discounted occupancy measure: $d^\pi f = (I_{\mathcal{S}} - \gamma \pi P)^{-1} f = \sum_{t=0}^{+\infty} (\gamma \pi P)^t f$.

**Value Functions and Optimality.** Given an MDP $\mathcal{M} \cup r$ and a policy $\pi$, the *Q-function* $Q_{\mathcal{M} \cup r}^\pi$ represents the expected discounted sum of rewards collected in $\mathcal{M} \cup r$ starting from $(s, a)$ and following policy $\pi$. Formally:

$$Q_{\mathcal{M} \cup r}^\pi(s, a) = \mathbb{E} \left[ \sum_{t=0}^{+\infty} \gamma^t r(s_t, a_t) | s_0 = s, a_0 = a \right],$$

where the expectation is taken w.r.t. the stochasticity of the policy and the environment, i.e., $s_{t+1} \sim p(\cdot|s_t, a_t)$ and $a_t \sim \pi(\cdot|s_t)$. The *V-function* $V_{\mathcal{M} \cup r}^\pi$ is the expectation of the $Q$-function over the action space, namely $V_{\mathcal{M} \cup r}^\pi = \pi Q_{\mathcal{M} \cup r}^\pi$. The *advantage function* $A_{\mathcal{M} \cup r}^\pi = Q_{\mathcal{M} \cup r}^\pi - E V_{\mathcal{M} \cup r}^\pi$ is the immediate gain of taking a given action, rather than following policy $\pi$. A policy $\pi^*$ is optimal if it has non-positive advantage in each-state action pair; namely $A_{\mathcal{M} \cup r}^{\pi^*} \leq 0$ holds element-wise.

**Inverse Reinforcement Learning.** An Inverse Reinforcement Learning [IRL, 26] problem is a tuple $\mathfrak{B} = (\mathcal{M}, \pi_E)$, where $\mathcal{M}$ is an MDP\R and $\pi_E \in \Delta_{\mathcal{S}}^{\mathcal{A}}$ is the expert policy. Given a reward function $r \in \mathbb{R}^{\mathcal{S} \times \mathcal{A}}$, we say that $r$ is *feasible* for $\mathfrak{B}$ if it is compatible with the behavior of the expert, namely $\pi_E$ is an optimal policy for the MDP $\mathcal{M} \cup r$. We denote with $\mathcal{R}_{\mathfrak{B}}$ the set of feasible reward functions:

$$\mathcal{R}_{\mathfrak{B}} = \left\{ r \in [0, 1]^{\mathcal{S} \times \mathcal{A}} : A_{\mathcal{M} \cup r}^{\pi_E} \leq 0 \right\}. \tag{1}$$

The set $\mathcal{R}_{\mathfrak{B}}$ is named *feasible reward set* [24, 19, 22]. To characterize the set $\mathcal{R}_{\mathfrak{B}}$, [24] have shown that a reward function $r$ belongs to $\mathcal{R}_{\mathcal{B}}$ if and only if there exists $\zeta \in \mathbb{R}_{\geq 0}^{\mathcal{S} \times \mathcal{A}}$ and $V \in \mathbb{R}^{\mathcal{S}}$ such that:

$$r = -\bar{B}^{\pi_E} \zeta + (E - \gamma P) V. \tag{2}$$

Thus, each reward function in $\mathcal{R}_{\mathfrak{B}}$, is the sum of two components. The first one, $-\bar{B}^{\pi_E} \zeta$, which is non-zero only when $\pi_E(a|s) = 0$, can be interpreted as the advantage function $A_{\mathcal{M} \cup r}^{\pi_E}$. The second one, $(E - \gamma P)V$, instead, can be interpreted as a reward-shaping via function $V$, which maintains the optimality of the expert's policy [26]. It follows that $\|V\|_\infty \leq (1 - \gamma)^{-1}$ and $\|\zeta\|_\infty \leq (1 - \gamma)^{-1}$.

## 3 Sub-Optimal Experts and the Feasible Reward Set

In this section, we extend the IRL formulation to problems where, in addition to demonstrations from an optimal expert, we observe the behaviors of multiple and sub-optimal experts. After having formulated and motivated the problem (Section 3.1), we delve into the theoretical properties of the induced feasible reward set, by providing both an *implicit* (Section 3.2) and *explicit* (Section 3.3) descriptions. Our results indicate that the presence of sub-optimal experts can significantly shrink the feasible set of compatible rewards, thus, mitigating the ambiguity issue of IRL.

### 3.1 Problem Formulation

We define the *Inverse Reinforcement Learning problem with multiple and Sub-optimal Experts* (IRL-SE) as a tuple $\mathfrak{B} = (\mathcal{M}, \pi_{E_1}, (\pi_{E_i})_{i=2}^{n+1}, (\xi_i)_{i=2}^{n+1})$, where $\mathcal{M}$ is an MDP\R, $\pi_{E_1}$ is the policy of an

---

[1]We use the symbol $\pi$ to indicate both the operator and the policy. In the following, the intended meaning will be clear from the context.

optimal expert,[2] $(\pi_{E_i})_{i=2}^{n+1}$ are a collection of $n$ sub-optimal experts policies, and $(\xi_i)_{i=2}^{n+1}$ are the corresponding sub-optimality bounds. More concretely, $\xi_i$ represents a known *upper bound* on the performance gap between the optimal expert and the $i$-th sub-optimal expert. Consequently, a reward function $r \in \mathbb{R}^{\mathcal{S} \times \mathcal{A}}$ is feasible for $\mathfrak{B}$ if $\pi_{E_1}$ is an optimal policy for the MDP $\mathcal{M} \cup r$ and if:

$$\forall i \in \{2, \ldots, n+1\}: \qquad \left\| V_{\mathcal{M} \cup r}^{\pi_{E_1}} - V_{\mathcal{M} \cup r}^{\pi_{E_i}} \right\|_\infty \leq \xi_i. \tag{3}$$

Thus, a feasible reward $r$ must make the value function $V_{\mathcal{M} \cup r}^{\pi_{E_i}}(s)$ of the $i$-th expert smaller than that of the optimal expert $V_{\mathcal{M} \cup r}^{\pi_{E_1}}(s) = V_{\mathcal{M} \cup r}^*(s)$ by no more than the threshold $\xi_i$, uniformly over $s \in \mathcal{S}$. We denote by $\mathcal{R}_{\bar{\mathfrak{B}}}$ the set of feasible rewards for $\bar{\mathfrak{B}}$, i.e., $r \in [0,1]^{\mathcal{S} \times \mathcal{A}}$ belongs to $\mathcal{R}_{\bar{\mathfrak{B}}}$ if (i) $A_{\mathcal{M} \cup r}^{\pi_{E_1}} \leq 0$ and (ii) Equation (3) holds. Notice that, whenever no sub-optimal expert is present, we recover the definition of the feasible set for single-expert IRL problems, i.e., $\mathcal{R}_{\mathfrak{B}}$ in Equation (1).

We remark that $\xi_i$ can even be a crude overestimate (i.e., an upper bound) of the sub-optimality of expert $i$. Nevertheless, as we shall see, the ability of the sub-optimal policy $\pi_{E_i}$ in mitigating the IRL ambiguity, i.e., shrinking the feasible reward set, will be tightly connected on the magnitude of $\xi_i$. The following examples illustrate how an expression $\xi_i$ can be obtained in common scenarios with no knowledge of the (possible) reward function optimized by the expert policy $\pi_{E_1}$. Formal proof for these statements are reported in Appendix C.

**Example 3.1.** Suppose that the $i$-th sub-optimal expert $\pi_{E_i}$ is optimal for the same reward function that $\pi_{E_1}$ is optimizing for, but under different transition model $P_i$. In this case, $\| V_{\mathcal{M} \cup r}^{\pi_{E_1}} - V_{\mathcal{M} \cup r}^{\pi_{E_i}} \|_\infty \leq \frac{2\gamma}{(1-\gamma)^2} \max_{(s,a) \in \mathcal{S} \times \mathcal{A}} \| P(\cdot|s,a) - P_i(\cdot|s,a) \|_1 =: \xi_i$ holds for all rewards $r$.

**Example 3.2.** Suppose that the $i$-th sub-optimal expert $\pi_{E_i}$ is optimal for the same reward function that $\pi_{E_1}$ is optimizing for, but using a different discount factor $\gamma'$. In this case, $\| V_{\mathcal{M} \cup r}^{\pi_{E_1}} - V_{\mathcal{M} \cup r}^{\pi_{E_i}} \|_\infty \leq 2 \frac{|\gamma - \gamma'|}{(1-\gamma)(1-\gamma')} =: \xi_i$ holds for all rewards $r$.

**Example 3.3.** Suppose that the $i$-th sub-optimal expert $\pi_{E_i}$ is sufficiently close to the optimal policy $\pi_{E_1}$, namely that $\max_{s \in \mathcal{S}} \| \pi_{E_1}(\cdot|s) - \pi_{E_i}(\cdot|s) \|_1 \leq \epsilon$. In this case, $\| V_{\mathcal{M} \cup r}^{\pi_{E_1}} - V_{\mathcal{M} \cup r}^{\pi_{E_i}} \|_\infty \leq \frac{1}{1-\gamma} \epsilon =: \xi_i$ holds for all rewards $r$.

### 3.2 Implicit Formulation of $\mathcal{R}_{\bar{\mathfrak{B}}}$

In this section, we analyze the implicit description of the feasible reward set $\mathcal{R}_{\bar{\mathfrak{B}}}$. From its definition (Equations (1) and (3)), a reward function $r \in [0,1]$ belongs to $\mathcal{R}_{\bar{\mathfrak{B}}}$ *if and only if* the following conditions are satisfied:

(i) $Q_{\mathcal{M} \cup r}^{\pi_{E_1}}(s,a) = V_{\mathcal{M} \cup r}^{\pi_{E_1}}(s) \qquad \forall (s,a) \in \mathcal{S} \times \mathcal{A} : \pi_{E_1}(a|s) > 0,$

(ii) $Q_{\mathcal{M} \cup r}^{\pi_{E_1}}(s,a) \leq V_{\mathcal{M} \cup r}^{\pi_{E_1}}(s) \qquad \forall (s,a) \in \mathcal{S} \times \mathcal{A} : \pi_{E_1}(a|s) = 0,$

(iii) $V_{\mathcal{M} \cup r}^{\pi_{E_1}}(s) \leq V_{\mathcal{M} \cup r}^{\pi_{E_i}}(s) + \xi_i \qquad \forall s \in \mathcal{S}, \forall i \in \{2, \ldots, n+1\}.$

Specifically, conditions (i) and (ii) directly encode the optimality of policy $\pi_{E_1}$ for $\mathcal{M} \cup r$, i.e., the advantage function $A_{\mathcal{M} \cup r}^{\pi_{E_1}}$ is non-positive. Condition (iii), on the other hand, arises from the presence of sub-optimal experts, and directly follows from Equation (3). At this point, by closely examining these conditions, it is possible to gain insight into the advantages and limitations associated with the presence of multiple and sub-optimal experts. Consider, the following illustrative examples.

**Example 3.4.** Suppose that $\pi_{E_i} = \pi_{E_1}$ holds for all $i \in \{2, \ldots, n+1\}$. Condition (iii) is clearly satisfied for any reward function $r$. Thus, the feasible reward set $\mathcal{R}_{\bar{\mathfrak{B}}}$ is determined by the requirement that the advantage function of $\pi_{E_1}$ is non-negative, and, as a consequence, the set $\mathcal{R}_{\bar{\mathfrak{B}}}$ coincides with the one of the single-expert IRL problem, namely $\mathcal{R}_{\bar{\mathfrak{B}}} = \mathcal{R}_{\mathfrak{B}}$. Analogously, if $\xi_i \geq (1-\gamma)^{-1}$ holds for all sub-optimal experts, condition (iii) is vacuous, and, similarly, $\mathcal{R}_{\bar{\mathfrak{B}}}$ reduces to $\mathcal{R}_{\mathfrak{B}}$.

**Example 3.5.** Consider the MDP with 2 states depicted in Figure 1, and suppose that only one additional sub-optimal expert is present. The optimal and the sub-optimal experts follow different policies in $S_0$. From the definition of $\mathcal{R}_{\bar{\mathfrak{B}}}$, we can see that, in addition to the constraint $r(S_0, A_1) \geq r(S_0, A_2)$ (i.e., $\pi_{E_1}$ is optimal), condition (iii) enforces a further relationship between $r(S_0, A_1)$ and $r(S_0, A_2)$, i.e., $r(S_0, A_1) - r(S_0, A_2) \leq \xi_i$. If $\xi_i$ is sufficiently small (i.e., $\xi_i < 1$), the presence of the sub-optimal experts significantly reduces the set of compatible rewards (Figure 2).

---

[2]For the sake of exposition, we consider a single optimal expert. The extension to cases where multiple optimal policies are available is straightforward. For further details on this point see Appendix B.

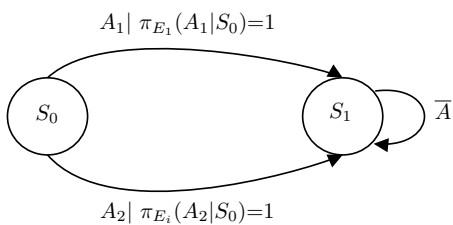

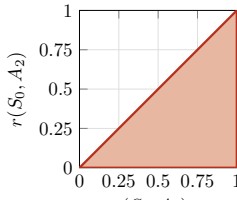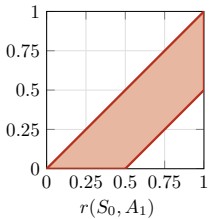

Figure 1: MDP example, with 2 states and 2 experts, that highlights the benefits of sub-optimal experts (Example 3.5). In $S_1$ both $\pi_{E_1}$ and $\pi_{E_i}$ are identical, i.e., $\pi_{E_1}(\bar{A}|S_1) = \pi_{E_i}(\bar{A}|S_1) = 1$.

Figure 2: Visualization of the feasible reward set (i.e., shaded red area) for the problems described in Example 3.5. On the left, the feasible reward set for the single-expert IRL problem and on the right the feasible reward set for the multiple and sub-optimal setting when using $\xi_i = 0.5$.

Abstracting away from the examples, we deduce that whenever (a) the sub-optimal experts exhibit behavior significantly different from that of the optimal expert and (b) their performance level is sufficiently close to being optimal, $\mathcal{R}_{\bar{\mathfrak{B}}}$ can notably shrink compared to $\mathcal{R}_{\mathfrak{B}}$. In the next section, through an explicit formulation of the feasible set, we analyze this phenomenon quantitatively.

### 3.3 Explicit Formulation of $\mathcal{R}_{\bar{\mathfrak{B}}}$

In this section, we continue by providing an explicit formulation of the feasible set $\mathcal{R}_{\bar{\mathfrak{B}}}$. The following result (proof in Appendix D) summarizes our findings.

**Theorem 3.6.** *Let $\bar{\mathfrak{B}}$ be an IRL problem with sub-optimal experts. Let $r \in [0,1]^{\mathcal{S} \times \mathcal{A}}$. Then, $r \in \mathcal{R}_{\bar{\mathfrak{B}}}$ if and only if there exists $\zeta \in \mathbb{R}_{\geq 0}^{\mathcal{S} \times \mathcal{A}}$ and $V \in \mathbb{R}^{\mathcal{S}}$ such that the following conditions are satisfied:*

$$r = -\bar{B}^{\pi^{E_1}}\zeta + (E - \gamma P)V, \tag{4}$$

*and, for all $i \in \{2, \ldots, n+1\}$:*

$$d^{\pi_{E_i}} \pi_{E_i} \bar{B}^{\pi_{E_1}} \zeta \leq \mathbf{1}_S \xi_i. \tag{5}$$

Theorem 3.6 deserves some comments. First, from Equation (4), we see that a necessary condition for having $r \in \mathcal{R}_{\bar{\mathfrak{B}}}$ is that it can be expressed as the sum of two components, namely $-\bar{B}^{\pi^{E_1}}\zeta$ and $(E - \gamma P)V$. This result is a direct consequence of the fact that $\pi_{E_1}$ is an optimal policy for $\mathcal{M} \cup r$, and, in this sense, it recovers the existing condition of single expert IRL settings (Equation 2) [24].[3]

***Sub-optimal experts constrain the sub-optimality gaps.*** The role of the sub-optimal experts is apparent in Equation (5) enforcing a set of *linear constraints* on the values that $\zeta$ can take.[4] Since, as discussed in Section 2, $-\bar{B}^{\pi_{E_1}}\zeta$ represents the advantage function of the optimal policy $\pi_{E_1}$, Equation (5) limits how much sub-optimal the values of actions not played by $\pi_{E_1}$ can be. Indeed, the resulting $Q$-function of the optimal expert can be expressed as $Q_{\mathcal{M} \cup r}^{\pi_{E_1}} = -\bar{B}^{\pi_{E_1}}\zeta + EV$ [24].

***Sub-optimal experts affect only states in which they behave sub-optimally.*** The linear constraints in Equation (5) are expressed in terms of $\pi_{E_i} \bar{B}^{\pi_{E_1}}\zeta$. As a consequence, they will only affect state-action pairs $(s, a) \in \mathcal{S} \times \mathcal{A}$ that are played by the sub-optimal experts (i.e., $\pi_{E_i}(a|s) > 0$) and that are not played by the optimal expert (i.e., $\pi_{E_1}(a|s) = 0$). Therefore, as previously highlighted with the implicit formulation of $\mathcal{R}_{\bar{\mathfrak{B}}}$, a sub-optimal expert $\pi_{E_i}$ should behave differently w.r.t. the optimal expert $\pi_{E_1}$ in order to provide meaningful information and reduce the feasible reward set. Furthermore, the constraints over $\zeta$ are expressed w.r.t. expected discounted occupancy of $\pi_{E_i}$.

---

[3]Contrary to single-experts IRL problems, now Equation (4) is only a necessary condition for having $r \in \mathcal{R}_{\bar{\mathfrak{B}}}$. Moreover, whenever $n = 1$ (i.e., we have the optimal expert $\pi_{E_1}$ only), Theorem 3.6 reduces to Equation (4), and, thus, it smoothly generalizes existing results for the classical IRL problem.

[4]Thanks to the linearity, testing whether a given $\zeta$ satisfies Equation (5) is computationally efficient.

Therefore, Equation (5) has provided a precise mathematical description (i.e., *if and only if conditions*) of the effect produced by the presence of sub-optimal experts on the feasible rewards $r$.

While in classical IRL problems, we have $\|\zeta\|_\infty \leq (1-\gamma)^{-1}$, the maximum value $\zeta$ can take in the sub-optimal experts case may be smaller. Let us fix a state $s' \in \mathcal{S}$ and a sub-optimal expert $i \in \{2, \ldots, n+1\}$; then, the constraint associated to state $s'$ in Equation (5) can be written as:

$$\sum_{s \in \mathcal{S}} d_{s'}^{\pi_{E_i}}(s) \sum_{a \in \mathcal{A}: \pi_{E_1}(a|s)=0} \pi_{E_i}(a|s)\zeta(s,a) \leq \xi_i, \tag{6}$$

where $d_{s'}^{\pi_{E_i}}(s)$ is the discounted expected number of times that policy $\pi_{E_i}$ visits state $s \in \mathcal{S}$ starting from state $s'$. From Equation (6), we obtain necessary conditions on the values of $\zeta$. More specifically, let $\mathcal{X}(s,a) \subseteq \{2, \ldots, n+1\}$ be the subset of sub-optimal experts such that $\pi_{E_i}(a|s) > 0$. Then, for each state-action pair $(s,a) \in \mathcal{S} \times \mathcal{A}$ such that $\pi_{E_1}(a|s) = 0$ and $\pi_{E_i}(a|s) > 0$, we have that:

$$\zeta(s,a) \leq \min\left\{k(s,a), \frac{1}{1-\gamma}\right\} := g(s,a), \quad \text{where} \quad k(s,a) := \min_{i \in \mathcal{X}(s,a), s' \in \mathcal{S}} \frac{\xi_i}{d_{s'}^{\pi_{E_i}}(s)\pi_{E_i}(a|s)}. \tag{7}$$

The term $k(s,a)$ directly follows from Equation (6), while $(1-\gamma)^{-1}$ is the maximum value that any $\zeta(s,a)$ can take, and arises, as in the classical IRL setting, from the fact that advantage functions are bounded by $(1-\gamma)^{-1}$ for any reward function. Thus, confirming our previous observations, Equation (7) implies a significant potential reduction in the maximum values that the advantage function can take, i.e., how much sub-optimal an action not played by $\pi_{E_1}$ can be in terms of $Q$-function values. [5]

**Example 3.7.** Consider an IRL problem with only one additional sub-optimal expert. Suppose that $\pi_{E_1}$ and $\pi_{E_i}$ are deterministic. For all state-action pairs in which $\pi_{E_1}(a|s) = 0$ and $\pi_{E_i}(a|s) = 1$, Equation (7) implies that $\zeta(s,a) \leq \min\left\{\xi_i, (1-\gamma)^{-1}\right\}$. If $\xi_i$ is significantly smaller than $(1-\gamma)^{-1}$, we obtain a notable restriction on the set of feasible reward functions.

**Remark 3.8** (About alternative Sub-optimality Formulations). We have described the sub-optimal experts by means of *upper bounds* on the value functions gaps; i.e., $V_{\mathcal{M} \cup r}^{\pi_{E_1}} - V_{\mathcal{M} \cup r}^{\pi_{E_i}} \leq \mathbf{1}_S \xi_i$. In principle, we may consider *lower bounds* on the value functions gaps; i.e., $V_{\mathcal{M} \cup r}^{\pi_{E_1}} - V_{\mathcal{M} \cup r}^{\pi_{E_i}} \geq \mathbf{1}_S \xi_i$. Intuitively, in the former case, we know that an expert is sub-optimal *at most* $\xi_i$, whereas, in the latter, that an expert is sub-optimal *at least* $\xi_i$. Our theory smoothly generalizes to these latter scenario obtaining an analogous of Theorem 3.6 where the constraints of Equation (5) are replaced with $d^{\pi_{E_i}} \pi_{E_i} \bar{B}^{\pi_{E_1}} \zeta \geq \mathbf{1}_S \xi_i$. We refer the reader to Theorem D.5 for a formal statement.

## 4 Learning the Feasible Set with Sub-Optimal Experts

In this section, we address the statistical complexity of estimating $\mathcal{R}_{\bar{\mathfrak{B}}}$ with a *generative model*. Specifically, we first introduce a Probabilistic Approximately Correct (PAC) framework (Section 4.1). Then, we study the statistical complexity of the problem by presenting lower bounds on the sample complexity any algorithm requires in order to correctly identify the feasible set (Section 4.1). Finally, we propose a uniform sampling algorithm and analyze its theoretical guarantees (Section 4.3).[6]

### 4.1 PAC Framework

We define a learning algorithm for an IRL-SE problem $\bar{\mathfrak{B}}$ as a tuple $\mathfrak{A} = (\tau, \nu)$ where $\tau$ is a stopping and $\nu = (\nu_t)_{t \in \mathbb{N}}$ is a history-dependent sampling strategy, i.e., $\nu_t \in \Delta_{\mathcal{D}_{t-1}}^{\mathcal{S} \times \mathcal{A}}$, where $\mathcal{D}_t = (\mathcal{S} \times \mathcal{A} \times \mathcal{S} \times (\mathcal{A})^{n+1})^t$ is the set of samples collected up to time step $t \in \mathbb{N}$. At each time step $t$, the algorithm selects a state-action pair $(s_t, a_t) \sim \nu_t$, and observes a sample $s_t' \sim p(\cdot|s_t, a_t)$ from the environment, together with actions sampled from the experts' policy, namely $(a_t^{(i)})_{i=1}^{n+1}$, where $a_t^{(i)} \sim \pi_{E_i}(\cdot|s_t)$. The observed realizations $\mathcal{D}_t$ are then used to update the sampling strategy $\nu_t$, and the process goes on until the stopping rule $\tau$ is satisfied. Then, the algorithm leverages the

---

[5]In Appendix I, we also provide some simple numerical experiments that aim at visualizing the reduction of the feasible reward set.

[6]For the sake of presentation, all results are presented under the assumption that $\pi_{E_1}$ is deterministic. The extension to the case in which $\pi_{E_1}$ is stochastic is presented in Appendix F.

collected data $\mathcal{D}_\tau$ to output the estimate of the feasible reward set $\widehat{\mathcal{R}}_\tau$. We are interested in designing algorithms that, for any desired accuracy $\epsilon \in (0, 1)$ and any risk parameter $\delta \in (0, 1)$, guarantee that:

$$\mathbb{P}_{\mathfrak{A}, \mathfrak{B}} \left( H_\infty(\mathcal{R}_{\bar{\mathfrak{B}}}, \widehat{\mathcal{R}}_\tau) > \epsilon \right) \le \delta. \tag{8}$$

We refer to these algorithms as $(\epsilon, \delta)$-correct identification strategies and we define their sample complexity as the total number of interactions with the generative model before stopping, i.e., $\tau$.

## 4.2 Lower Bound

In this section, we present lower bounds on the sample complexity that any $(\epsilon, \delta)$-correct algorithm needs to perform to learn $\mathcal{R}_{\bar{\mathfrak{B}}}$. The following theorem (proof in Appendix E) reports our result.

**Theorem 4.1.** *Let $\mathfrak{A}$ be a $(\epsilon, \delta)$-correct algorithm for the IRL problem with sub-optimal experts. There exists a problem instance $\mathfrak{B}$ such that the expected sample complexity is lower bounded by:*

$$\mathbb{E}_{\mathfrak{A}, \bar{\mathfrak{B}}}[\tau] \ge \Omega \left( \frac{SA}{\epsilon^2 (1-\gamma)^2} \left( \log \left( \frac{1}{\delta} \right) + S \right) \right), \tag{9}$$

*where $\Omega(\cdot)$ hides constant dependencies. Let $\pi_{\min} > 0$ be:*

$$\pi_{\min} := \min_{i \in \{2, \dots n+1 \dots\}} \min_{(s,a) \in \mathcal{S} \times \mathcal{A} : \pi_{E_i}(a|s) > 0} \pi_{E_i}(a|s), \tag{10}$$

*and let $q_0 := \pi_{\min}^{-1} \max_{i \in \{2, \dots, n+1\}} \xi_i$. Then, there exists an instance $\bar{\mathfrak{B}}'$ in which $q_0 < 1$ such that:*

$$\mathbb{E}_{\mathfrak{A}, \bar{\mathfrak{B}}'}[\tau] \ge \Omega \left( \frac{q_0^2 S \log \left( \frac{1}{\delta} \right)}{\epsilon^2 \pi_{\min}} \right). \tag{11}$$

Theorem 4.1 provides two distinct lower bounds (i.e., Equations 9 and 11) for IRL problems with sub-optimal experts. Whenever $q_0 < 1$ holds, the lower bound for the IRL-SE setting can be expressed as the maximum between Equation (9) and (11). We now comment in-depth on these two equations.

***Sub-optimal experts do not reduce the statistical complexity of IRL.*** Concerning Equation (9), as our analysis reveals, it directly arises from the problem of estimating rewards functions that are compatible with $\pi_{E_1}$ (i.e., with Equation (4) in Theorem 3.6). In this sense, it represents the complexity of single-expert IRL problems.[7] As a consequence of the structure of the feasible set we derived in Theorem 3.6, this leads to a lower bound also for the multiple sub-optimal experts setting. Therefore, Equation (9) formally shows that the sub-optimal expert setting is always at least as difficult as the single expert IRL problem.

***Sub-optimal experts have introduce further statistical complexities.*** Equation (11), on the other hand, is tightly related to the presence of sub-optimal experts. More precisely, under the assumption that $q_0 < 1$ (e.g., for sufficiently small values of $\xi_i$), it shows a dependency in the lower bound of a factor $\pi_{\min}^{-1}$, where $\pi_{\min}$ is the minimum probability with which sub-optimal experts plays their actions. From an intuitive perspective, its presence is related to the difficulty in estimating reward functions that are compatible with Equation (5) in Theorem 3.6. As we have shown in Section 3, the presence of sub-optimal experts can limit the value of $\zeta$ with a relationship that involves $\pi_{\min}^{-1}$ (i.e., Equation (7)). As our analysis will reveal, the proof of Equation (11) is directly related to these worst-case upper-bounds on $\zeta$. We remark that, according to the value of $\pi_{\min}$, Equation (11) can be significantly larger than Equation (5), thus showing an increased difficulty in the statistical complexity that is related to the stochasticity of sub-optimal experts.

We remark that $\tau$ represents the number of calls to the generative model and that each call, in our setting, provides demonstrations from *each* (sub-)optimal expert. It can be shown that, by slightly modifying the learning formalism, Equation (11) actually represents a *lower bound to the number of samples that should be gathered from* each *sub-optimal expert*.[8] In this sense, the statistical complexity increases significantly in the sub-optimal expert setting compared to the single expert one.

---

[7]We notice that similar results were presented in [22] for the finite-horizon single expert IRL problem. In this work, we extend their analysis to the infinite-horizon IRL setting.

[8]For further details, we refer the reader to Appendix G.

Therefore, to conclude, we notice that, in order to gain the reduction in the feasible reward set that we discussed in Section 5, we need to gather additional sampled demonstrations from the sub-optimal experts. This unavoidable trade-off is a direct consequence of the structure of the feasible set $\mathcal{R}_{\overline{\mathfrak{B}}}$ that we derived in Theorem 3.6, and, it arises from the statistical complexity of estimating reward functions that are compatible with the linear constraints of Equation (5).

### 4.3 Uniform Sampling Algorithm

In this section, we present the *Uniform Sampling algorithm for Inverse RL with Suboptimal Experts* (US-IRL-SE).

**Algorithm.** The pseudo-code can be found in Algorithm 1 US-IRL-SE takes as input the number of samples $m$ that will be queried to the generative model in each state-action pair. Then, it uniformly gathers data across the entire state-action space, and it updates the empirical estimates $\hat{p}_t$ and $(\hat{\pi}_{t,E_i})_{i=1}^{n+1}$ of the transition model and expert's policies. Let $\mathcal{D}_t$ be a dataset of $t \in \{1, \ldots, m\}$ tuples $\mathcal{D}_t = \{(s_j, a_j, s'_j, (a_j^{(i)})_{i=1}^{n+1})\}_{j=1}^t$, where $s'_j \sim p(\cdot|s_j, a_j)$, and $a_j^{(i)} \sim \pi_{E_i}(\cdot|s_j)$. Given $\mathcal{D}_t$, we define the empirical transition model $\hat{p}_t$ and the empirical experts' policy $\hat{\pi}_{t,E_i}$ as follows:

---

**Algorithm 1** US-IRL-SE Algorithm

**Require:** Number of samples for each $(s, a)$ pair $m$

1: **for** $t \in \{1, \ldots, m\}$ **do**
2:     Collect $(s', (a^{(i)})_{i=1}^{n+1})$ where $s' \sim p(\cdot|s, a)$ and $a^{(i)} \sim \pi_{E_i}(\cdot|s)$ from each $(s, a) \in \mathcal{S} \times \mathcal{A}$ and $i \in \{1 \ldots, n+1\}$
3:     Update $\hat{p}_t$ and $(\hat{\pi}_{t,E_i})_{i=1}^{n+1}$ according to Equation (12)
4: **end for**
5: **return** $\mathcal{R}_{\widehat{\mathfrak{B}}_m}$

---

$$\hat{p}_t(s'|s,a) = \begin{cases} \frac{N_t(s,a,s')}{N_t(s,a)} & \text{if } N_t(s,a) > 0 \\ \frac{1}{S} & \text{otherwise} \end{cases}, \quad \hat{\pi}_{t,E_i}(a|s) = \begin{cases} \frac{N_t^{(i)}(s,a)}{N_t(s)} & \text{if } N_t(s) > 0 \\ \frac{1}{A} & \text{otherwise} \end{cases}, \quad (12)$$

where $N_t(s, a, s') = \sum_{j=1}^t \mathbb{1}\{(s_j, a_j, s'_j) = (s, a, s')\}$, $N_t(s, a) = \sum_{s' \in \mathcal{S}} N_t(s, a, s')$, $N_t(s) = \sum_{(a,s') \in \mathcal{A} \times \mathcal{S}} N_t(s, a, s')$, and $N_t^{(i)}(s, a) = \mathbb{1}\{(s_j, a_j^{(i)}) = (s, a)\}$. Then, we denote with $\widehat{\mathfrak{B}}_t$ the empirical IRL problem induced by $\hat{p}_t$ and $(\hat{\pi}_{t,E_i})_{i=1}^{n+1}$. Finally, the algorithm returns the feasible set $\mathcal{R}_{\widehat{\mathfrak{B}}_m}$ corresponding to the estimated IRL-SE problem $\widehat{\mathfrak{B}}_m$ defined in terms of $\hat{p}_m$ and $(\hat{\pi}_{m,E_i})_{i=1}^{n+1}$.

**Sample Complexity Upper Bound**    The following theorem (proof in Appendix E), describes the theoretical guarantees of US-IRL-SE.

**Theorem 4.2.** *Let* $q_1 = \min\left\{\pi_{\min}^{-1} \max_{i \in \{2, \ldots, n+1\}} \xi_i, (1-\gamma)^{-1}\right\}$, *and* $q_2 = \max\{1, q_1\}$. *Then, with a total budget of:*

$$\widetilde{\mathcal{O}}\left(\max\left\{\frac{q_1^2 S \log\left(\frac{1}{\delta}\right)}{\epsilon^2 \pi_{\min}}, \frac{q_2^2 SA(S + \log\left(\frac{1}{\delta}\right))}{\epsilon^2(1-\gamma)^2}\right\}\right), \quad (13)$$

*US-IRL-SE is* $(\epsilon, \delta)$-*correct and* $\widetilde{\mathcal{O}}(\cdot)$ *hides constant and logarithmic dependencies.*

Theorem 4.2 deserves some comments. First of all, it shows that when the total number of queries to the generative is sufficiently large, US-IRL-SE is $(\epsilon, \delta)$-correct, and its sample complexity is provided in Equation (13). Since $m$ represents the number of calls to the generative model in each state-action pair, its expression can simply be calculated dividing Equation (13) by $SA$.[9] As a consequence, we remark that, in order to compute the value of $m$, the algorithm requires knowledge of the minimum probability $\pi_{\min}$ with which the sub-optimal experts play their actions.

***US-IRL-SE is minimax optimal when*** $\pi_{\min}^{-1} \xi_i \leq 1$**.**    Equation (13) is the maximum between two terms whose expressions closely resemble the lower bound of Theorem 4.1. The only difference arises in the definition of $q_0$, $q_1$ and $q_2$. We remark that, whenever the sub-optimal expert's performance level is sufficiently close to the one of the optimal expert, i.e., $\pi_{\min}^{-1} \xi_i \leq 1$ for all $i \in \{2, \ldots, n+1\}$,

---

[9]The exact expression of $m$ (i.e., constants and hidden logarithmic factors) is provided in Appendix H.

Equation (13) exactly recovers the lower bound that we presented in Theorem 4.1. Indeed, under this condition, it holds that $q_0 = q_1$, and $q_2 = 1$. We recall that, according to Theorem 3.6, as the values of $\xi_i$'s decrease, the feasible reward set is substantially reduced. In this sense, *US-IRL-SE enjoys minimax optimality in the most interesting scenarios where the presence of sub-optimal experts is particularly useful for mitigating the intrinsic ambiguity that affects IRL problems.*

***Technical challenges of Theorem 4.2.*** We highlight that the proof of Theorem 4.2 poses notable technical challenges beyond the ones of the analysis for standard IRL [24]. In our setting, studying how the Hausdorff distance between $\mathcal{R}_{\bar{\mathfrak{B}}}$ and $\mathcal{R}_{\widehat{\bar{\mathfrak{B}}}_t}$ decreases as we collect more samples requires taking into account that these feasible reward sets are subject to the peculiar structure identified in Theorem 3.6, namely the set of constraints of Equation (5) arising from the presence of sub-optimal experts. We study, with probabilistic arguments, error terms of the form $\|\zeta - \text{proj}_{\widehat{\Omega}_t} \zeta\|_\infty$, where $\zeta$ is feasible for the exact problem $\bar{\mathfrak{B}}$, $\widehat{\Omega}_t$ denotes the set of $\zeta$ that are compatible with the empirical problem $\widehat{\bar{\mathfrak{B}}}_t$, and $\text{proj}(\cdot)$ denotes the infinite norm projection. Analyzing these terms to obtain nearly optimal rates requires careful considerations on the geometry of feasible reward set that the sub-optimal experts induce. For further details, we refer the interested reader to Appendix E.

## 5 Related Works

In this section, we survey the works about IRL and the presence of multiple (sub-optimal) experts that are related to our proposal.

**Inverse Reinforcement Learning.** Historically, solving an IRL problem [1] involves determining a reward function that is compatible with the behavior of an optimal expert. Since the seminal work of [26], the problem has been recognized as ill-posed, as multiple reward functions that satisfies this requirement exists [34]. For this reason, over the years, several algorithmic criteria have been introduced to address this ambiguity issue. These criteria includes maximum margin [30], Bayesian approaches [28], maximum entropy [37], and many others [e.g., 21, 23, 36]. More recently, a new line of works have circumvented the ambiguity issue by redefining the IRL task as the problem of estimating the entire feasible reward set [24, 19, 22]. In our work, we take this novel perspective, and, in this sense, this recent research strand is the most related to our document. Specifically, of particular interests is the work of [22]. In their work, the authors study, for the first time, lower bounds for the single-expert IRL problem in finite horizon settings. Furthermore, they show that uniform sampling algorithm is minimax optimal for this task. Nevertheless, it has to remarked that this recent strand of research focuses entirely on single expert problems. As we have shown, however, the extension to the multiple and sub-optimal experts setting requires non-trivial effort. The reason is that the feasible reward set significantly differ (see, e.g., Theorem 3.6), and the problem is harder from a statistical perspective (see, e.g., Theorem 4.1).

**Multiple and/or Sub-optimal Experts.** The presence of multiple/sub-optimal experts has garnered attention in the Imitation Learning [IL, 11] community. In IL problems, contrary to IRL, the goal lies in directly leveraging demonstrations of optimal behavior to accelerate the training process of reinforcement learning algorithms. In this context, works that are close in spirit to ours are [17, 13, 6, 20]. Here, the authors extend the IL formulation to account for the fact that demonstrations are provided from multiple and/or sub-optimal experts. However, unlike our specific focus, their emphasis is on understanding how to effectively exploit imperfect demonstrations to improve training of RL agents. In our work, instead, we exploit the presence of sub-optimal experts to reduce the intrinsic ambiguity that affects the IRL formulation. In this sense, our work is complementary to several studies that analyzed how to improve the identifiability of the reward function in IRL problems by making additional structural assumptions. These include the possibility of observing an optimal expert interacting with several MDPs [e.g., 30, 3, 2] and focusing on peculiar types of MDPs that allows for strong theoretical guarantees [e.g., 8, 15, 5]. Along this line of work, the most related to ours is [32]. Here, the authors study how the presence of multiple experts impact the identifiability of the reward function. Contrary to our work, however, the authors assume each expert to follow an entropy regularized objective and, furthermore, they focus on the case in which all experts act optimally in the underlying environment. In this sense, our work encompasses a wider spectrum of applications, as we do not require optimality for each of the expert, nor an entropy

regularized objective. Furthermore, it has to be remarked that the multiple expert setting and IRL have been studied in [18] with the goal of providing practical algorithms that can be used in real-world applications. Also in this scenario, each expert is assumed to act optimally in the underlying domain. Finally, our work is related to approaches that aimed at extracting a *single* reward function by leveraging possibly sub-optimal demonstrations [e.g., 35, 16, 12, 29]. In our work, instead, we take a different theoretical perspective, and focus on the *set* of reward functions that are compatible with multiple sub-optimal experts.

## 6 Conclusions

In this work, we studied the novel problem of Inverse RL where, in addition to demonstrations from an optimal expert, we can observe the behavior of multiple and sub-optimal experts. More precisely, we first investigated the theoretical properties of the class of reward functions that are compatible with a given set of experts, i.e., the feasible reward set. Our results formally show that, by exploiting this additional structure, it is possible to significantly reduce the intrinsic ambiguity that affects the IRL formulation. Secondly, we have tackled the statistical complexity of estimating the feasible reward set from a generative model. More precisely, we have shown that a uniform sampling algorithm is minimax optimal whenever the performance level of the sub-optimal expert is sufficiently close to the one of the optimal expert.

Our research opens up intriguing avenues for future studies. In the following, we highlight several possibilities.

**Closing the Theoretical Gap** The results that we presented in Section 4 do not completely match. When the performance of the sub-optimal experts are not sufficiently close to the one of the optimal agent, the upper bound differs from the lower bound. Closing this gap, either by (i) developing tighter lower bounds or (ii) proposing novel algorithms and/or refining the analysis of US-IRL to achieve tighter upper bounds, is an interesting future direction. This would allow for a complete understanding of the statistical complexity of the problem.

**Offline IRL with Sub-Optimal Experts** Secondly, we note that this work leverages the presence of a generative model in the algorithm design. In the future, it would be interesting to remove this assumption by considering an offline setting where only a dataset of collected demonstrations is available to the learning system. This formulation is more practical since, in several real-world IRL applications [18], only a dataset of pre-collected demonstrations is available to the designer of the IRL system. We also note that this setting is, in principle, more challenging, as the data coverage of the dataset is not under the control of the IRL system.

**Large State-Action Spaces** For instance, since we have shown that sub-optimal experts can improve the identifiability of the reward function, future research should focus on building practical algorithms that can exploit this additional structure. To this end, as an intermediate step, it might be interesting to extend our results to the case in which the reward function is expressed as a linear combination of features. This approach would enable addressing infinite state-spaces [e.g., 26].

## Acknowledgments and Disclosure of Funding

Funded by the European Union – Next Generation EU within the project NRPP M4C2, Investment 1.,3 DD. 341 - 15 march 2022 – FAIR – Future Artificial Intelligence Research – Spoke 4 - PE00000013 - D53C22002380006. AI4REALNET has received funding from European Union's Horizon Europe Research and Innovation programme under the Grant Agreement No 101119527. Views and opinions expressed are however those of the author(s) only and do not necessarily reflect those of the European Union. Neither the European Union nor the granting authority can be held responsible for them.

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

The structure of the supplementary materials is organized as follows:

- Appendix A provide tables that summaries the main symbols used in this document.

- Appendix B describes how to extend our result to the multiple optimal expert setting.

- Appendix C provides proof of Examples 3.1-3.3.

- Appendix D provides formal proofs of the theoretical claims of Section 3, together with discussions on alternative formulations of the IRL-SE problem.

- Appendix E provides formal proofs of the theoretical claims of Section 4.

- Appendix F discusses how to extend our result to the setting in which $\pi_{E_1}$ is stochastic.

- Appendix G discusses how to change the learning formalism to derive a lower-bound that directly depends on the number of samples that are needed from each sub-optimal expert.

- Appendix H provides additional details on US-IRL-SE (i.e., exact description of $m$) and computational complexity analysis.

- Appendix I presents a simple numerical experiments that highlights the reduction of the feasible reward set due to the presence of multiple and sub-optimal experts.

## A  Symbols and Notation

To begin, we provide tables that summaries the main symbols used in this document. Table 1 reports a summary on the notation used throughout the paper. Table 2 reports a precise definition of the operators used.

Table 1: Notation

| SYMBOL | MEANING |
| --- | --- |
| $\mathfrak{B}$ | Inverse RL problem with a single optimal agent. |
| $\bar{\mathfrak{B}}$ | Inverse RL problem with multiple and sub-optimal experts. |
| $n$ | Number of sub-optimal experts, $n \in \mathbb{N}_{>0}$. |
| $\pi_{E_i}$ | Policy of the $i$-th expert. If $i = 1$, the expert is optimal. |
| $\xi_i$ | Sub-optimality of the $i$-th expert, where $i \in \{2, \ldots, n+1\}$. |
| $\hat{\pi}_{E_i}$ | Empirical estimates of the $i$-th expert policy $\pi_{E_i}$. |
| $\hat{p}$ | Empirical estimate of the transition model $p$. |
| $\mathcal{D}_t$ | Dataset of $t$ tuples from the generative model. |
| $N_t(s,a)$ | Number of samples gathered at state-action pair $(s,a)$ in $\mathcal{D}_t$. |
| $N_t(s)$ | Number of samples gathered at state $s$ in $\mathcal{D}_t$. |
| $\widehat{\bar{\mathfrak{B}}}$ | Empirical estimate of the IRL-SE problem induced by $\hat{p}$ and $\hat{\pi}_{E_i}$. |
| $\mathcal{R}_{\mathfrak{B}}$ | Feasible reward set of a single-agent IRL problem. |
| $\mathcal{R}_{\bar{\mathfrak{B}}}$ | Feasible reward set of a IRL-SE problem. |
| $\mathcal{R}_{\widehat{\bar{\mathfrak{B}}}}$ | Feasible reward set of the IRL-SE problem induced by $\hat{p}$ and $\hat{\pi}_{E_i}$. |
| $H_\infty(\mathcal{X}, \mathcal{X}')$ | Hausdorff distance between set $\mathcal{X}$ and $\mathcal{X}'$. |
| $\mathfrak{A}$ | Learning algorithm for the US-IRL-SE problem. |
| $\nu$ | Sampling strategy of a learning algorithm $\mathfrak{A}$. |
| $\tau$ | Stopping time (i.e., sample complexity) of a learning algorithm $\mathfrak{A}$. |
| $\epsilon$ | Desired level of accuracy when estimating the feasible reward set. |
| $\delta$ | Maximum risk tolerated when estimating the feasible reward set. |
| $m$ | Number of samples that the US-IRL-SE algorithm gathers in each state-action pair. |

## B  Additional Multiple Optimal Expert Setting

In this section, we discuss the extension of the IRL-SE setting to the case in which multiple optimal policies are available. More specifically, we define this IRL-SE setting as a tuple

Table 2: Operators

| SYMBOL | SIGNATURE | DEFINITION |
|--------|-----------|------------|
| $P$ | $\mathbb{R}^S \to \mathbb{R}^{S \times A}$ | $(Pf)(s,a) = \sum_{s' \in S} p(s'|s,a)f(s')$ |
| $\pi$ | $\mathbb{R}^{S \times A} \to \mathbb{R}^S$ | $(\pi f)(s) = \sum_{a \in \mathcal{A}} \pi(a|s)f(s,a)$ |
| $E$ | $\mathbb{R}^S \to \mathbb{R}^{S \times A}$ | $(Ef)(s,a) = f(s)$ |
| $\bar{B}^\pi$ | $\mathbb{R}^{S \times A} \to \mathbb{R}^{S \times A}$ | $(\bar{B}^\pi f)(s,a) = \mathbb{1}\{\pi(a|s) = 0\} f(s,a)$ |
| $B^\pi$ | $\mathbb{R}^{S \times A} \to \mathbb{R}^{S \times A}$ | $(B^\pi f)(s,a) = \mathbb{1}\{\pi(a|s) > 0\} f(s,a)$ |
| $d^\pi$ | $\mathbb{R}^S \to \mathbb{R}^S$ | $(d^\pi f)(s) = \sum_{t=0}^{+\infty} ((\gamma \pi P)^t f)(s)$ |
| $I_S$ | $\mathbb{R}^S \to \mathbb{R}^S$ | $(I_S f)(s) = f(s)$ |

$\tilde{\mathfrak{B}} = \left( \mathcal{M}, (\pi_{E_i}^*)_{i=1}^{n_1}, (\pi_{E_i})_{i=1}^{n_2}, (\xi_i)_{i=1}^{n_2} \right)$, where $(\pi_{E_i}^*)_{i=1}^{n_1}$ is a set of $n_1$ optimal policies, $(\pi_{E_i})_{i=1}^{n_2}$ is a set of $n_2$ sub-optimal policies with known degree of sub-optimality $(\xi_i)_{i=1}^{n_2}$.

At this point, one can verify that the shape of the feasible set $\mathcal{R}_{\tilde{\mathfrak{B}}}$, is exactly the one of Theorem 3.6, where Equations (4) an (5) are obtained by replacing $\pi_{E_1}$ with each optimal policy $\pi_{E_i}^*$ with $i \in \{1, \dots, n_1\}$.

Concerning learning the feasible reward set, it is sufficient to extend the generative model so that samples are gathered from all optimal and sub-optimal experts. The proof of Section 4 holds almost unchanged.

## C  Proof of Examples 3.1-3.3

We now give formal proofs of Examples 3.1-3.3.

*Proof of Example 3.1.* Consider two MDPs \R $\underline{\mathcal{M}}_1$ and $\underline{\mathcal{M}}_2$ that differs only in the transition kernel, which we denote by $p_1$ and $p_2$. Suppose that $\pi_{E_1}$ and $\pi_{E_2}$ are optimal for $\underline{\mathcal{M}}_1$ and $\underline{\mathcal{M}}_2$ respectively. For any state $s$, it holds that:

$$V_{\underline{\mathcal{M}}_1 \cup r}^{\pi_{E_1}}(s) - V_{\underline{\mathcal{M}}_1 \cup r}^{\pi_{E_2}}(s) \le V_{\underline{\mathcal{M}}_1 \cup r}^{\pi_{E_1}}(s) - V_{\underline{\mathcal{M}}_2 \cup r}^{\pi_{E_1}}(s) + V_{\underline{\mathcal{M}}_2 \cup r}^{\pi_{E_2}}(s) - V_{\underline{\mathcal{M}}_1 \cup r}^{\pi_{E_2}}(s),$$

where we added and subtracted $V_{\underline{\mathcal{M}}_2 \cup r}^{\pi_{E_1}}$, and we used $V_{\underline{\mathcal{M}}_1 \cup r}^{\pi_{E_1}} \le V_{\underline{\mathcal{M}}_2 \cup r}^{\pi_{E_2}}$ due to the optimality of $\pi_{E_2}$ in $\underline{\mathcal{M}}_2$. Then, focus on $V_{\underline{\mathcal{M}}_1 \cup r}^{\pi_{E_1}}(s) - V_{\underline{\mathcal{M}}_2 \cup r}^{\pi_{E_1}}(s)$ (but an identical reasoning can be applied for the second difference as well):

$$V_{\underline{\mathcal{M}}_1 \cup r}^{\pi_{E_1}}(s) - V_{\underline{\mathcal{M}}_2 \cup r}^{\pi_{E_1}}(s) = \gamma \sum_a \pi_{E_1}(a|s) \sum_{s'} p_1(s'|s,a)(V_{\underline{\mathcal{M}}_1}^{\pi_{E_1}}(s') - V_{\underline{\mathcal{M}}_2}^{\pi_{E_1}}(s')) +$$
$$+ (p_1(s'|s,a) - p_2(s'|s,a))V_{\underline{\mathcal{M}}_2}^{\pi_{E_1}}(s'),$$

which, in turn, can be further bounded by:

$$\frac{\gamma}{1-\gamma}||p_1 - p_2||_1 + \gamma \sum_a \pi_{E_1}(a|s) \sum_{s'} p_1(s'|s,a)(V_{\underline{\mathcal{M}}_1}^{\pi_{E_1}}(s') - V_{\underline{\mathcal{M}}_2}^{\pi_{E_1}}(s'))$$

Unrolling the summation to iterate the aforementioned argument, and using the fact that $\sum_{t=0}^{+\infty} \gamma^t = \frac{1}{1-\gamma}$ concludes the proof. $\square$

*Proof of Example 3.2.* In this proof, we explicit the relationship of the value function $V$ with the discount factor $\gamma$ by writing $V_{\underline{\mathcal{M}} \cup r}^{\pi, \gamma}$. Then, it holds that:

$$V_{\underline{\mathcal{M}} \cup r}^{\pi_{E_1}, \gamma}(s) - V_{\underline{\mathcal{M}} \cup r}^{\pi_{E_2}, \gamma}(s) \le V_{\underline{\mathcal{M}} \cup r}^{\pi_{E_1}, \gamma}(s) - V_{\underline{\mathcal{M}} \cup r}^{\pi_{E_1}, \gamma'}(s) + V_{\underline{\mathcal{M}} \cup r}^{\pi_{E_2}, \gamma'}(s) - V_{\underline{\mathcal{M}} \cup r}^{\pi_{E_2}, \gamma}(s),$$

where we added and subtracted $V_{\underline{\mathcal{M}} \cup r}^{\pi_{E_1}, \gamma'}(s)$, and we used $V_{\underline{\mathcal{M}} \cup r}^{\pi_{E_1}, \gamma'}(s) \le V_{\underline{\mathcal{M}} \cup r}^{\pi_{E_2}, \gamma'}(s)$ due to the optimality of $\pi_{E_2}$ for the discount factor $\gamma'$. Then, focus on $V_{\underline{\mathcal{M}} \cup r}^{\pi_{E_1}, \gamma}(s) - V_{\underline{\mathcal{M}} \cup r}^{\pi_{E_1}, \gamma'}(s)$. By the

definition of the value function, together with the fact that rewards are bounded in $[0, 1]$, we can rewrite this difference as:

$$\mathbb{E}[\sum_{t=0}^{+\infty}(\gamma^t - \gamma'^t)r(s_t, a_t)|s_0 = s] \leq \sum_{t=0}^{+\infty}\gamma^t - \gamma'^t = \frac{\gamma - \gamma'}{(1-\gamma)(1-\gamma')}.$$

An identical argument holds for the second difference, thus concluding the proof. $\qquad\square$

*Proof of Example 3.3.* By using the definition of value function, we can rewrite $V_{\mathcal{M}\cup r}^{\pi_{E_1}}(s) - V_{\mathcal{M}\cup r}^{\pi_{E_2}}(s)$ as $\sum_a(\pi_{E_1}(a|s) - \pi_{E_2}(a|s))r(s,a) + \gamma \sum_{s'} p(s'|s,a)(V_{\mathcal{M}\cup r}^{\pi_{E_1}}(s') - V_{\mathcal{M}\cup r}^{\pi_{E_2}}(s')))$. At this point, since rewards are bounded in $[0, 1]$ and by the policy similarity assumption, we have that the difference in value functions can be upper bounded by $\epsilon + \gamma \sum_{s'} p(s'|s,a)(V_{\mathcal{M}\cup r}^{\pi_{E_1}}(s') - V_{\mathcal{M}\cup r}^{\pi_{E_2}}(s'))$. Unrolling the summation to iterate the aforementioned argument, and using the fact that $\sum_{t=0}^{+\infty}\gamma^t = \frac{1}{1-\gamma}$ concludes the proof. $\qquad\square$

# D   Proofs and Derivations of Section 3

In this section, we provide formal proofs of the theoretical results of Section 3. We begin by reporting for completeness some results from [24] that will be used in our analysis.

**Lemma D.1.** *Let $\mathfrak{B} = (\mathcal{M}, \pi_E)$ be a single-agent IRL problem. Let $r \in [0, 1]^{S \times A}$, then $r$ is a feasible reward if and only if for all $(s, a) \in (S, A)$ it holds that:*

$$\begin{aligned}(i) \qquad & Q_{\mathcal{M}\cup r}^{\pi_E}(s, a) - V_{\mathcal{M}\cup r}^{\pi_E}(s) = 0 \qquad & if\ \pi_E(a|s) > 0, \\ (ii) \qquad & Q_{\mathcal{M}\cup r}^{\pi_E}(s, a) - V_{\mathcal{M}\cup r}^{\pi_E}(s) \leq 0 \qquad & if\ \pi_E(a|s) = 0.\end{aligned}$$

**Lemma D.2.** *Let $\mathfrak{B} = (\mathcal{M}, \pi_E)$ be a single-agent IRL problem. A Q-function satisfies condition of Lemma D.1 if and only if there exist $\zeta \in \mathbb{R}_{\geq 0}^{S \times A}$ and $V \in \mathbb{R}^S$ such that:*

$$Q_{\mathcal{M}\cup r} = -\bar{B}^{\pi_E}\zeta + EV$$

**Lemma D.3.** *Let $\mathfrak{B} = (\mathcal{M}, \pi_E)$ be a single-agent IRL problem. Let $r \in \mathbb{R}^{S \times A}$, then $r$ is a feasible reward, if and only if there exist $\zeta \in \mathbb{R}_{\geq 0}^{S \times A}$ and $V \in \mathbb{R}^S$ such that:*

$$r = -\bar{B}^{\pi_E}\zeta + (E - \gamma P)V$$

At this point, we proceed by proving a Lemma that implicitly describes the set of reward functions that are compatible with a IRL-SE problem.

**Lemma D.4.** *Let $\bar{\mathfrak{B}}$ be an IRL problem with sub-optimal experts. Let $r \in [0, 1]^{S \times A}$. Then, $r \in \mathcal{R}_{\bar{\mathfrak{B}}}$ if and only if the following conditions are satisfied:*

*(i) $Q_{\mathcal{M}\cup r}^{\pi_{E_1}}(s, a) = V_{\mathcal{M}\cup r}^{\pi_{E_1}}(s) \qquad \forall(s, a) : \pi_{E_1}(a|s) > 0,$*

*(ii) $Q_{\mathcal{M}\cup r}^{\pi_{E_1}}(s, a) \leq V_{\mathcal{M}\cup r}^{\pi_{E_1}}(s) \qquad \forall(s, a) : \pi_{E_1}(a|s) = 0,$*

*(iii) $Q_{\mathcal{M}\cup r}^{\pi_{E_1}} \leq V_{\mathcal{M}\cup r}^{\pi_{E_i}} + \mathbf{1}_S\xi_i \qquad \forall i \in \{2, \ldots, n+1\}.$*

*Proof.* Condition (i) and (ii) are necessary conditions for the claim to hold. This directly follows by the definition of $\mathcal{R}_{\bar{\mathfrak{B}}}$ and from Lemma D.1. At this point, what remains to be proven is condition (iii).

We begin by showing that, if $r \in \mathcal{R}_{\mathfrak{B}}$ then condition (iii) is satisfied. Since (i) and (ii) holds, it sufficies to plug $V_{\mathcal{M}\cup r}^{\pi_{E_1}} - V_{\mathcal{M}\cup r}^{\pi_{E_i}} \leq \xi_i$ within (i) and (ii) to obtain the desired result.

We conclude by showing that if (i)-(iii) are satisfied, then $r \in \mathcal{R}_{\mathfrak{B}}$. Since (i) and (ii) holds, then, by Lemma D.1, we know that $\pi_{E_1}$ is optimal for $\mathcal{M}\cup r$. It remains to be proven that $V_{\mathcal{M}\cup r}^{\pi_{E_1}} - V_{\mathcal{M}\cup r}^{\pi_{E_i}} \leq \xi_i$ holds as well. However, since (i) holds, (iii) implies that $V_{\mathcal{M}\cup r}^{\pi_{E_1}}(s) - V_{\mathcal{M}\cup r}^{\pi_{E_i}}(s) \leq \xi_i(s)$ holds for all $s \in \mathcal{S}$, thus concluding the proof. $\qquad\square$

We continue by proving Theorem 3.6.

**Theorem 3.6.** *Let $\bar{\mathfrak{B}}$ be an IRL problem with sub-optimal experts. Let $r \in [0,1]^{\mathcal{S} \times \mathcal{A}}$. Then, $r \in \mathcal{R}_{\bar{\mathfrak{B}}}$ if and only if there exists $\zeta \in \mathbb{R}^{\mathcal{S} \times \mathcal{A}}_{\geq 0}$ and $V \in \mathbb{R}^{\mathcal{S}}$ such that the following conditions are satisfied:*

$$r = -\bar{B}^{\pi^{E_1}}\zeta + (E - \gamma P)V, \tag{4}$$

*and, for all $i \in \{2, \ldots, n+1\}$:*

$$d^{\pi_{E_i}} \pi_{E_i} \bar{B}^{\pi_{E_1}} \zeta \leq \mathbf{1}_S \xi_i. \tag{5}$$

*Proof.* First of all, Equation (4) is a necessary condition for the claim to hold. Indeed, from Lemma D.3 and Lemma D.4, Equation (4) is a necessary and sufficient condition to make $\pi_{E_1}$ optimal in $\mathcal{M} \cup r$.

At this point, we proceed conditioning on the fact that Equation (4) holds. We need to show that, if $r \in \mathcal{R}_{\bar{\mathfrak{B}}}$, then Equation (5) holds, and if Equation (5) holds, then $r \in \mathcal{R}_{\bar{\mathfrak{B}}}$.

From Lemma D.4, we know that a necessary and sufficient condition for the previous statements to hold is that:

$$Q^{\pi_{E_1}}_{\mathcal{M} \cup r}(s,a) \leq V^{\pi_{E_i}}_{\mathcal{M} \cup r}(s) + \xi_i,$$

holds for any state-action pair $(s,a)$ and for all experts $i \in \{2, \ldots, n+1\}$. Given Equation (4) and Lemma D.2, the previous equation can be conveniently written as:

$$-\bar{B}^{\pi^{E_1}}\zeta + EV \leq EV^{\pi_{E_i}}_{\mathcal{M} \cup r} + E\mathbf{1}_S \xi_i. \tag{14}$$

At this point, consider $(s,a)$ such that $\pi_{E_1}(a|s) > 0$. Then, Equation (14) reduces to:

$$V(s) \leq V^{\pi_{E_i}}_{\mathcal{M} \cup r}(s) + \xi_i. \tag{15}$$

Conversely, consider $(s,a)$ such that $\pi_{E_1}(a|s) = 0$, then Equation (14) reduces to:

$$-\zeta(s,a) + V(s) \leq V^{\pi_{E_i}}_{\mathcal{M} \cup r}(s) + \xi_i. \tag{16}$$

Since $\zeta(s,a) \geq 0$ by assumption, Equation (16) is directly implied by Equation (15). Therefore, it suffices to study:

$$V(s) \leq V^{\pi_{E_i}}_{\mathcal{M} \cup r}(s) + \xi_i,$$

which can be rewritten as:

$$V - \mathbf{1}_S \xi_i \leq V^{\pi_{E_i}}_r = (I_{\mathcal{S}} - \gamma \pi_{E_i} P)^{-1} \pi_{E_i} r.$$

At this point, we can plug Equation (4) within the previous Equation. More specifically, we obtain:

$$V - \mathbf{1}_S \xi_i \leq (I_{\mathcal{S}} - \gamma \pi_{E_i} P)^{-1} \pi_{E_i} \left( -\bar{B}^{\pi_{E_1}}\zeta + (E - \gamma P)V \right). \tag{17}$$

We now proceed by rewriting the right hand side of Equation (17). More precisely, we notice that:

$$(I_{\mathcal{S}} - \gamma \pi_{E_i} P)^{-1} \pi_{E_i} ((E - \gamma P)V) = (I_{\mathcal{S}} - \gamma \pi_{E_i} P)^{-1} V - \gamma (I_{\mathcal{S}} - \gamma \pi_{E_i} P)^{-1} \pi_{E_i} PV$$

$$= (I_{\mathcal{S}} - \gamma \pi_{E_i} P)^{-1} (I_{\mathcal{S}} - \gamma \pi_{E_i} P)V$$

$$= V.$$

Plugging this result within Equation (17), we arrive at:

$$(I_{\mathcal{S}} - \gamma \pi_{E_i} P)^{-1} \pi_{E_i} \left( \bar{B}^{\pi_{E_1}}\zeta \right) \leq \mathbf{1}_S \xi_i,$$

which concludes the proof. $\qquad\square$

## D.1 Other Assumptions on the Behavior of the Sub-optimal Experts

In this section, we investigate the generality of the results presented in Theorem 3.6. Specifically, we remark that all the results that we derived in Section 3 can be easily extended to other assumptions on the sub-optimal experts. More specifically, the results of Theorem 3.6 can easily be extended to the following cases:

$$V^{\pi_{E_1}}_{\mathcal{M} \cup r} - V^{\pi_{E_i}}_{\mathcal{M} \cup r} \geq \mathbf{1}_{\mathcal{S}} \xi_i, \tag{18}$$

or

$$V_{\mathcal{M} \cup r}^{\pi_{E_1}} - V_{\mathcal{M} \cup r}^{\pi_{E_i}} = \mathbf{1}_\mathcal{S} \xi_i. \tag{19}$$

Equation (18) encodes the fact that that a given sub-optimal expert $i$ is at least $\xi_i$ sub-optimal w.r.t. the optimal policy $\pi_{E_1}$, while Equation (19) encodes the the fact that the sub-optimal expert $i$ is exactly $\xi_i$ sub-optimal w.r.t. the optimal agent. In these cases, it is possible to derive the following generalizations of Theorem 3.6.

**Theorem D.5.** *Let $\bar{\mathfrak{B}}$ be an IRL problem with sub-optimal experts where $V_{\mathcal{M} \cup r}^{\pi_{E_1}} - V_{\mathcal{M} \cup r}^{\pi_{E_i}} \geq \mathbf{1}_\mathcal{S} \xi_i$ holds for all sub-optimal experts $i$. Let $r \in [0,1]^{\mathcal{S} \times \mathcal{A}}$. Then, $r \in \mathcal{R}_{\bar{\mathfrak{B}}}$ if and only if there exists $\zeta \in \mathbb{R}_{\geq 0}^{\mathcal{S} \times \mathcal{A}}$ and $V \in \mathbb{R}^\mathcal{S}$ such that the following conditions are satisfied:*

$$r = -\bar{B}^{\pi^{E_1}} \zeta + (E - \gamma P)V, \tag{20}$$

*and, for all $i \in \{2, \ldots, n+1\}$:*

$$d^{\pi_{E_i}} \pi_{E_i} \bar{B}^{\pi_{E_1}} \zeta \geq \mathbf{1}_S \xi_i. \tag{21}$$

**Theorem D.6.** *Let $\bar{\mathfrak{B}}$ be an IRL problem with sub-optimal experts where $V_{\mathcal{M} \cup r}^{\pi_{E_1}} - V_{\mathcal{M} \cup r}^{\pi_{E_i}} = \mathbf{1}_\mathcal{S} \xi_i$ holds for all sub-optimal experts $i$. Let $r \in [0,1]^{\mathcal{S} \times \mathcal{A}}$. Then, $r \in \mathcal{R}_{\bar{\mathfrak{B}}}$ if and only if there exists $\zeta \in \mathbb{R}_{\geq 0}^{\mathcal{S} \times \mathcal{A}}$ and $V \in \mathbb{R}^\mathcal{S}$ such that the following conditions are satisfied:*

$$r = -\bar{B}^{\pi^{E_1}} \zeta + (E - \gamma P)V, \tag{22}$$

*and, for all $i \in \{2, \ldots, n+1\}$:*

$$d^{\pi_{E_i}} \pi_{E_i} \bar{B}^{\pi_{E_1}} \zeta = \mathbf{1}_S \xi_i. \tag{23}$$

The proofs of Theorem D.5 and D.6 are identical to the one of Theorem 3.6. In terms of results, the only difference lies in the fact that the set of linear constraints introduces a different type of relationship between $\zeta$ and $\xi_i$.[10]

At this point, we remark that Theorem D.5 follows a very similar interpretation of the one we presented in Section 3. In other words, the sub-optimal experts introduces *lower bounds* on the values of the advantage function that are played by the sub-optimal experts $\pi_{E_i}$ and that are not played by the optimal expert.[11] This result is as expected. Indeed, if we know that a given policy is sub-optimal at least by a given quantity $\xi_i$, then, intuitively, we can extrapolate knowledge, expressed as lower bounds, on how sub-optimal certain actions are.

Concerning Theorem D.6, instead, we notice that the result is much more stronger w.r.t. to the case in which inequalities are involved in the problem (i.e., Theorem 3.6 and Theorem D.5). More specifically, we notice that in this case, starting from Equation (23), it is possible to obtain the following result on the values of $\zeta$:

$$\pi_{E_i} \bar{B}^{\pi_{E_1}} \zeta = (I_\mathcal{S} - \gamma \pi_{E_i} P)\mathbf{1}_S \xi_i. \tag{24}$$

Developing this constraint for a specific state $s$, we obtain the following linear constraint:

$$\sum_{a:\pi_{E_1}(a|s)=0} \pi_{E_i}(a|s)\zeta(s,a) = \xi_i(1-\gamma). \tag{25}$$

In other words, it provides a set of hard constraints that the values of $\zeta$ should satisfy. At this point, we remark on two important observations. The first one is that Equation (25) might not be satisfied for any choice of $\zeta \in \mathbb{R}^{\mathcal{S} \times A}$. In particular, suppose that there is only one sub-optimal expert. In this case, if $\pi_{E_1} = \pi_{E_i}$, Equation (25) reduces to:

$$0 = \xi_i(1-\gamma),$$

---

[10]As a consequence of these results, we notice that it is direct to extend the properties of the feasible reward set to the case in which, e.g., for some experts it holds that $V_{\mathcal{M} \cup r}^{\pi_{E_1}} - V_{\mathcal{M} \cup r}^{\pi_{E_i}} \leq \xi_i$, while for other experts it holds that $V_{\mathcal{M} \cup r}^{\pi_{E_1}} - V_{\mathcal{M} \cup r}^{\pi_{E_i}} \geq \xi_i$.

[11]As a consequence, highly sub-optimal experts with different behaviors leads to the most effective reduction of the feasible reward set $\mathcal{R}_{\bar{\mathfrak{B}}}$. This is opposed to the results of Section 3.3, where close-to-optimal experts with different behaviors lead to the most effective reduction.

which is clearly false for any strictly positive value of $\xi_i$. We notice that this result is as expected. If the two policies are identical, then there should be no gap in the performance of $\pi_{E_1}$ and $\pi_{E_i}$. As a consequence, the feasible reward set of this IRL problem is empty.[12] Secondly, instead, suppose that all the experts are deterministic, and suppose that they all behave differently to $\pi_{E_1}$ in each state-action pair (so that the feasible reward set is non-empty). Then, focus on the $i$-th expert and consider a state-action pair $(s, a)$ such that $\pi_{E_i}(a|s) = 1$. Then, Equation (25) reduces to:

$$\zeta(s, a) = \xi_i(1 - \gamma),$$

In this sense, the presence of the sub-optimal expert implies a unique value that $\zeta(s, a)$ can take.[13] It follows that, is for each state-action pair $(s, a)$ such that $\pi_{E_1}(a|s) = 0$, there exists a single sub-optimal expert $i$ such that $\pi_{E_i}(a|s) = 1$, we are able to recover entirely a unique vector $\bar{\zeta}$ that is compatible with the underlying IRL problem. In other words, in this scenario, we are able to exactly recover the values of the advantage function that express the sub-optimality gaps of actions that are not played by the optimal agent.

### D.2 Measuring Volumes of the Feasible Values of $\zeta$

Finally, we conclude this section by reporting an additional analysis that quantitatively measure the reduction in the feasible values of $\zeta$ that are compatible with the presence of the sub-optimal experts.

To begin, we first introduce some notation. Let $\mathcal{X}$ be a measurable subset of $\mathbb{R}^n$, we denote with $\mathrm{Vol}(\mathcal{X})$ the Lebesgue measure of $\mathcal{X}$ [7]. In other words, $\mathrm{Vol}(\mathcal{X})$ represents the $n$-dimensional volume of $\mathcal{X}$.

At this point, from Theorem 3.6, we know that the presence of sub-optimal experts can effectively limit the values that $\zeta$ can assume. Consequently, in order to measure the reduction of the feasible reward set, we will compute upper bounds on the volume of the region of $\zeta$ that induces at least one feasible reward function in $\mathcal{R}_{\bar{\mathfrak{B}}}$. In the remainder of this section, for a generic IRL problem $\mathfrak{B}$, we will denote with $Z(\mathfrak{B})$ such set. More specifically, we define:

$$Z(\mathfrak{B}) = \left\{ \zeta \in \mathbb{R}_{\geq 0}^{S \times A} : \exists r \in \mathcal{R}_{\mathfrak{B}} : \exists V \in \mathbb{R}^S : \right.$$
$$\left. r = -\bar{B}^{\pi_{E_1}} \zeta + (E - \gamma P)V \text{ and } \zeta(s, a) = 0 \ \forall (s, a) : \pi_{E_1}(a|s) = 0 \right\}.$$

Notice that, we are directly restricting the analysis to state-action pairs for which $\pi_{E_1}(a|s) = 0$ holds. Indeed, fix $(s, a)$ such that $\pi_{E_1}(a|s) = 0$ holds. Then, changing the value of $\zeta(s, a)$ does not affect class of compatible reward functions.

At this point, the following proposition provides upper bounds on the volume of the feasible values of $\zeta$ for IRL and IRL-SE problems.

**Proposition D.7.** *Let $\mathfrak{B}$ and $\bar{\mathfrak{B}}$ be an IRL and an IRL-SE problem. Let $g(s, a)$ be defined as follows:*

$$g(s, a) := \begin{cases} \min\left\{ k(s, a), \frac{1}{1-\gamma} \right\} & \text{if } (s, a) : \pi_{E_1}(a|s) = 0 \text{ and } \left| \mathcal{X}(s, a) \right| > 0 \\ \frac{1}{1-\gamma} & \text{otherwise} \end{cases},$$

*where $k(s, a)$ is defined as in Equation (7), and $\mathcal{X}(s, a)$ denotes the subset of sub-optimal experts such that $\pi_{E_i}(a|s) > 0$. Then, it holds that:*

$$\mathrm{Vol}(Z(\mathfrak{B})) \leq \prod_{(s,a):\pi_{E_1}(s,a)=0} \frac{1}{1-\gamma}, \tag{26}$$

$$\mathrm{Vol}(Z(\bar{\mathfrak{B}})) \leq \prod_{(s,a):\pi_{E_1}(s,a)=0} g(s, a). \tag{27}$$

---

[12]Notice that, to obtain an empty feasible region, it is sufficient that the two experts behave identically in a single state-action pair. This is a direct consequence of the fact that the sub-optimality constraint is imposed with equality for each state-action pair.

[13]Notice, in this sense, that, if any other deterministic sub-optimal expert $j$ is present, if it holds that $\pi_{E_j}(a|s) = \pi_{E_i}(a|s) = 1$, then we should have $\xi_i = \xi_j$ to avoid obtaining an empty reward set.

*Proof.* The proof of Equation (26) follows directly by noticing that $\|\zeta\|_\infty \le (1-\gamma)^{-1}$. Therefore, as a consequence:

$$\text{Vol}(Z(\mathfrak{B})) \le \prod_{(s,a):\pi_{E_1}(a|s)=0} \int_0^{(1-\gamma)^{-1}} 1 dx = \prod_{(s,a):\pi_{E_1}(s,a)=0} \frac{1}{1-\gamma}.$$

Equation (27), instead, follows from worst-case upper bounds on $\zeta(s,a)$ that arise from Equation (5). Specifically, as discussed in Section 3.3, it is possible to show that:

$$\zeta(s,a) \le g(s,a).$$

As a consequence, we have that:

$$\text{Vol}(Z(\bar{\mathfrak{B}})) \le \prod_{(s,a):\pi_{E_1}(a|s)=0} \int_0^{g(s,a)} 1 dx = \prod_{(s,a):\pi_{E_1}(s,a)=0} g(s,a),$$

which concludes the proof. $\qquad\square$

At this point, we recall that $k(s,a)$ is given by:

$$k(s,a) := \min_{i \in \mathcal{X}(s,a), s' \in \mathcal{S}} \frac{\xi_i}{d_{s'}^{\pi_{E_i}}(s)\pi_{E_i}(a|s)}.$$

Therefore, Proposition D.7 highlights that, for sufficiently small values of $\xi_i$, we obtain a notable reduction in the upper bounds of $\text{Vol}(Z(\bar{\mathfrak{B}}))$.

# E   Proofs and Derivations of Section 4

In this section, we provide formal proofs for the statements of Section 4. We first proceed with the proof of the lower bound (Theorem 4.1), and then we continue with the analysis US-IRL-SE (Theorem 4.2).

## E.1   Proof of Theorem 4.1

In this section, we prove Theorem 4.1. We recall that Theorem 4.1 is composed of two parts. The first one is related to learning reward functions that are compatible with the behavior of the optimal expert (i.e., Equation (9)), while the second one directly arises from the structure of the sub-optimal experts (i.e., Equation (11)).

We begin by proving Equation (9). As we have anticipated in Section 4.2, Equation (9) is directly connected with the problem of learning reward functions that are compatible with the behavior of the optimal expert $\pi_{E_1}$. In this sense, we recall that, recently, [22] have provided lower bounds for the single-agent IRL problem in finite-horizon MDPs. In our work, we provide an extension of these results for the infinite-horizon IRL formulation.

**Theorem E.1.** *Let $\mathfrak{A}$ be a $(\epsilon, \delta)$-correct algorithm for the IRL problem with sub-optimal experts. There exists a problem instance $\mathfrak{B}$ such that the expected sample complexity is lower bounded by:*

$$\mathbb{E}_{\mathfrak{A},\bar{\mathfrak{B}}}[\tau] \ge \Omega\left(\frac{SA}{\epsilon^2(1-\gamma)^2}\left(\log\left(\frac{1}{\delta}\right) + S\right)\right). \tag{28}$$

*Proof.* Similarly to [22], this results follows from two different lower bounds (i.e., Theorem E.2 and Theorem E.3), and by assuming to observe instances like the ones of Theorem E.2 with probability $\frac{1}{2}$ and instances like the ones of Theorem E.3 with probability $\frac{1}{2}$. $\qquad\square$

As discussed, Theorem E.1 follows from two intermediate results that leverage two different constructions (i.e., Theorem E.2 and Theorem E.3). We now delve into the proofs of these intermediate theorems.

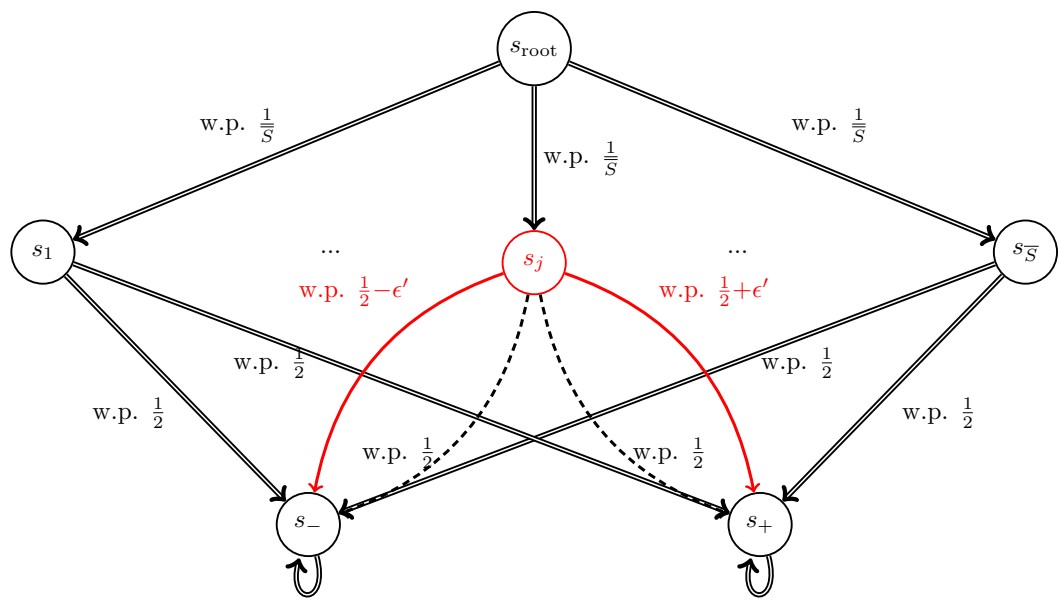

Figure 3: Representation of the IRL-SE problem for the instances used in Theorem E.2.

**Theorem E.2.** *Let $\mathfrak{A}$ be a $(\epsilon, \delta)$-correct algorithm for the IRL problem with sub-optimal experts. There exists a problem instance $\mathfrak{B}$ such that the expected sample complexity is lower bounded by:*

$$\mathbb{E}_{\mathfrak{A}, \mathfrak{B}}[\tau] \geq \Omega \left( \frac{SA}{\epsilon^2 (1 - \gamma)^2} \log \left( \frac{1}{\delta} \right) \right). \tag{29}$$

*Proof.* The proof is split into several steps. Since most of the arguments are borrowed from the proof of Theorem B.2 in [22], we only provide a short sketch that involves the main differences between the finite-horizon and the infinite horizon IRL model.

**Step 1: Base Instance and Alternative Instances**  We consider MDP\R instances that are presented in Figure 3. More specifically, the state space is given by $\mathcal{S} = \{s_{\text{root}}, s_1, \ldots, s_{\bar{S}}, s_-, s_+\}$, the action space is composed of $k$ actions $\mathcal{A} = \{a_1, a_2, \ldots, a_k\}$, the transition model is described below, and $\gamma \in (0, 1)$.

In state $s_{\text{root}}$ all actions have the same effect, and they lead, with probability $\frac{1}{\bar{S}}$ to a state in $\{s_1, \ldots, s_{\bar{S}}\}$. Similarly, in state $s_-$ and $s_+$ all actions have deterministic effect, and they all lead to $s_-$ and $s_+$ respectively. All the experts are deterministic and take action $a_1$ with probability 1 in all states. We then consider a base instance, where, in each state $s_j \in \{s_1, \ldots, s_{\bar{S}}\}$, all actions lead, with probability $\frac{1}{2}$ to $s_-$ and $s_+$.

We then consider a set of $\bar{S} \times A$ alternative instances by varying the behavior of the transition model in state-action pairs $(s_j, a_k) \in \{s_1, \ldots s_{\bar{S}}\} \times \mathcal{A}$. Specifically, by taking $a_k$, the agent will end up, with probability $\frac{1}{2} + \epsilon'$ in $s_+$, and, with probability $\frac{1}{2} - \epsilon'$ in $s_-$.

**Step 2: Feasible Reward Set and Hausdorff Distance lower bound**  At this point, we study the structure of the feasible reward set that is compatible with the instances we described. Specifically, we are interested in the behavior of the reward function related to actions taken in states $s_j \in \{s_1, \ldots s_{\bar{S}}\}$.

Specifically, for the base instance, we have:

$$r(s_j, a_1) + \frac{\gamma}{1 - \gamma} \left[ \frac{1}{2} r(s_-, a_1) + \frac{1}{2} r(s_+, a_1) \right] \geq r(s_j, a_k) + \frac{\gamma}{1 - \gamma} \left[ \frac{1}{2} r(s_-, a_1) + \frac{1}{2} r(s_+, a_1) \right],$$

which can be rewritten as:

$$r(s_j, a_1) \geq r(s_j, a_k).$$

For the alternative instance in which we varied the behavior of the state-action pair $(s_j, a_k)$, instead, we obtain:

$$r(s_j, a_1) + \frac{\gamma}{1-\gamma}\left[\frac{1}{2}r(s_-, a_1) + \frac{1}{2}r(s_+, a_1)\right] \geq$$
$$r(s_j, a_k) + \frac{\gamma}{1-\gamma}\left[\left(\frac{1}{2} - \epsilon'\right)r(s_-, a_1) + \left(\frac{1}{2} + \epsilon'\right)r(s_+, a_1)\right],$$

thus leading to:

$$r(s_j, a_1) \geq r(s_j, a_k) - \epsilon'\frac{\gamma}{1-\gamma}\left[r(s_-, a_1) - r(s_+, a_1)\right],$$

Given this construction, we can lower bound the Hausdorff distance between the feasible reward set of the base and the alternative instance. Specifically, we first pick a reward function $r'$ compatible with the alternative instance as follows: $r'(s_-, a_1) = 1, r'(s_+, a_1) = 0, r'(s_j, a_k) = 1$ and $r'(s_j, a_1) = 1 - \frac{\epsilon'\gamma}{1-\gamma}$. Then, we study which reward function compatible with the base instance minimizes the infinite norm w.r.t. $r'$. Given these choices, similarly to [22], it is possible to obtain the following lower-bound to the Hausdorff distance:

$$H_\infty(\mathcal{R}, \mathcal{R}') \geq \frac{\epsilon'\gamma}{1-\gamma},$$

where $\mathcal{R}$ denotes the feasible reward set of the base instance, while $\mathcal{R}'$ denotes the feasible reward set of the alternative instance. We enforce the following constraint on the previous equation:

$$\frac{\epsilon'\gamma}{1-\gamma} \geq 2\epsilon,$$

which, in turns, leads to the following requirement on $\epsilon'$:

$$\epsilon' \geq \frac{2\epsilon(1-\gamma)}{\gamma}.$$

**Step 3: Lower bounding the sample complexity**  At this point, the rest of the proof follows identical to Theorem B.2 in [22], and leads to the desired result.

$\square$

We now continue with the proof of second intermidiate result that is needed for the proof of Theorem E.1.

**Theorem E.3.** *Let $\mathfrak{A}$ be a $(\epsilon, \delta)$-correct algorithm for the IRL problem with sub-optimal experts. There exists a problem instance $\mathfrak{B}$ such that the expected sample complexity is lower bounded by:*

$$\mathbb{E}_{\mathfrak{A},\mathfrak{B}}[\tau] \geq \Omega\left(\frac{S^2 A}{\epsilon^2(1-\gamma)^2}\right). \tag{30}$$

*Proof.* The proof is split into several steps. Since most of the arguments are borrowed from the proof of Theorem B.3 in [22], we only provide a short sketch that involves the main differences between the finite-horizon and the infinite horizon IRL model.

**Step 1: Base Instance and Alternative Instances**  We consider MDP\R instances that are presented in Figure 4. More specifically, the state space is given by $\mathcal{S} = \{s_{\text{root}}, s_1, \dots, s_{\bar{S}}, s'_1, \dots, s'_{\bar{S}}\}$, the action space is composed of $k$ actions $\mathcal{A} = \{a_1, a_2, \dots, a_k\}$, the transition model is described below, and $\gamma \in (0, 1)$. In the following, we will assume $\bar{S}$ to be divisible by 16.

In state $s_{\text{root}}$ all actions have the same effect, and they lead, with probability $\frac{1}{\bar{S}}$ to a state in $\{s_1, \dots, s_{\bar{S}}\}$. In state $s \in \{s'_1, \dots, s'_{\bar{S}}\}$ all actions have deterministic effect, and they all lead to the same state in which the action is taken. All the experts are deterministic and take action $a_1$

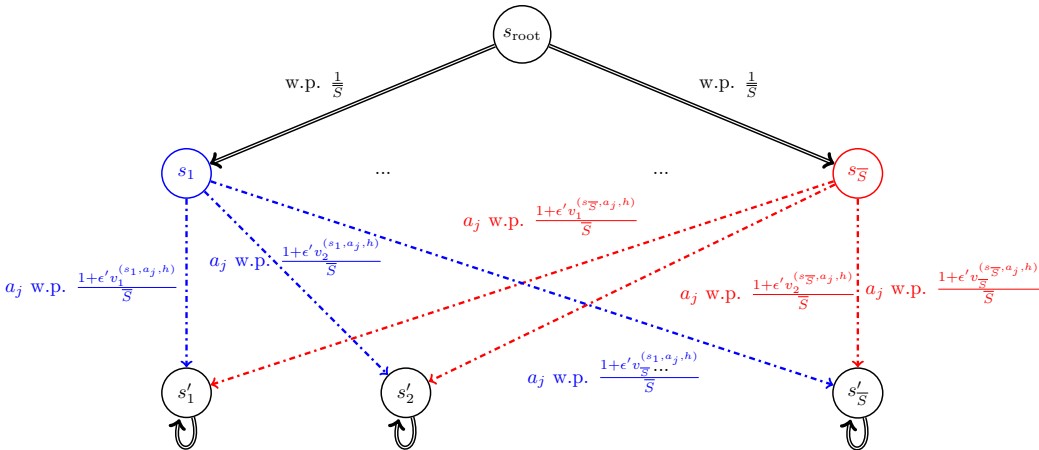

Figure 4: Representation of the IRL-SE problem for the instances used in Theorem E.3.

with probability 1 in all states. Finally, in all states $s \in \{s_1, \dots s_{\bar{S}}\}$, action $a_1$ leads w.p. $\frac{1}{\bar{S}}$ to any state in $s'_{\bar{S}} \in \{s'_1, \dots, s'_{\bar{S}}\}$.

We then consider a class of instances by varying the behavior of the transition model in state-action pairs $(s_j, a_k) \in \{s_1, \dots s_{\bar{S}}\} \times \mathcal{A} \setminus \{a_1\}$. Specifically, for each triplet $(s_j, s'_i, a_k)$, $p(s'_i|s_j, a_k) = \frac{1+\epsilon' v_i^{(s_j, a_k)}}{2}$, where $v^{(s_j,k)} = \left( v_1^{(s_i, a_j)}, \dots v_{\bar{S}}^{(s_i, a_j)} \right) \in \mathcal{V}$, and $\mathcal{V} = \left\{ \{-1, 1\}^{\bar{S}} : \sum_i^{\bar{S}} v_j = 0 \right\}$ and $\epsilon' \in \left[ 0, \frac{1}{2} \right]$.

**Step 2: Feasible Reward Set and Hausdorff Distance lower bound**    We now proceed with lower bounding the Hausdorff distance between the feasible reward set that is induced by two instances that belongs to $\mathcal{V}$, namely $\mathcal{R}_{\bar{\mathfrak{B}}_v}$ and $\mathcal{R}_{\bar{\mathfrak{B}}_w}$. To this end, first of all, we notice that our class of reward functions admits elements that are bounded in $[0, 1]$. Let us denote with $\bar{\mathcal{R}}_{\bar{\mathfrak{B}}_v}$ the class of compatible rewards functions that are bounded in $[-1, 1]$. Then, we have that:

$$H_\infty \left( \mathcal{R}_{\bar{\mathfrak{B}}_v}, \mathcal{R}_{\bar{\mathfrak{B}}_w} \right) \geq \frac{1}{2} H_\infty \left( \bar{\mathcal{R}}_{\bar{\mathfrak{B}}_v}, \bar{\mathcal{R}}_{\bar{\mathfrak{B}}_w} \right). \tag{31}$$

The proof of Equation (31) follows by the following reasonings. First of all, we can see that, for any $v, \bar{r} \in \bar{\mathcal{R}}_{\bar{\mathfrak{B}}_v}$ holds if and only if $\frac{\bar{r}+1}{2} \in \mathcal{R}_{\bar{\mathfrak{B}}_v}$ holds.[14] Then, with simple algebraic manipulations, we have that:

$$\sup_{x \in \mathcal{R}_{\bar{\mathfrak{B}}_v}} \inf_{y \in \mathcal{R}_{\bar{\mathfrak{B}}_w}} \|x - y\|_\infty = \sup_{x \in \bar{\mathcal{R}}_{\bar{\mathfrak{B}}_v}} \inf_{y \in \bar{\mathcal{R}}_{\bar{\mathfrak{B}}_w}} \left\| \frac{x+1}{2} - \frac{y+1}{2} \right\|_\infty$$

$$= \frac{1}{2} \sup_{x \in \bar{\mathcal{R}}_{\bar{\mathfrak{B}}_v}} \inf_{y \in \bar{\mathcal{R}}_{\bar{\mathfrak{B}}_w}} \|x - y\|_\infty,$$

from which it follows Equation (31).

At this point, our analysis follows by lower bounding the Hausdorff distance using reward functions that are bounded in $[-1, 1]$. To this end, first of all, we analyze the feasible reward set for a single instance $\bar{\mathfrak{B}}_v$. In this case, since $a_1$ is played by the optimal expert, we have that:

$$r(s_j, a_1) + \frac{1}{\bar{S}} \frac{\gamma}{1-\gamma} \sum_{s'_i} r^v(s'_i) \geq r(s_j, a_k) + \frac{1}{\bar{S}} \frac{\gamma}{1-\gamma} \sum_{s'_i} \left( 1 + \epsilon' v_i \right) r^v(s'_i),$$

---

[14] Notice that we are considering single-experts IRL problems. Indeed, in the considered IRL-SE problem, all sub-optimal experts behave identically to the optimal agent, and the constraints of Equation (5) are vacuous.

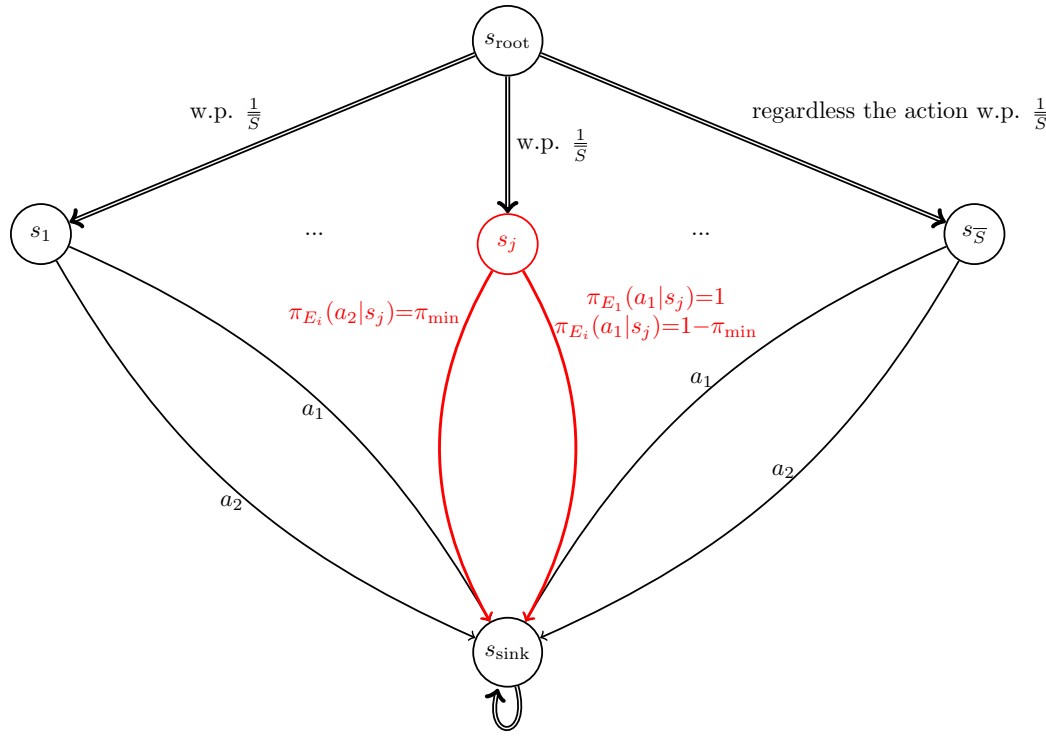

Figure 5: Representation of the IRL-SE problem for the instances used in Theorem E.4.

which, in turns, leads to:

$$r(s_j, a_1) \geq r(s_j, a_k) + \epsilon' \frac{1}{\bar{S}} \frac{\gamma}{1-\gamma} \sum_{s_i'} v_i r^v(s_i').$$

At this point, following the same steps of [22], we obtain:

$$H_\infty \left( \mathcal{R}_{\bar{\mathfrak{B}}_v}, \mathcal{R}_{\bar{\mathfrak{B}}_w} \right) \geq \frac{1}{2} H_\infty \left( \bar{\mathcal{R}}_{\bar{\mathfrak{B}}_v}, \bar{\mathcal{R}}_{\bar{\mathfrak{B}}_w} \right) \geq \frac{\epsilon'}{64} \frac{\gamma}{1-\gamma},$$

on which we enforce the following constraint:

$$\frac{\epsilon'}{64} \frac{\gamma}{1-\gamma} \geq 2\epsilon,$$

that is $\epsilon' \geq 128\epsilon \frac{1-\gamma}{\gamma}$.

**Step 3: Lower-bounding the sample complexity** At this point, the rest of the proof follows identical to Theorem B.3 in [22], and leads to the desired result.

$\square$

We continue by proving the novel statement related to Equation (11), which captures the role of the estimating reward functions that are compatible with the behavior of the sub-optimal experts.

**Theorem E.4.** *Let $\mathfrak{A}$ be a $(\epsilon, \delta)$-correct algorithm for the IRL problem with sub-optimal experts. For sufficiently small values of $\epsilon$, there exists an instance $\mathfrak{B}$ in which $q_0 < 1$ such that the expected sample complexity is lower bounded by:*

$$\mathbb{E}_{\mathfrak{A}, \mathfrak{B}}[\tau] \geq \Omega \left( \frac{q_0^2 S \log \left( \frac{1}{\delta} \right)}{\epsilon^2 \pi_{\min}} \right). \tag{32}$$

*Proof.* The proof follows as the combination of several steps. We first describe the construction considering the case in which there is a single sub-optimal expert. Then, the results will follow by iterating the procedure for each sub-optimal expert in the given set.

**Step 1: Base Instance and Alternative Instance**   We consider MDP\R instances that are presented in Figure 5. More specifically, the state space is given by $\mathcal{S} = \{s_{\text{root}}, s_1, \ldots, s_{\bar{S}}, s_{\text{sink}}\}$, the action space is composed of two actions $\mathcal{A} = \{a_1, a_2\}$, the transition model is described below, and $\gamma \in (0, 1)$.

In state $s_{\text{root}}$ all actions have the same effect, and they lead, with probability $\frac{1}{\bar{S}}$ to a state in $\{s_1, \ldots, s_{\bar{S}}\}$. In each state $s_i \in \{s_1, \ldots, s_{\bar{S}}\}$, again, all actions have the same effect, and they lead, deterministically to state $s_{\text{sink}}$. In state $s_{\text{sink}}$ all actions are deterministic and they all lead to $s_{\text{sink}}$.

We then consider a base instance, where, in states $s_{\text{root}}$ and state $s_{\text{sink}}$ all experts take action $a_1$ with probability 1. In each state $s_j \in \{s_1, \ldots, s_{\bar{S}}\}$, instead, $\pi_{E_1}(a_1|s_j) = 1$, while $\pi_{E_i}(a_1|s_j) = 1 - \pi_{\min}$ and $\pi_{E_i}(a_2|s_j) = \pi_{\min}$ for the sub-optimal expert.

We then consider a set of $\bar{S}$ alternative instances by varying the behavior of the sub-optimal expert in state $s_j \in \{s_1, \ldots s_{\bar{S}}\}$. Specifically, the sub-optimal expert will act accordingly to the following policy: $\pi_{E_i}(a_1|s_j) = 1 - \alpha\pi_{\min}$, and $\pi_{E_i}(a_2|s_j) = \alpha\pi_{\min}$ for some $\alpha > 1$ (to be defined later). We will denote our set of instances as $\mathbb{M} = \left\{\pi_{E_i}^{(l)} : l \in [\bar{S}]\right\}$, and 0 denotes the behavior of the base instance we described above.

**Step 2: Feasible Reward Set**   At this point, we study the structure of the feasible reward set that is compatible with the each instance we described. Specifically, we are interested in the properties of feasible reward functions in states $s_j \in \{s_1, \ldots, s_{\bar{S}}\}$.

For the base instance, in each state $s_j \in \{s_1, \ldots, s_{\bar{S}}\}$, we have that, since $a_1$ is the action taken by the optimal expert:

$$r(s_j, a_1) + \frac{\gamma}{1-\gamma}r(s_{\text{sink}}, a_1) \geq r(s_j, a_2) + \frac{\gamma}{1-\gamma}r(s_{\text{sink}}, a_1),$$

which reduces to:

$$r(s_j, a_1) \geq r(s_j, a_2). \tag{33}$$

Moreover, due to the presence of the sub-optimal expert, we have that $V_{\mathcal{M}\cup r}^{\pi_{E_1}^{(0)}}(s_j) \leq V_{\mathcal{M}\cup r}^{\pi_{E_i}}(s_j) + \xi_i$, which results in:

$$r(s_j, a_1) + \frac{\gamma}{1-\gamma}r(s_{\text{sink}}, a_1) \leq$$

$$(1 - \pi_{\min})\left(r(s_j, a_1) + \frac{\gamma}{1-\gamma}r(s_{\text{sink}}, a_1)\right) + \pi_{\min}\left(r(s_j, a_2) + \frac{\gamma}{1-\gamma}r(s_{\text{sink}}, a_1)\right) + \xi_i,$$

which can be rewritten as:

$$r(s_j, a_1) - r(s_j, a_2) \leq \frac{\xi_i}{\pi_{\min}}. \tag{34}$$

Similarly, for each alternative instance in $\mathbb{M}$, we obtain:

$$r(s_j, a_1) \geq r(s_j, a_2) \tag{35}$$

$$r(a_1, s_j) - r(s_j, a_2) \leq \frac{\xi_i}{\alpha\pi_{\min}}. \tag{36}$$

**Step 3: Lower-bounding the Hausdorff Distance**   At this point, we proceed by lower bounding the Hausdorff distance between the feasible reward set of the base instance, namely $\mathcal{R}_{\mathfrak{B}(0)}$, and that of alternative instance $j$ in which the policy of the sub-optimal expert is modified in state $s_j$, namely $\mathcal{R}_{\mathfrak{B}(j)}$.

To this end, we notice that it is sufficient to pick any $r \in \mathcal{R}_{\mathfrak{B}(0)}$, and then study the following optimization problem:

$$\inf_{r' \in \mathcal{R}_{\mathfrak{B}(j)}} \|r - r'\|_\infty. \tag{37}$$

We pick the following choice for $r$. For action $a_1$ in state $s_j$ we pick $r(s_j, a_1) = \frac{\xi_i}{\pi_{\min}}$. Instead, for all the other state-action pair, we pick $r(\cdot, \cdot) = 0$.[15] In this case, one can verify that the optimization problem in (37) can be rewritten as follows:

$$
\begin{aligned}
\inf_{x,y} \quad & \max\left\{\left|\frac{\xi_i}{\pi_{\min}} - x\right|, y\right\} \\
\text{s.t.} \quad & x, y \in [0, 1] \\
& x - y \leq \frac{\xi_i}{\alpha \pi_{\min}} \\
& x \geq y,
\end{aligned}
\tag{38}
$$

where $x$ corresponds to $r'(s_j, a_1)$, and $y$ to $r'(s_j, a_2)$. The optimization problem in (38) can be further lower-bounded by:

$$
\begin{aligned}
\inf_{x,y} \quad & \left|\frac{\xi_i}{\pi_{\min}} - x\right| \\
\text{s.t.} \quad & x, y \in [0, 1] \\
& x - y \leq \frac{\xi_i}{\alpha \pi_{\min}} \\
& x \geq y \\
& \left|\frac{\xi_i}{\pi_{\min}} - x\right| \geq y
\end{aligned}
\tag{39}
$$

We study the optimization problem in (39) by cases (i.e., $\xi_i \pi_{\min}^{-1} < x$, and $\xi_i \pi_{\min}^{-1} \geq x$). First, we suppose that $\xi_i \pi_{\min}^{-1} < x$. However, in this case, we would have $y \leq x - \xi_i \pi_{\min}^{-1}$, that is $x \geq y + \xi_i \pi_{\min}^{-1}$. Chaining this inequality with $x \leq y + \xi_i (\alpha \pi_{\min})^{-1}$ yields:

$$
y + \frac{\xi_i}{\pi_{\min}} \leq x \leq y + \frac{\xi_i}{\alpha \pi_{\min}},
$$

which is impossible since $\alpha > 1$. Concerning the case in which $\xi_i \pi_{\min}^{-1} \geq x$, instead, we have that $y \leq \xi_i \pi_{\min}^{-1} - x$. It follows that, since $x \leq \xi_i \pi_{\min}^{-1} + y$, the maximum value of $x$ will be attained for the maximum value of $y$. Therefore, we obtain:

$$
x \leq \frac{\xi_i}{\alpha \pi_{\min}} + \frac{\xi_i}{\pi_{\min}} - x,
$$

Thus leading to:

$$
x \leq \frac{1}{2} \frac{\xi_i}{\pi_{\min}}\left(\frac{1}{\alpha} + 1\right).
$$

Plugging this result into the objective function, we obtain the solution to optimization problem (39), which in turn provides a lower bound to the Hausdorff distance. More specifically, we have that:

$$
H_\infty\left(\mathcal{R}_{\mathfrak{B}(0)}, \mathcal{R}_{\mathfrak{B}(j)}\right) \geq \frac{1}{2} \frac{\xi_i}{\pi_{\min}}\left(1 - \frac{1}{\alpha}\right).
\tag{40}
$$

We enforce the following constraint on this quantity, that is:

$$
\frac{1}{2} \frac{\xi_i}{\pi_{\min}}\left(1 - \frac{1}{\alpha}\right) = 2\epsilon.
\tag{41}
$$

We notice that in this sense, we need to impose $\alpha = \xi_i (\xi_i - 4\epsilon \pi_{\min})^{-1}$, which results being greater than 1 for all values of $\xi_i > 4\epsilon \pi_{\min}$. Furthermore, since $\alpha \pi_{\min} < 1$,[16] we obtain the following condition on the values of $\epsilon$ which guarantes that the construction is valid:

$$
\epsilon < \min\left\{\frac{\xi_i}{4\pi_{\min}}, \frac{1 - \xi_i}{4}\right\}.
$$

---

[15]Notice that in this step we have used the assumption that $q_0 < 1$.

[16]This condition directly arises from the fact that $\alpha \pi_{\min}$ represents the probability with which the sub-optimal expert takes a certain action.

**Step 4: Lower-bounding Probability**  At this point, consider an $(\epsilon, \delta)$-correct algorithm $\mathfrak{A}$ that outputs the correct feasible set $\mathcal{R}_{\widehat{\mathfrak{B}}}$ for each IRL-SE problem $\bar{\mathfrak{B}}$. For $\mathfrak{A}$ it holds that:

$$\delta \geq \sup_{\bar{\mathfrak{B}}} \mathbb{P}_{\bar{\mathfrak{B}},\mathfrak{A}} \left( H\left(\mathcal{R}_{\bar{\mathfrak{B}}}, \mathcal{R}_{\widehat{\mathfrak{B}}}\right) \geq \epsilon \right) \tag{42}$$

$$\geq \max_{\mathfrak{B} \in \mathbb{M}} \mathbb{P}_{\bar{\mathfrak{B}},\mathfrak{A}} \left( H\left(\mathcal{R}_{\bar{\mathfrak{B}}}, \mathcal{R}_{\widehat{\mathfrak{B}}}\right) \geq \epsilon \right), \tag{43}$$

where $\mathbb{P}_{\bar{\mathfrak{B}},\mathfrak{A}}$ denotes the joint probability distribution of all events realized by the execution of algorithm $\mathfrak{A}$ in the IRL-SE problem $\bar{\mathfrak{B}}$, that is:

$$\mathbb{P}_{\bar{\mathfrak{B}},\mathfrak{A}} = \prod_{t=1}^{\tau} \nu_t\left(s_t, a_t | \mathcal{D}_{t-1}\right) p(s_t'|s_t, a_t) \prod_{i=1}^{n+1} \pi_{E_i}(a_t^{(i)}|s_t)$$

At this point, following the same reasonings of, e.g., Theorem B.2 in [22], one can see that, for all the alternative instances in $l \in \mathbb{M}$, the previous equation can be further lower-bounded to obtain:

$$\delta \geq \frac{1}{4} \exp\left(-\mathrm{KL}\left(\mathbb{P}_{\bar{\mathfrak{B}}^{(0)},\mathfrak{A}}, \mathbb{P}_{\bar{\mathfrak{B}}^{(l)},\mathfrak{A}}\right)\right), \tag{44}$$

**Step 5: KL-divergence**  As a final step, we will proceed by analyzing Equation (46). We will start by analyzing the KL divergence between $\mathbb{P}_{\bar{\mathfrak{B}}^{(0)},\mathfrak{A}}$ and $\mathbb{P}_{\bar{\mathfrak{B}}^{(0)},\mathfrak{A}}$. Specifically, we have that:

$$\mathrm{KL}\left(\mathbb{P}_{\bar{\mathfrak{B}}^{(0)},\mathfrak{A}}, \mathbb{P}_{\bar{\mathfrak{B}}^{(l)},\mathfrak{A}}\right) = \mathbb{E}_{\bar{\mathfrak{B}}^{(0)},\mathfrak{A}} \left[ \sum_{t=1}^{\tau} \mathrm{KL}\left(\pi_{E_i}^{(0)}(\cdot|s_t), \pi_{E_i}^{(l)}(\cdot|s_t)\right) \right]$$

$$\leq \mathbb{E}_{\bar{\mathfrak{B}}^{(0)},\mathfrak{A}} \left[N_\tau(s_j)\right] \frac{\mathrm{TV}\left(\pi_{E_i}^{(0)}(\cdot|s_j), \pi_{E_i}^{(l)}(\cdot|s_j)\right)^2}{\pi_{\min}}$$

$$\leq \mathbb{E}_{\bar{\mathfrak{B}}^{(0)},\mathfrak{A}} \left[N_\tau(s_j)\right] \frac{(\alpha\pi_{\min} - \pi_{\min})^2}{\pi_{\min}},$$

where $\mathrm{TV}(p_1, p_2)$ represents the total variation distance between distribution $p_1$ and $p_2$. The first inequality step follows from the reverse Pinkser's inequality [33], and the second one, instead, from the definition of the total variation distance.

At this point, however, from Equation (41), we obtain that:

$$\alpha\pi_{\min} - \pi_{\min} = \frac{4\alpha\pi_{\min}^2 \epsilon}{\xi_i},$$

thus leading to:

$$\mathrm{KL}\left(\mathbb{P}_{\bar{\mathfrak{B}}^{(0)},\mathfrak{A}}, \mathbb{P}_{\bar{\mathfrak{B}}^{(l)},\mathfrak{A}}\right) \leq \mathbb{E}_{\bar{\mathfrak{B}}^{(0)},\mathfrak{A}} \left[N_\tau(s_j)\right] \frac{16\alpha^2 \pi_{\min}^3 \epsilon^2}{\xi_i^2} \tag{45}$$

Plugging this result into Equation (46), and rearranging the terms, we obtain:

$$\mathbb{E}_{\bar{\mathfrak{B}}^{(0)},\mathfrak{A}} \left[N_\tau(s_j)\right] \geq \frac{\log\left(\frac{1}{4\delta}\right)\xi_i^2}{16\epsilon^2 \pi_{\min}^3 \alpha^2} \geq \frac{\xi_i^2 \log\left(\frac{1}{4\delta}\right)}{16\epsilon^2 \pi_{\min}^3}. \tag{46}$$

Finally, to conclude, we notice that:

$$\mathbb{E}_{\bar{\mathfrak{B}}^{(0)},\mathfrak{A}} [\tau] \geq \sum_{s \in \{s_1, \ldots, s_{\bar{S}}\}} \mathbb{E}_{\bar{\mathfrak{B}}^{(0)},\mathfrak{A}} \left[N_\tau(s)\right] \tag{47}$$

$$\geq \frac{(S-2)\xi_i^2}{16\epsilon^2 \pi_{\min}^3} \log\left(\frac{1}{4\delta}\right) \tag{48}$$

Iterating this procedure over all the possible experts yields the desired result.

$$\square$$

Finally, we are now ready to prove Theorem 4.1.

**Theorem 4.1.** *Let $\mathfrak{A}$ be a $(\epsilon, \delta)$-correct algorithm for the IRL problem with sub-optimal experts. There exists a problem instance $\mathfrak{B}$ such that the expected sample complexity is lower bounded by:*

$$\mathbb{E}_{\mathfrak{A},\mathfrak{B}}[\tau] \geq \Omega\left(\frac{SA}{\epsilon^2(1-\gamma)^2}\left(\log\left(\frac{1}{\delta}\right) + S\right)\right), \tag{9}$$

*where $\Omega(\cdot)$ hides constant dependencies. Let $\pi_{\min} > 0$ be:*

$$\pi_{\min} := \min_{i\in\{2,\dots n+1\dots\}} \min_{(s,a)\in\mathcal{S}\times\mathcal{A}:\pi_{E_i}(a|s)>0} \pi_{E_i}(a|s), \tag{10}$$

*and let $q_0 := \pi_{\min}^{-1}\max_{i\in\{2,\dots,n+1\}} \xi_i$. Then, there exists an instance $\bar{\mathfrak{B}}'$ in which $q_0 < 1$ such that:*

$$\mathbb{E}_{\mathfrak{A},\bar{\mathfrak{B}}'}[\tau] \geq \Omega\left(\frac{q_0^2 S\log\left(\frac{1}{\delta}\right)}{\epsilon^2\pi_{\min}}\right). \tag{11}$$

*Proof.* The proof follows directly by combining Theorem E.1 and Theorem E.4. $\square$

## E.2 Proof of Theorem 4.2

In this section, we provide a formal proof for Theorem 4.2.

First of all, we begin by defining a good event $\mathcal{E}$ that holds with probability at least $1 - \delta$. To this end, we begin by reporting, for the sake of completeness, the concentration tools that are used in controlling the probability with which $\mathcal{E}$ holds.

**Lemma E.5** (Multiplicative Chernoff bound [10]). *Consider $t$ independent random variables $X_1,\dots,X_t$ taking values in $[0,1]$. Suppose that $\mathbb{E}[X_i] = \mu$ holds for all $i \in \{1,\dots,t\}$. Consider $\alpha \in (0,1)$. Then, we have that:*

$$\mathbb{P}\left(\sum_{i=1}^t X_i \geq t\mu(1+\alpha)\right) \leq \exp\left(-\frac{1}{3}t\mu\alpha^2\right).$$

**Lemma E.6** (Proposition 1 in [14]). *Let $p$ be a categorical distribution over the simplex of dimension $y$, and let $\hat{p}$ be the maximum likelihood estimate of $p$ obtained with $t \geq 1$ independent samples. Then, for all $\delta \in (0,1)$ it holds that:*

$$\mathbb{P}\left(\exists t \geq 1 : t\mathrm{KL}\left(\hat{p},p\right) > \log\left(\frac{1}{\delta}\right) + (y-1)\log\left(e\left(1+\frac{t}{y-1}\right)\right)\right) \leq \delta,$$

*where $\mathrm{KL}(q_1, q_2)$ denotes the Kullback–Leibler divergence between distributions $q_1$ and $q_2$.*

At this point, we proceed by defining our good event. In the following, the subscript $t$ is used to denote the iteration number of the US-IRL-SE algorithm.

**Lemma E.7** (Good event). *Consider $t$ such that:*

$$\sqrt{\frac{3\log\left(\frac{3SAn}{\delta}\right)}{\pi_{E_i}(a|s)N_t(s)}} < 1 \tag{49}$$

*holds for all $i \in \{2,\dots,n+1\}$ and for all $(s,a) \in \mathcal{S} \times \mathcal{A}$ such that $\pi_{E_i}(a|s) > 0$. Let us define the following events:*

$$\mathcal{E}_p = \bigcap_{(s,a)\in\mathcal{S}\times\mathcal{A}} \left\{N_t(s,a)\mathrm{KL}\left(\hat{p}_t(\cdot|s,a), p(\cdot|s,a)\right) \leq \log\left(\frac{3SAn}{\delta}\right) + (S-1)\log\left(e\left(1+\frac{N_t(s,a)}{S-1}\right)\right)\right\}$$

$$\mathcal{E}_\pi = \bigcap_{i\in\{2,\dots,n+1\}} \bigcap_{s\in\mathcal{S}} \left\{N_t(s)\mathrm{KL}\left(\hat{\pi}_{E_i,t}(\cdot|s), \pi_{E_i}(\cdot|s)\right) \leq \log\left(\frac{3SAn}{\delta}\right) + (A-1)\log\left(e\left(1+\frac{N_t(s)}{A-1}\right)\right)\right\}$$

$$\mathcal{E}_{\pi_{\min}} = \bigcap_{i\in\{2,\dots,n+1\}} \bigcap_{(s,a)\in\mathcal{S}\times\mathcal{A}:\pi_{E_i}(a|s)>0} \left\{\hat{\pi}_{E_i,t}(a|s) \leq \pi_{E_i}(a|s)\left(1+\sqrt{\frac{3\log\left(\frac{3SAn}{\delta}\right)}{\pi_{E_i}(a|s)N_t(s)}}\right)\right\}$$

*Consider $\mathcal{E} = \mathcal{E}_p \cap \mathcal{E}_\pi \cap \mathcal{E}_{\pi_{\min}}$. Then, it holds that $\mathbb{P}(\mathcal{E}) \geq 1 - \delta$.*

*Proof.* The proof follows by controlling the probability of the complementary event $\mathcal{E}^c$. More specifically, we have that:

$$\mathbb{P}\left(\mathcal{E}^c\right) = \mathbb{P}\left(\mathcal{E}_p^c \cup \mathcal{E}_\pi^c \cup \mathcal{E}_{\pi_{\min}}^c\right) \leq \mathbb{P}\left(\mathcal{E}_p^c\right) + \mathbb{P}\left(\mathcal{E}_\pi^c\right) + \mathbb{P}\left(\mathcal{E}_{\pi_{\min}}^c\right),$$

where in the last passage we have used Boole's inequality. At this point, we will use Lemma E.5 and Lemma E.6 to control $\mathbb{P}\left(\mathcal{E}_p^c\right) + \mathbb{P}\left(\mathcal{E}_\pi^c\right) + \mathbb{P}\left(\mathcal{E}_{\pi_{\min}}^c\right)$. More specifically, concerning $\mathcal{E}_p^c$, we have that:

$$\begin{aligned}
\mathbb{P}\left(\mathcal{E}_p^c\right) &\leq \sum_{(s,a)\in\mathcal{S}\times\mathcal{A}} \mathbb{P}\Bigg( N_t(s,a)\mathrm{KL}\left(\hat{p}_t(\cdot|s,a), p(\cdot|s,a)\right) > \log\left(\frac{3SAn}{\delta}\right) \\
&\qquad\qquad + (S-1)\log\left(e\left(1+\frac{N_t(s,a)}{S-1}\right)\right)\Bigg) \\
&\leq \sum_{(s,a)\in\mathcal{S}\times\mathcal{A}} \frac{\delta}{3SAn} \leq \frac{\delta}{3},
\end{aligned}$$

where, in the first step we have applied Boole's inequality, and in the second one Lemma E.6. From identical reasoning, we can upper bound $\mathbb{P}\left(\mathcal{E}_\pi^c\right)$, thus obtaining:

$$\mathbb{P}\left(\mathcal{E}_\pi^c\right) \leq \frac{\delta}{3}.$$

Finally, concerning $\mathcal{E}_{\pi_{\min}}^c$, we have that:

$$\begin{aligned}
\mathbb{P}\left(\mathcal{E}_{\pi_{\min}}^c\right) &\leq \sum_{i\in\{2,\ldots,n+2\}} \sum_{(s,a):\pi_{E_i}(a|s)>0} \mathbb{P}\left( \hat{\pi}_{E_i,t}(a|s) > \pi_{E_i}(a|s)\left(1 + \sqrt{\frac{3\log\left(\frac{3SAn}{\delta}\right)}{\pi_{E_i}(a|s)N_t(s)}}\right)\right) \\
&\leq \sum_{i\in\{2,\ldots,n+2\}} \sum_{(s,a):\pi_{E_i}(a|s)>0} \frac{\delta}{3SAn} \leq \frac{\delta}{3},
\end{aligned}$$

where, in the first step we have used Boole's inequality, and in the second one Lemma E.5 together with Equation (49). At this point, combining these results, we obtain $\mathbb{P}\left(\mathcal{E}^c\right) \leq \delta$, thus concluding the proof. $\square$

At this point, it has to be remarked that Lemma E.7 holds whenever:

$$N_t(s) > \frac{3\log\left(\frac{3SAn}{\delta}\right)}{\pi_{\min}}.$$

We remark that this term is due to the requirement that Lemma E.5 requires $\alpha$ in $(0,1)$. As we shall see, however, this will be sufficient to carry out the theoretical analysis of US-IRL-SE.

We now proceed by presenting error propagation results that study the Hausdorff distance between $\mathcal{R}_{\bar{\mathfrak{B}}}$ and $\mathcal{R}_{\widehat{\bar{\mathfrak{B}}}}$. Before diving into the details, we introduce the following notation. We denote with $\Psi$ the set of $\zeta$'s that are compatible with the linear constraints of Equation (5). More specifically:

$$\Psi := \left\{\zeta \in \mathbb{R}_{\geq 0}^{S\times A} : \forall i \in \{2,\ldots,n+1\} \quad d^{\pi_{E_i}}\pi_{E_i}\bar{B}^{\pi_{E_1}}\zeta \leq \mathbf{1}_S\xi_i\right\}. \tag{50}$$

Similarly, we denote with $\widehat{\Psi}$, the set of $\zeta$'s that are compatible with the linear constraints of Equation (5) that are induced by the empirical IRL problem $\widehat{\bar{\mathfrak{B}}}$. More precisely:

$$\hat{\Psi} := \left\{\zeta \in \mathbb{R}_{\geq 0}^{S\times A} : \forall i \in \{2,\ldots,n+1\} \quad \hat{d}^{\hat{\pi}_{E_i}}\hat{\pi}_{E_i}\bar{B}^{\hat{\pi}_{E_1}}\zeta \leq \mathbf{1}_S\xi_i\right\}, \tag{51}$$

where $\hat{d}^\pi$ denotes the discounted expected occupancy of policy $\pi$ under the transition model $\hat{p}$.

At this point, we provide a preliminary Lemma that will be used to study the Hausdorff distance.

**Lemma E.8** (Error Propagation). *Let $\bar{\mathfrak{B}}$ be an IRL-SE problem and let $\widehat{\bar{\mathfrak{B}}}$ be its empirical estimate. Then, for any $r \in \mathcal{R}_{\bar{\mathfrak{B}}}$ such that $r = -\bar{B}^{\pi_{E_1}}\zeta + (E - \gamma P)V$, and $(I - \gamma\pi_{E_i}P)^{-1}\pi_{E_i}\bar{B}^{\pi_{E_1}}\zeta \leq \xi_i$, there exists $\hat{r} \in \mathcal{R}_{\widehat{\bar{\mathfrak{B}}}}$ such that element-wise it holds that:*

$$|r - \hat{r}| \leq \bar{B}^{\pi_{E_1}}B^{\hat{\pi}_{E_1}}\zeta + \gamma|(P - \hat{P})V| + \bar{B}^{\hat{\pi}_{E_1}}\bar{B}^{\pi_{E_1}}\|\zeta - \mathrm{proj}_{\widehat{\Psi}}(\zeta)\|_\infty\mathbf{1}_{S\times\mathcal{A}}, \tag{52}$$

*where $\mathrm{proj}_{\widehat{\Psi}}(\cdot)$ denotes the infinite norm projection on the set $\widehat{\Psi}$. More formally, for $\zeta \in \mathbb{R}^{S\times A}$:*

$$\mathrm{proj}_{\widehat{\Psi}}(\zeta) = \underset{x\in\widehat{\Psi}}{\arg\min} \|x - \zeta\|_\infty$$

*Proof.* From Theorem 3.6, we know that we can express the reward functions as:

$$r = -\bar{B}^{\pi_{E_1}}\zeta + (E - \gamma P)V,$$
$$\hat{r} = -\bar{B}^{\hat{\pi}_{E_1}}\hat{\zeta} + (E - \gamma \hat{P})\hat{V},$$

where $V$ and $\hat{V}$ belongs to $\mathbb{R}^{\mathcal{S}}$, and $\zeta, \hat{\zeta} \in \mathbb{R}_{\geq 0}^{\mathcal{S} \times \mathcal{A}}$ satisfies the following equations for all $i \in \{2, \ldots, n+1\}$:

$$(I - \gamma \pi_{E_i} P)^{-1} \pi_{E_i} \bar{B}^{\pi_{E_1}} \zeta \leq \xi_i,$$
$$(I - \gamma \hat{\pi}_{E_i} \hat{P})^{-1} \hat{\pi}_{E_i} \bar{B}^{\hat{\pi}_{E_1}} \hat{\zeta} \leq \xi_i.$$

At this point, we recall that, for any $r \in \mathcal{R}_{\mathfrak{B}}$, we are interested in a specific reward function $\hat{r} \in \mathcal{R}_{\widehat{\mathfrak{B}}}$ that is sufficiently close to $r$. For this reason, we pick $\hat{V} = V$, and $\hat{\zeta} = \bar{B}^{\pi_{E_1}} \text{proj}_{\widehat{\Psi}}(\zeta)$.[17] Plugging these choices within $|r - \hat{r}|$, and applying the triangular inequality, we obtain:

$$\left| r - \hat{r} \right| \leq \left| -(\bar{B}^{\pi_{E_1}}\zeta + \bar{B}^{\hat{\pi}_{E_1}}\hat{\zeta}) \right| + \gamma \left| (P - \hat{P})V \right|. \tag{53}$$

Let us focus now on $\left| -(\bar{B}^{\pi_{E_1}}\zeta + \bar{B}^{\hat{\pi}_{E_1}}\hat{\zeta}) \right|$.

$$\left| -(\bar{B}^{\pi_{E_1}}\zeta + \bar{B}^{\hat{\pi}_{E_1}} \bar{B}^{\pi_{E_1}} \text{proj}_{\widehat{\Psi}}(\zeta)) \right| = \left| -(\bar{B}^{\pi_{E_1}}\zeta + \bar{B}^{\hat{\pi}_{E_1}} \bar{B}^{\pi_{E_1}} \text{proj}_{\widehat{\Psi}}(\zeta)) \pm \bar{B}^{\hat{\pi}_{E_1}} \bar{B}^{\pi_{E_1}} \zeta \right|$$

$$\leq \left| -\bar{B}^{\pi_{E_1}}\zeta + \bar{B}^{\hat{\pi}_{E_1}} \bar{B}^{\pi_{E_1}} \zeta \right| + \left| \bar{B}^{\hat{\pi}_{E_1}}\hat{\zeta} - \bar{B}^{\hat{\pi}_{E_1}} \bar{B}^{\pi_{E_1}} \zeta \right|$$

$$= \left| -\bar{B}^{\pi_{E_1}} B^{\hat{\pi}_{E_1}} \zeta \right| + \left| \bar{B}^{\hat{\pi}_{E_1}} \bar{B}^{\pi_{E_1}} \text{proj}_{\widehat{\Psi}}(\zeta)) - \bar{B}^{\hat{\pi}_{E_1}} \bar{B}^{\pi_{E_1}} \zeta \right|,$$

where in the last step we have used $B^{\pi} + \bar{B}^{\pi} = I_{\mathcal{S} \times \mathcal{A}}$.[18] We now focus on the last term:

$$\left| \bar{B}^{\hat{\pi}_{E_1}} \bar{B}^{\pi_{E_1}} \text{proj}_{\widehat{\Omega}}(\zeta)) - \bar{B}^{\hat{\pi}_{E_1}} \bar{B}^{\pi_{E_1}} \zeta \right| = \bar{B}^{\hat{\pi}_{E_1}} \bar{B}^{\pi_{E_1}} \left| \zeta - \text{proj}_{\widehat{\Omega}}(\zeta) \right|$$

$$\leq \bar{B}^{\hat{\pi}_{E_1}} \bar{B}^{\pi_{E_1}} \| \zeta - \text{proj}_{\widehat{\Omega}}(\zeta) \|_{\infty} \mathbf{1}_{\mathcal{S} \times \mathcal{A}}.$$

Plugging these results within Equation (53) concludes the proof. $\qquad \square$

Before providing some interpretation to Lemma E.8, we recall the definition of the Hausdorff distannce we are interested in:

$$H_{\infty}(\mathcal{R}_{\bar{\mathfrak{B}}}, \mathcal{R}_{\widehat{\mathfrak{B}}}) = \max \left\{ \sup_{r \in \mathcal{R}_{\bar{\mathfrak{B}}}} \inf_{\hat{r} \in \mathcal{R}_{\widehat{\mathfrak{B}}}} \| r - \hat{r} \|_{\infty}, \sup_{\hat{r} \in \mathcal{R}_{\widehat{\mathfrak{B}}}} \inf_{r \in \mathcal{R}_{\bar{\mathfrak{B}}}} \| r - \hat{r} \|_{\infty} \right\}. \tag{54}$$

Given this definition, we can appreciate how Lemma E.8 can be used to upper-bound the error of the first component of the Hausdorff distance, namely:

$$\sup_{r \in \mathcal{R}_{\bar{\mathfrak{B}}}} \inf_{\hat{r} \in \mathcal{R}_{\widehat{\mathfrak{B}}}} \| r - \hat{r} \|_{\infty}.$$

Nevertheless, it is possible to obtain a symmetric version of Lemma E.8 that analyzes the existence of an $r \in \mathcal{R}_{\bar{\mathfrak{B}}}$ that is sufficiently close to any $\hat{r} \in \mathcal{R}_{\widehat{\mathfrak{B}}}$. At this point, we notice that Lemma E.8 decomposes the error on the recovered reward set as the sum of three different components. The first one is related to the error in the estimation of the optimal policy $\pi_{E_1}$, and, to zero it out, it is sufficient to estimate for each state one action that is played by the optimal expert. The second one, instead, is related to the errors in the estimation of the transition model, and, finally, the last one is related to the presence of multiple and sub-optimal experts. More precisely, we notice that the first two components are present also for the single-agent setting [24], and, in this sense, they arise from the difficulty of estimating reward functions that are compatible with Equation (4). The last term, instead, is related to the presence of sub-optimal experts, and it denotes the infinite norm between any $\zeta \in \Psi$ and its infinite-norm projection to the set $\widehat{\Psi}$. As one can imagine, and as our proofs will reveal, studying how this last source of error decreases with the iterations of US-IRL-SE introduces significant challenges in the analysis w.r.t. the single agent setting. Precisely, this complexity will be tackled within the proof of the following Lemma, which analyzes how the error of Equation (52) decreases with the number of iterations of US-IRL.

---

[17]Notice that, if $x \in \widehat{\Psi}$, $\bar{B}^{\pi_{E_1}} x$ belongs to $\widehat{\Psi}$ as well.

[18]Consider $\mathbb{R}^{\mathcal{S} \times \mathcal{A}}$, then $B^{\pi}$ is the operator defined as $(B^{\pi}g)(s, a) = \mathbb{1}\{\pi(a|s) > 0\} g(s, a)$.

**Lemma E.9** (High-Probability Error Propagation). *Let $t$ be the iteration of US-IRL. Suppose that $N_t(s,a) \geq 1$ for all $(s,a) \in \mathcal{S} \times \mathcal{A}$ and, furthermore, suppose that $t$ is such that:*

$$t \geq \frac{3 \log\left(3SAn\delta^{-1}\right)}{A\pi_{\min}} \tag{55}$$

$$t \geq \frac{8\gamma^2}{(1-\gamma)^2} \Bigg[ \log\left(\frac{3SAn}{\delta}\right) \tag{56}$$

$$+ (S-1)\log\left(\frac{64\gamma^4}{(1-\gamma)^4}\left(\log\left(\frac{3SAn}{\delta}\right) + (S-1)\left(\sqrt{e} + \sqrt{\frac{1}{S-1}}\right)^2\right)\right)\Bigg] \tag{57}$$

$$t \geq \frac{8\gamma^2}{A(1-\gamma)^2} \Bigg[ \log\left(\frac{3SAn}{\delta}\right) \tag{58}$$

$$+ (A-1)\log\left(\frac{64\gamma^4}{(1-\gamma)^4}\left(\log\left(\frac{3SAn}{\delta}\right) + (A-1)\left(\sqrt{e} + \sqrt{\frac{1}{A-1}}\right)^2\right)\right)\Bigg]. \tag{59}$$

*Then let $\bar{\mathfrak{B}}$ be an IRL-SE problem and let $\widehat{\bar{\mathfrak{B}}}$ be its empirical estimate after $t$ iteration of US-IRL. Then, with probability at least $1 - \delta$, for any $r \in \mathcal{R}_{\bar{\mathfrak{B}}}$, there exists $\hat{r} \in \mathcal{R}_{\widehat{\bar{\mathfrak{B}}}}$ such that:*

$$\|r - \hat{r}\|_\infty \leq \frac{2\sqrt{2}\gamma}{1-\gamma}\beta_t + \left(\rho_t + \frac{2\sqrt{2}\gamma}{1-\gamma}(\alpha_t + \beta_t)\right)\min\left\{\pi_{\min}^{-1}\max_i \xi_i, (1-\gamma)^{-1}\right\}, \tag{60}$$

*where:*

$$\beta_t := \sqrt{\frac{\log\left(\frac{3SAn}{\delta}\right) + (S-1)\log\left(e\left(1 + \frac{t}{S-1}\right)\right)}{t}} \tag{61}$$

$$\alpha_t := \sqrt{\frac{\log\left(\frac{3SAn}{\delta}\right) + (A-1)\log\left(e\left(1 + \frac{tA}{A-1}\right)\right)}{tA}} \tag{62}$$

$$\rho_t := \sqrt{\frac{3\log\left(\frac{3SAn}{\delta}\right)}{\pi_{\min}tA}}. \tag{63}$$

*Proof.* The proof follows by analyzing in greater detail the result of Lemma E.8. Indeed, from Lemma E.8, we know that there exists $\hat{r} \in \mathcal{R}_{\widehat{\bar{\mathfrak{B}}}}$ such that:

$$\|r - \hat{r}\|_\infty \leq \|\bar{B}^{\pi_{E_1}} B^{\hat{\pi}_{E_1}}\zeta\|_\infty + \gamma\|(P - \hat{P})V\|_\infty + \|\zeta - \text{proj}_{\widehat{\Psi}}(\zeta)\|_\infty \tag{64}$$

We split the analysis of Equation (64) into two three parts. First of all, we will focus on the two first terms, that are $\|\bar{B}^{\pi_{E_1}} B^{\hat{\pi}_{E_1}}\zeta\|_\infty$ and $\gamma\|(P - \hat{P})V\|_\infty$, and then we will tackle the most challenging aspect, that is $\|\zeta - \text{proj}_{\widehat{\Psi}}(\zeta)\|_\infty$.

Concerning $\|\bar{B}^{\pi_{E_1}} B^{\hat{\pi}_{E_1}}\zeta\|_\infty$, we notice that, whenever $N_t(s,a) > 1$ holds, we have that:

$$\bar{B}^{\pi_{E_1}} B^{\hat{\pi}_{E_1}}\zeta = 0,$$

for any possible value of $\zeta$. This is a direct consequence of the fact that $\pi_{E_1}$ is deterministic.

Secondly, let us focus on $\gamma\|(P - \hat{P})V\|_\infty$. First of all, we notice that, since $N_t(s) > 3\log\left(3SAn\delta^{-1}\right)\pi_{\min}$ holds for all $s \in \mathcal{S}$, then $\mathcal{E}$ holds with probability at least $1 - \delta$ (Lemma E.7).

At this point, conditioning on $\mathcal{E}$, we have that:

$$
\begin{aligned}
\gamma \|(P - \hat{P})V\|_\infty &= \gamma \max_{s,a} \Big| \sum_{s' \in \mathcal{S}} (p(s'|s,a) - \hat{p}(s'|s,a))V(s') \Big| \\
&\leq \frac{\gamma}{1-\gamma} \max_{s,a} \Big| \sum_{s' \in \mathcal{S}} (p(s'|s,a) - \hat{p}(s'|s,a)) \Big| \\
&\leq \frac{\gamma}{1-\gamma} \max_{s,a} \|p(\cdot|s,a) - \hat{p}(\cdot|s,a)\|_1 \\
&\leq \frac{2\sqrt{2}\gamma}{(1-\gamma)} \max_{s,a} \sqrt{\mathrm{KL}(\hat{p}(\cdot|s,a), p(\cdot|s,a))} \\
&\leq \frac{2\sqrt{2}\gamma}{1-\gamma} \max_{s,a} \sqrt{\frac{\log\left(\frac{3SAn}{\delta}\right) + (S-1)\log\left(e\left(1 + \frac{N_t(s,a)}{S-1}\right)\right)}{N_t(s,a)}},
\end{aligned}
$$

where the first inequality follows the fact that $\|V\|_\infty \leq \frac{1}{1-\gamma}$, the third one by Pinksker's inequality, and the last one from Lemma E.7. At this point, since US-IRL gathers samples uniformly from each state-action pair, we have that $N_t(s,a) = t$ for all $(s,a) \in \mathcal{S} \times \mathcal{A}$. Thus leading to:

$$
\gamma \|(P - \hat{P})V\|_\infty \leq \frac{2\sqrt{2}\gamma}{1-\gamma} \sqrt{\frac{\log\left(\frac{3SAn}{\delta}\right) + (S-1)\log\left(e\left(1 + \frac{t}{S-1}\right)\right)}{t}} := \frac{2\sqrt{2}\gamma}{1-\gamma}\beta_t.
$$

Finally, we focus on $\|\zeta - \mathrm{proj}_{\widehat{\Psi}}(\zeta)\|_\infty$. To upper-bound this last term we proceed in several steps.

**Step 1: Relationship between $\Psi$ and $\widehat{\Psi}$** First of all, we begin by finding a more explicit relationship between $\Psi$ and $\widehat{\Psi}$. To this end, we recall that $\zeta \in \mathbb{R}_{\geq 0}^{S \times A}$ belongs to $\widehat{\Psi}$ if the following condition is satisfied for all $i \in \{2, \ldots, n+1\}$:

$$
\hat{d}^{\hat{\pi}_{E_i}} \hat{\pi}_{E_i} \bar{B}^{\hat{\pi}_{E_1}} \zeta \leq \mathbf{1}_\mathcal{S} \xi_i.
$$

Under the assumption that $N_t(s,a) \geq 1$, since the expert policy $\pi_{E_1}$ is deterministic, the previous Equation can be equivalently rewritten as:

$$
\hat{d}^{\hat{\pi}_{E_i}} \hat{\pi}_{E_i} \bar{B}^{\pi_{E_1}} \zeta \leq \mathbf{1}_\mathcal{S} \xi_i.
$$

At this point, we proceed with some algebraic manipulations of the left-hand side of the previous Equation. Specifically:

$$
\begin{aligned}
\hat{d}^{\hat{\pi}_{E_i}} \hat{\pi}_{E_i} \bar{B}^{\pi_{E_1}} \zeta &= \left(\hat{d}^{\hat{\pi}_{E_i}} \pm d^{\pi_{E_i}}\right) \hat{\pi}_{E_i} \bar{B}^{\pi_{E_1}} \zeta \\
&= \left(\hat{d}^{\hat{\pi}_{E_i}} - d^{\pi_{E_i}}\right) \hat{\pi}_{E_i} \bar{B}^{\pi_{E_1}} \zeta + d^{\pi_{E_i}} \hat{\pi}_{E_i} \bar{B}^{\pi_{E_1}} \zeta \\
&= \left(\hat{d}^{\hat{\pi}_{E_i}} - d^{\pi_{E_i}}\right) \hat{\pi}_{E_i} \bar{B}^{\pi_{E_1}} \zeta + d^{\pi_{E_i}} \left(\hat{\pi}_{E_i} \pm \pi_{E_i}\right) \bar{B}^{\pi_{E_1}} \zeta \\
&= \left(\hat{d}^{\hat{\pi}_{E_i}} - d^{\pi_{E_i}}\right) \hat{\pi}_{E_i} \bar{B}^{\pi_{E_1}} \zeta + d^{\pi_{E_i}} \left(\hat{\pi}_{E_i} - \pi_{E_i}\right) \bar{B}^{\pi_{E_1}} \zeta + d^{\pi_{E_i}} \pi_{E_i} \bar{B}^{\pi_{E_1}} \zeta
\end{aligned}
$$

At this point, focus on $\hat{d}^{\hat{\pi}_{E_i}} - d^{\pi_{E_i}}$:

$$
\begin{aligned}
\hat{d}^{\hat{\pi}_{E_i}} - d^{\pi_{E_i}} &= (I_\mathcal{S} - \gamma\hat{\pi}_{E_i}\hat{P})^{-1} I_\mathcal{S} - (I_\mathcal{S} - \gamma\pi_{E_i}P)^{-1} I_\mathcal{S} \\
&= (I_\mathcal{S} - \gamma\pi_{E_i}P)^{-1} \left[(I_\mathcal{S} - \gamma\pi_{E_i}P) - (I_\mathcal{S} - \gamma\hat{\pi}_{E_i}\hat{P})\right] \hat{d}^{\hat{\pi}_{E_i}} \\
&= \gamma d^{\pi_{E_i}} \left[\hat{\pi}_{E_i}\hat{P} - \pi_{E_i}P\right] \hat{d}^{\hat{\pi}_{E_i}}
\end{aligned}
$$

At this point, plugging this result into the previous Equation, we obtain that $\zeta$ belongs to $\widehat{\Psi}$ if and only if the following holds for all experts $i \in \{2, \ldots, n+1\}$:

$$
d^{\pi_{E_i}} \pi_{E_i} \bar{B}^{\pi_{E_1}} \zeta + \gamma d^{\pi_{E_i}} \left[\hat{\pi}_{E_i}\hat{P} - \pi_{E_i}P\right] \hat{d}^{\hat{\pi}_{E_i}} \hat{\pi}_{E_i} \bar{B}^{\pi_{E_1}} \zeta + d^{\pi_{E_i}} \left(\hat{\pi}_{E_i} - \pi_{E_i}\right) \bar{B}^{\pi_{E_1}} \zeta \leq \mathbf{1}_\mathcal{S} \xi_i.
\tag{65}
$$

As we can appreciate, Equation (65) relates the sets $\Psi$ and $\widehat{\Psi}$ by an explicit relationship. Indeed, by renaming $\epsilon_1(\zeta) := \gamma d^{\pi_{E_i}} \left[ \hat{\pi}_{E_i} \hat{P} - \pi_{E_i} P \right] \hat{d}^\pi \hat{\pi}_{E_i} \bar{B}^{\pi_{E_1}} \zeta$ and $\epsilon_2(\zeta) := d^{\pi_{E_i}} \left( \hat{\pi}_{E_i} - \pi_{E_i} \right) \bar{B}^{\pi_{E_1}} \zeta$, Equation (65) can be rewritten as:

$$d^{\pi_{E_i}} \pi_{E_i} \bar{B}^{\pi_{E_1}} \zeta + \epsilon_1(\zeta) + \epsilon_2(\zeta) \leq \mathbf{1}_{\mathcal{S}} \xi_i,$$

which closely resambles the definition of $\Psi$.

**Step 2: Restricting the set $\widehat{\Psi}$**  We continue our proof by defining a new set $\widetilde{\Psi}$ such that $\widetilde{\Psi} \subseteq \widehat{\Psi}$ with probability at least $1 - \delta$. Indeed, by definition of the projection according to the infinite norm, this allows to upper-bound $\| \zeta - \text{proj}_{\widehat{\Psi}}(\zeta) \|_\infty$. More specifically, for any $\widetilde{\Psi} \subseteq \widehat{\Psi}$, we have that:

$$\| \zeta - \text{proj}_{\widehat{\Psi}}(\zeta) \|_\infty \leq \| \zeta - \text{proj}_{\widetilde{\Psi}}(\zeta) \|_\infty. \tag{66}$$

More specifically, in order to define $\widetilde{\Psi}$, we will first proceed by upper bounding $\epsilon_1(\zeta)$ and $\epsilon_2(\zeta)$. Precisely, consider $\zeta \in \widehat{\Psi}$. For $\epsilon_1(\zeta)$ we have that:

$$\begin{aligned}
\epsilon_1(\zeta) &= \gamma d^{\pi_{E_i}} \left[ \hat{\pi}_{E_i} \hat{P} - \pi_{E_i} P \right] \hat{d}^{\hat{\pi}_{E_i}} \hat{\pi}_{E_i} \bar{B}^{\pi_{E_1}} \zeta \\
&= \gamma \left[ d^{\pi_{E_i}} \left( \hat{\pi}_{E_i} - \pi_{E_i} \right) \hat{P} \right] \hat{d}^{\hat{\pi}_{E_i}} \hat{\pi}_{E_i} \bar{B}^{\pi_{E_1}} \zeta + \gamma \left[ d^{\pi_{E_i}} \pi_{E_i} (\hat{P} - P) \right] \hat{d}^{\hat{\pi}_{E_i}} \hat{\pi}_{E_i} \bar{B}^{\pi_{E_1}} \\
&\leq \gamma \left[ d^{\pi_{E_i}} \left| \hat{\pi}_{E_i} - \pi_{E_i} \right| \hat{P} \right] \hat{d}^{\hat{\pi}_{E_i}} \hat{\pi}_{E_i} \bar{B}^{\pi_{E_1}} \zeta + \gamma \left[ d^{\pi_{E_i}} \pi_{E_i} \left| \hat{P} - P \right| \right] \hat{d}^{\hat{\pi}_{E_i}} \hat{\pi}_{E_i} \bar{B}^{\pi_{E_1}},
\end{aligned}$$

where, given $f \in \mathbb{R}^S$, and $g \in \mathbb{R}^{S \times A}$, $\left| \hat{\pi} - \pi \right|$ denotes the operator defined as $\left| \hat{\pi} - \pi \right| g(s, a) = \sum_a \left| \hat{\pi}(a|s) - \pi(a|s) \right| g(s, a)$, and, similarly $\left| \hat{P} - P \right| f(s) = \sum_{s'} \left| \hat{p}(s'|s, a) - p(s'|s, a) \right| f(s')$. At this point, we notice that, since, for all $\zeta \in \widehat{\Psi}$, it holds that $\hat{d}^{\hat{\pi}_{E_i}} \hat{\pi}_{E_i} \bar{B}^{\pi_{E_1}} \leq \mathbf{1}_{\mathcal{S}} \xi_i$. Therefore, we can further upper bound the previous equation to obtain:

$$\epsilon_1(\zeta) \leq \gamma \left\| \left[ d^{\pi_{E_i}} \left| \hat{\pi}_{E_i} - \pi_{E_i} \right| \right] \mathbf{1}_{\mathcal{S} \times \mathcal{A}} \xi_i \right\|_\infty \mathbf{1}_{\mathcal{S}} + \gamma \left\| \left[ d^{\pi_{E_i}} \pi_{E_i} \left| \hat{P} - P \right| \right] \mathbf{1}_{\mathcal{S}} \xi_i \right\|_\infty \mathbf{1}_{\mathcal{S}}. \tag{67}$$

At this point, let us focus on $\gamma \left\| \left[ d^{\pi_{E_i}} \left| \hat{\pi}_{E_i} - \pi_{E_i} \right| \right] \mathbf{1}_{\mathcal{S} \times \mathcal{A}} \xi_i \right\|_\infty$:

$$\begin{aligned}
\gamma \left\| \left[ d^{\pi_{E_i}} \left| \hat{\pi}_{E_i} - \pi_{E_i} \right| \right] \mathbf{1}_{\mathcal{S} \times \mathcal{A}} \xi_i \right\|_\infty &\leq \gamma \xi_i \max_{s'} \sum_s d_{s'}^{\pi_{E_i}}(s) \sum_a \left| \hat{\pi}_{E_i}(a|s) - \pi_{E_i}(a|s) \right| \\
&= \gamma \xi_i \max_{s'} \sum_s d_{s'}^{\pi_{E_i}}(s) \| \hat{\pi}_{E_i}(\cdot|s) - \pi_{E_i}(\cdot|s) \|_1 \\
&\leq \frac{2\sqrt{2}\gamma \xi_i}{1 - \gamma} \max_{s'} \sqrt{\text{KL}(\hat{\pi}_{E_i}(\cdot, s'), \pi_{E_i}(\cdot, s'))} \\
&\leq \frac{2\sqrt{2}\gamma \xi_i}{1 - \gamma} \max_s \sqrt{\frac{\log\left(\frac{3SAn}{\delta}\right) + (A-1)\log\left(e\left(1 + \frac{N_t(s)}{S-1}\right)\right)}{N_t(s)}} \\
&= \frac{2\sqrt{2}\gamma \xi_i}{1 - \gamma} \sqrt{\frac{\log\left(\frac{3SAn}{\delta}\right) + (A-1)\log\left(e\left(1 + \frac{tA}{A-1}\right)\right)}{tA}} \\
&:= \frac{2\sqrt{2}\gamma \xi_i}{1 - \gamma} \alpha_t,
\end{aligned}$$

where in the third step we have used Pinkser's inequality, in the fourth we have used Lemma E.7, and in the fifth one we have used the fact that, in US-IRL-SE, $N_t(s) = \sum_{a \in \mathcal{A}} N_t(s, a) = tA$. At this

point, focus on the second term of Equation (67), namely $\gamma \left|\left| \left[ d^{\pi_{E_i}} \pi_{E_i} \left| \hat{P} - P \right| \right] \mathbf{1}_{\mathcal{S}} \xi_i \right|\right|_\infty \mathbf{1}_{\mathcal{S}}$:

$$\gamma \left|\left| \left[ d^{\pi_{E_i}} \pi_{E_i} \left| \hat{P} - P \right| \right] \mathbf{1}_{\mathcal{S}} \xi_i \right|\right|_\infty \leq \gamma \xi_i \max_{s'} \sum_s d_{s'}^{\pi_{E_i}}(s) \sum_a \pi_{E_i}(a|s) \| p(\cdot|s,a) - \hat{p}(\cdot|s,a) \|_1$$

$$\leq \frac{2\sqrt{2}\gamma\xi_i}{1-\gamma} \max_{s,a} \sqrt{\mathrm{KL}(\hat{p}(\cdot|s,a) - p(\cdot|s,a))}$$

$$\leq \frac{2\sqrt{2}\gamma\xi_i}{1-\gamma} \max_{s,a} \sqrt{\frac{\log\left(\frac{3SAn}{\delta}\right) + (S-1)\log\left(e\left(1 + \frac{N_t(s)}{S-1}\right)\right)}{N_t(s)}}$$

$$\leq \frac{2\sqrt{2}\gamma\xi_i}{1-\gamma} \sqrt{\frac{\log\left(\frac{3SAn}{\delta}\right) + (S-1)\log\left(e\left(1 + \frac{t}{S-1}\right)\right)}{t}}$$

$$:= \frac{2\sqrt{2}\gamma\xi_i}{1-\gamma} \beta_t,$$

where in the second step we have used Pinkser's inequality, in the third one Lemma E.7, and in the fourth one we have used the fact that, in US-IRL-SE, $N_t(s) = t$ for all states $s \in \mathcal{S}$. Therefore, plugging these results within Equation (67), we obtain an high-probability upper bound on $\epsilon_1(\zeta)$, that is:

$$\|\epsilon_1(\zeta)\|_\infty \leq \frac{2\sqrt{2}\gamma\xi_i}{1-\gamma} (\alpha_t + \beta_t). \tag{68}$$

We now proceed with similar reasoning to obtain an upper bound on $\epsilon_2(\zeta)$. Nevertheless, contrary to $\epsilon_1(\zeta)$, here we proceed with an element-wise upper bound on $\epsilon_2(\zeta)$. Specifically, for each state $s' \in \mathcal{S}$:

$$\epsilon_2(\zeta)(s') = \sum_s d_{s'}^{\pi_{E_i}}(s) \sum_{a:\pi_{E_1}(a|s)=0} \left| \hat{\pi}_{E_i}(a|s) - \pi_{E_i}(a|s) \right| \zeta(s,a)$$

$$= \sum_s d_{s'}^{\pi_{E_i}}(s) \sum_{\substack{a:\pi_{E_1}(a|s)=0, \\ \pi_{E_i}(a|s)>0}} \left| \hat{\pi}_{E_i}(a|s) - \pi_{E_i}(a|s) \right| \zeta(s,a)$$

$$= \sum_s d_{s'}^{\pi_{E_i}}(s) \sum_{\substack{a:\pi_{E_1}(a|s)=0, \\ \pi_{E_i}(a|s)>0}} \left| \frac{\hat{\pi}_{E_i}(a|s) - \pi_{E_i}(a|s)}{\pi_{E_i}(a|s)} \right| \pi_{E_i}(a|s) \zeta(s,a)$$

$$\leq \max_{(s'',a''):\pi_{E_i}(a''|s'')>0} \left| \frac{\hat{\pi}_{E_i}(a''|s'') - \pi_{E_i}(a''|s'')}{\pi_{E_i}(a''|s'')} \right| \sum_s d_{s'}^{\pi_{E_i}}(s) \sum_{a:\pi_{E_1}(a|s)=0} \pi_{E_i}(a|s) \zeta(s,a)$$

$$\leq \sqrt{\frac{3\log\left(\frac{3SAn}{\delta}\right)}{\pi_{\min} N_t(s)}} \sum_s d_{s'}^{\pi_{E_i}}(s) \sum_{a:\pi_{E_1}(a|s)=0} \pi_{E_i}(a|s) \zeta(s,a)$$

$$= \sqrt{\frac{3\log\left(\frac{3SAn}{\delta}\right)}{\pi_{\min} tA}} \sum_s d_{s'}^{\pi_{E_i}}(s) \sum_{a:\pi_{E_1}(a|s)=0} \pi_{E_i}(a|s) \zeta(s,a),$$

where the third step follows from the fact that, since $N_t(s) \geq 1$, $\hat{\pi}_{E_i}(a|s) = 0$ for all $(s,a)$ such that $\pi_{E_i}(a|s) = 0$, the fifth one, instead, from Lemma E.7, and the last one from the fact that, in US-IRL-SE, $N_t(s) = \sum_a N_t(s,a) = tA$.[19] Therefore, we have obtained an element-wise upper-bound on $\epsilon_2(\zeta)$ of the following form:

$$\epsilon_2(\zeta) \leq \sqrt{\frac{3\log\left(\frac{3SAn}{\delta}\right)}{\pi_{\min} tA}} d^{\pi_{E_i}} \pi_{E_i} \bar{B}^{\pi_{E_1}} \zeta := \rho_t d^{\pi_{E_i}} \pi_{E_i} \bar{B}^{\pi_{E_1}} \zeta. \tag{69}$$

---

[19]Notice that Equation (55) guarantees that we can apply Lemma E.7.

We are finally ready to define the set $\widetilde{\Psi}$. More specifically:

$$\widetilde{\Psi} := \left\{ \zeta \in \mathbb{R}_{\geq 0}^{S \times A} : (1 + \rho_t) d^{\pi_{E_i}} \pi_{E_i} \bar{B}^{\pi_{E_1}} \zeta \leq \xi_i - \frac{2\sqrt{2}\gamma\xi_i}{1 - \gamma} (\alpha_t + \beta_t) \right\}. \tag{70}$$

As one can verify, the definition of $\widetilde{\Psi}$ follows from upper-bounding $\epsilon_1(\zeta)$ and $\epsilon_2(\zeta)$ with Equations (68) and (69). As a direct consequence, whenever $\zeta \in \widetilde{\Psi}$, we have that $\zeta$ belongs to $\widehat{\Psi}$ as well.

**Step 3: Ensuring that the feasible region of $\widetilde{\Psi}$ is non-empty**   At this point, one might be tempted to directly study the projection of $\|\zeta - \text{proj}_{\widetilde{\Psi}}(\zeta)\|_\infty$. Nevertheless, we notice that, for sufficiently large values of $\alpha_t$ and $\beta_t$, $\widetilde{\Psi}$ might be empty.

Sufficient conditions to guarantee that $\widetilde{\Psi}$ is not empty are the following ones:

$$\frac{2\sqrt{2}\gamma\xi_i}{1 - \gamma} \alpha_t \leq \frac{\xi_i}{2} \tag{71}$$

$$\frac{2\sqrt{2}\gamma\xi_i}{1 - \gamma} \beta_t \leq \frac{\xi_i}{2}. \tag{72}$$

Using Lemma 12 in [14], it is possible to verify that Equations (56) and (58) are sufficient conditions for Equations (71)-(72) to hold.

**Step 4: Picking $\tilde{\zeta} \in \widetilde{\Psi}$ to upper-bound $\|\zeta - \text{proj}_{\widehat{\Psi}}(\zeta)\|_\infty$**   At this point, we are ready to conclude our proof. As we have previously verified, the set $\widetilde{\Psi}$ is non-empty. We can now study $\|\zeta - \text{proj}_{\widehat{\Psi}}(\zeta)\|_\infty$. More precisely, we have that:

$$\|\zeta - \text{proj}_{\widehat{\Psi}}(\zeta)\|_\infty \leq \|\zeta - \text{proj}_{\widetilde{\Psi}}(\zeta)\|_\infty.$$

To further upper bound this Equation, we notice that we can always pick, by definition of the infinite norm projection, any $\tilde{\zeta} \in \widetilde{\Psi}$. In other words, we have that:

$$\|\zeta - \text{proj}_{\widehat{\Psi}}(\zeta)\|_\infty \leq \|\zeta - \text{proj}_{\widetilde{\Psi}}(\zeta)\|_\infty \leq \|\zeta - \tilde{\zeta}\|_\infty.$$

More specifically, we choose $\tilde{\zeta}$ in the following way. If for all $i \in \{2, \ldots, n + 1\}$, $\pi_{E_i}(a|s) = 0$, then we pick $\tilde{\zeta}(s, a) = \zeta(s, a)$;[20] otherwise, we pick $\tilde{\zeta}(s, a)$ in the following way:

$$\tilde{\zeta}(s, a) = \frac{1 - \frac{2\sqrt{2}\gamma}{1 - \gamma} (\alpha_t + \beta_t)}{1 + \rho_t} \zeta(s, a). \tag{73}$$

First of all, we verify that this choice of $\tilde{\zeta}$ belongs to $\widetilde{\Psi}$. Plugging Equation (73) into the definition of $\widetilde{\Psi}$ we obtain that:

$$\left( 1 - \frac{2\sqrt{2}\gamma}{1 - \gamma} (\alpha_t + \beta_t) \right) \frac{1 + \rho_t}{1 + \rho_t} d^{\pi_{E_i}} \pi_{E_i} \bar{B}^{\pi_{E_1}} \zeta \leq \xi_i - \frac{2\sqrt{2}\gamma\xi_i}{1 - \gamma} (\alpha_t + \beta_t).$$

However, since $\zeta \in \Psi$, we have that $d^{\pi_{E_i}} \pi_{E_i} \bar{B}^{\pi_{E_1}} \zeta \leq \xi_i$, thus leading to:

$$\left( \xi_i - \frac{2\sqrt{2}\gamma\xi_i}{1 - \gamma} (\alpha_t + \beta_t) \right) \leq \xi_i - \frac{2\sqrt{2}\gamma\xi_i}{1 - \gamma} (\alpha_t + \beta_t),$$

which is always true, and, consequently $\tilde{\zeta} \in \widetilde{\Psi}$.

---

[20]Notice that, if for all $i \in \{2, \ldots, n + 1\}$, $\pi_{E_i}(a|s) = 0$, $\zeta(s, a)$ does not contribute to any of the linear constraints that introduced by Equation (5).

To conclude the proof, it remains to analyze $\|\zeta - \tilde{\zeta}\|_\infty$. At this point, we notice that:

$$\|\zeta - \tilde{\zeta}\|_\infty = \max_{\substack{(s,a):\pi_{E_1}(a|s)=0, \\ \exists i:\pi_{E_i}(a|s)>0}} \left| \zeta(s,a) - \frac{1 - \frac{2\sqrt{2}\gamma}{1-\gamma}(\alpha_t + \beta_t)}{1+\rho_t}\zeta(s,a) \right|,$$

Indeed, for a state-action pair $(s,a)$ such that $\pi_{E_1}(a|s) = 0$, we can notice that any value $\zeta(s,a)$ will not affect the resulting reward function.[21] Furthermore, whenever there is no sub-optimal expert such that $\pi_{E_i}(a|s) > 0$, then $\tilde{\zeta}(s,a) = \zeta(s,a)$ holds by definition. To conclude, with simple algebraic manipulations we can obtain the following result:

$$\|\zeta - \tilde{\zeta}\|_\infty \leq \max_{\substack{(s,a):\pi_{E_1}(a|s)=0, \\ \exists i:\pi_{E_i}(a|s)>0}} \left| (1+\rho_t)\zeta(s,a) - \left(1 - \frac{2\sqrt{2}\gamma}{1-\gamma}(\alpha_t + \beta_t)\right)\zeta(s,a) \right|$$

$$= \max_{\substack{(s,a):\pi_{E_1}(a|s)=0, \\ \exists i:\pi_{E_i}(a|s)>0}} \left| \left(\rho_t + \frac{2\sqrt{2}\gamma}{1-\gamma}(\alpha_t + \beta_t)\right)\zeta(s,a) \right|$$

$$= \left(\rho_t + \frac{2\sqrt{2}\gamma}{1-\gamma}(\alpha_t + \beta_t)\right) \max_{\substack{(s,a):\pi_{E_1}(a|s)=0, \\ \exists i:\pi_{E_i}(a|s)>0}} \left| \zeta(s,a) \right|$$

$$\leq \left(\rho_t + \frac{2\sqrt{2}\gamma}{1-\gamma}(\alpha_t + \beta_t)\right) \min\left\{\pi_{\min}^{-1}\max_i \xi_i, (1-\gamma)^{-1}\right\},$$

where in the last step we have upper-bounded $\zeta(s,a)$ with an upper bound that follows directly from Equation (7), thus concluding the proof. $\qquad\square$

We can appreciate as Lemma E.9 provides, under certain conditions on the time-step $t$, a high-probability upper bound on the difference $\inf_{\hat{r} \in \mathcal{R}_{\widehat{\mathfrak{B}}}} \|r - \hat{r}\|_\infty$ for any choice of $r \in \mathcal{R}_{\mathfrak{B}}$. Furthermore, as we can see, the proof is fairly involved due to the necessity of upper bounding the error term $\|\zeta - \text{proj}_{\widehat{\Psi}}(\zeta)\|_\infty$. At this point, by deriving symmetric results (i.e., Lemma E.8 and Lemma E.9), it is possible to derive an identical upper-bound for $\inf_{r \in \mathcal{R}_{\mathfrak{B}}} \|r - \hat{r}\|_\infty$ for any choice of $\hat{r} \in \mathcal{R}_{\widehat{\mathfrak{B}}}$. As a consequence, Lemma E.9 provides the following high-probability upper-bound on the Hausdorff distance between $\mathcal{R}_{\widehat{\mathfrak{B}}}$ and $\mathcal{R}_{\mathfrak{B}}$:

$$H_\infty\left(\mathcal{R}_{\mathfrak{B}}, \mathcal{R}_{\widehat{\mathfrak{B}}}\right) \leq \frac{2\sqrt{2}\gamma}{1-\gamma}\beta_t + \left(\rho_t + \frac{2\sqrt{2}\gamma}{1-\gamma}(\alpha_t + \beta_t)\right)\min\left\{\pi_{\min}^{-1}\max_i \xi_i, (1-\gamma)^{-1}\right\}. \quad (74)$$

**Theorem 4.2.** *Let* $q_1 = \min\left\{\pi_{\min}^{-1}\max_{i \in \{2,\dots,n+1\}} \xi_i, (1-\gamma)^{-1}\right\}$, *and* $q_2 = \max\{1, q_1\}$. *Then, with a total budget of:*

$$\widetilde{\mathcal{O}}\left(\max\left\{\frac{q_1^2 S \log\left(\frac{1}{\delta}\right)}{\epsilon^2 \pi_{\min}}, \frac{q_2^2 SA(S + \log\left(\frac{1}{\delta}\right))}{\epsilon^2 (1-\gamma)^2}\right\}\right), \quad (13)$$

*US-IRL-SE is* $(\epsilon, \delta)$-*correct and* $\widetilde{\mathcal{O}}(\cdot)$ *hides constant and logarithmic dependencies.*

*Proof.* Let us start from Lemma E.9 and consider $t$ such that the condition of Lemma E.9 are satisfied. As previously discussed, due to Lemma E.9, we have that, at time $t$, the algorithm US-IRL-SE induces an estimated feasible reward set such that the following holds with high-probability:

$$H_\infty\left(\mathcal{R}_{\mathfrak{B}}, \mathcal{R}_{\widehat{\mathfrak{B}}}\right) \leq \frac{2\sqrt{2}\gamma}{1-\gamma}\beta_t + \left(\rho_t + \frac{2\sqrt{2}\gamma}{1-\gamma}(\alpha_t + \beta_t)\right)\min\left\{\pi_{\min}^{-1}\max_i \xi_i, (1-\gamma)^{-1}\right\}.$$

---

[21]In other words, if $(s,a)$ is such that $\pi_{E_1}(a|s) = 0$, we can add the following constraint to any of the set we defined: $\zeta(s,a) = 0$. The resulting feasible reward set is left unchanged.

To conclude the proof, we need to find a sufficiently large $t$ such that the following holds:

$$H_\infty\left(\mathcal{R}_{\bar{\mathfrak{B}}}, \mathcal{R}_{\widehat{\mathfrak{B}}}\right) \leq \frac{2\sqrt{2}\gamma}{1-\gamma}\beta_t + \left(\rho_t + \frac{2\sqrt{2}\gamma}{1-\gamma}(\alpha_t + \beta_t)\right)\min\left\{\pi_{\min}^{-1}\max_i \xi_i, (1-\gamma)^{-1}\right\} \leq \epsilon,$$

for any $\epsilon \in (0,1)$. To this end, it is sufficient to find the smallest $t$ that satisfies the following conditions:

$$\rho_t \min\left\{\pi_{\min}^{-1}\max_i \xi_i, (1-\gamma)^{-1}\right\} \leq \frac{\epsilon}{4} \tag{75}$$

$$\frac{2\sqrt{2}\gamma}{(1-\gamma)}\beta_t \leq \frac{\epsilon}{4} \tag{76}$$

$$\frac{2\sqrt{2}\gamma}{(1-\gamma)}\min\left\{\pi_{\min}^{-1}\max_i \xi_i, (1-\gamma)^{-1}\right\}\beta_t \leq \frac{\epsilon}{4} \tag{77}$$

$$\frac{2\sqrt{2}\gamma}{(1-\gamma)}\min\left\{\pi_{\min}^{-1}\max_i \xi_i, (1-\gamma)^{-1}\right\}\alpha_t \leq \frac{\epsilon}{4} \tag{78}$$

At this point, let us focus on $\rho_t \min\left\{\pi_{\min}^{-1}\max_i \xi_i, (1-\gamma)^{-1}\right\} \leq \frac{\epsilon}{4}$. By simple algebraic manipulations we obtain the following sufficient condition for $t$:

$$tA \geq \frac{48q_1^2 \log\left(\frac{3SAn}{\delta}\right)}{\pi_{\min}\epsilon^2}.$$

For $\frac{2\sqrt{2}\gamma}{(1-\gamma)}\beta_t \leq \frac{\epsilon}{4}$, instead, we obtain the following condition on $t$:[22]

$$t \geq \frac{128\gamma^2}{(1-\gamma)^2\epsilon^2}\Bigg[\log\left(\frac{3SAn}{\delta}\right)$$
$$+ (S-1)\log\left(\frac{16384\gamma^4}{(1-\gamma)^4\epsilon^4}\left(\log\left(\frac{3SAn}{\delta}\right) + (S-1)\left(\sqrt{e} + \sqrt{\frac{1}{S-1}}\right)^2\right)\right)\Bigg].$$

Similarly, for $\frac{2\sqrt{2}\gamma}{(1-\gamma)}\min\left\{\pi_{\min}^{-1}\max_i \xi_i, (1-\gamma)^{-1}\right\}\beta_t \leq \frac{\epsilon}{4}$, we need at least the following number of samples:

$$t \geq \frac{128q_1^2\gamma^2}{(1-\gamma)^2\epsilon^2}\Bigg[\log\left(\frac{3SAn}{\delta}\right)$$
$$+ (S-1)\log\left(\frac{16384q_1^4\gamma^4}{(1-\gamma)^4\epsilon^4}\left(\log\left(\frac{3SAn}{\delta}\right) + (S-1)\left(\sqrt{e} + \sqrt{\frac{1}{S-1}}\right)^2\right)\right)\Bigg]$$

Finally, for $\frac{2\sqrt{2}\gamma}{(1-\gamma)}\min\left\{\pi_{\min}^{-1}\max_i \xi_i, (1-\gamma)^{-1}\right\}\alpha_t \leq \frac{\epsilon}{4}$, we obtain:

$$tA \geq \frac{128q_1^2\gamma^2}{(1-\gamma)^2\epsilon^2}\Bigg[\log\left(\frac{3SAn}{\delta}\right)$$
$$+ (A-1)\log\left(\frac{16384q_1^2\gamma^4}{(1-\gamma)^4\epsilon^4}\left(\log\left(\frac{3SAn}{\delta}\right) + (A-1)\left(\sqrt{e} + \sqrt{\frac{1}{A-1}}\right)^2\right)\right)\Bigg].$$

At this point, we notice that the total number of samples gathered from the generative at step $t$ is given by $tSA$. Therefore, from the previous equations, together with the conditions of Lemma E.9, we obtian the following sufficient condition to guarantee that US-IRL-SE is $(\epsilon, \delta)$-correct.

$$tSA = \mathcal{O}\left(\max\left\{\frac{q_1^2 S \log\left(\frac{1}{\delta}\right)}{\pi_{\min}^2\epsilon^2}, \frac{q_2^2 SA\left(S + \log\left(\frac{1}{\delta}\right)\right)}{(1-\gamma)^2\epsilon^2}, \frac{S \log\left(\frac{1}{\delta}\right)}{\pi_{\min}^2}\right\}\right), \tag{79}$$

---

[22]This follows by applying Lemma 12 in [14].

where the first term is due to Equation (75), the second one comes from Equations (76)-(77), and the last term arises from Equation (55), and is independent w.r.t. the desired accuracy $\epsilon$.[23] Specifically, we notice that for sufficiently small values of $\epsilon$ (i.e., $\epsilon < q_1$), the last term is always dominated by the first one, which concludes the proof.

$\square$

## F   Stochastic Optimal Expert

In this section, we discuss how to extend our analysis to the case in which the optimal expert plays a stochastic policy. First of all, let us define $\pi_{\min, E_1}$ as the minimum probability with which the optimal expert plays its actions.[24]

At this point, we notice that, for obtaining sample-complexity guarantees of the US-IRL-SE algorithm, Lemma E.8 implies that we are interested in learning *which are* the actions that are played with positive probability by the optimal expert. In other words, for any non-zero vector $x \in \mathbb{R}^{S \times A}$, we want the following to hold with high-probability:

$$\left| \left( \bar{B}^{\pi_{E_1}} - \bar{B}^{\hat{\pi}_{E_1}} \right) x \right| = 0. \tag{80}$$

To this end, one can apply Lemma D.3 of [22], which implies that with a number of samples that is:

$$\mathcal{O}\left( S \frac{\log\left(\frac{1}{\delta}\right)}{\log\left(1/(1 - \pi_{\min, E_1})\right)} \right), \tag{81}$$

Equation (80) holds w.p. at least $1 - \delta$. Once this condition is verified, the rest of the proof of the complexity of US-IRL-SE follows identically to the one of Theorem 4.2.[25] We notice that this introduces an additional maximum in the result of the sample-complexity. However, it is also possible to extend the proof of the lower bound of [22] (see Theorem D.1 in [22]), to show that the dependency resulting from (81) is unavoidable. In this sense, US-IRL-SE remains minimax optimal whenever the performance of the sub-optimal experts are sufficiently close to the ones of the optimal agent.

## G   Per-expert Complexity of IRL-SE

In this section, we provide a description of how to change the PAC learning formalism to show that Equation (11) actually represents a lower bound to the number of data that is necessary to gather from each of the different sub-optimal experts.

Specifically, we define a learning algorithm for an IRL problem $\bar{\mathfrak{B}}$ as a tuple $\mathfrak{A} = (\tau, \nu)$, where $\tau$ is a stopping time that controls the end of the data acquisition phase, and $\nu = (\nu_t)_{t \in \mathbb{N}}$ is a history-dependent sampling strategy over $\mathcal{S} \times \mathcal{A} \times (\mathcal{S})^{n+1}$. More precisely, $\nu_t \in \Delta_{\mathcal{D}_t}^{\mathcal{S} \times \mathcal{A} \times (\mathcal{S})^{n+1}}$, where $\mathcal{D}_t = \left( (\mathcal{S} \times \mathcal{A})^{n+2} \times \mathcal{S} \right)^t$. At each time step $t \in \mathbb{N}$, the algorithm selects, by means of $\nu_t$:

(i) a state-action pair $(S_t, A_t)$ and observes a sample $S_t' \sim p(\cdot|S_t, A_t)$ from the environment

(ii) a state $S_t^{(i)}$ for each expert $i \in \{1, \ldots, n+1\}$ and observes a sample $A_t^{(i)} \sim \pi_{E_i}(\cdot|S_t^{(i)})$

The observed realizations are then used to update the sampling strategy $\nu_t$, and the process goes on until the stopping rule is satisfied. At the end of the data acquisition phase, the algorithm leverages the collected data to output the estimate of the feasible reward set $\mathcal{R}_{\widehat{\mathfrak{B}}_\tau}$ that is induced by the resulting empirical IRL problem $\widehat{\mathfrak{B}}_\tau$. Given this formalism, we are interested in designing learning algorithms that, for any desired accuracy $\epsilon \in (0, 1)$ and any risk parameter $\delta \in (0, 1)$, guarantee that:

$$\mathbb{P}_{\mathfrak{A}, \bar{\mathfrak{B}}}\left( H_\infty(\mathcal{R}_{\bar{\mathfrak{B}}}, \mathcal{R}_{\widehat{\mathfrak{B}}_\tau}) > \epsilon \right) \leq \delta. \tag{82}$$

---

[23]All the other equations introduce negligible terms in the $\mathcal{O}(\cdot)$ notation.

[24]Notice that, in principle, $\pi_{\min, E_1} \neq \pi_{\min}$ that we defined for the sub-optimal experts.

[25]Notice, indeed, that we can apply Lemma E.9 as-is, but introducing the further constraints that $t$ should be sufficiently large so that Equation (80) holds.

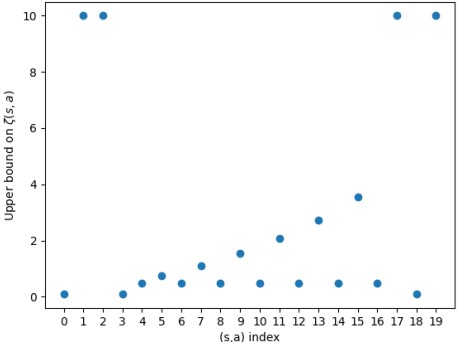 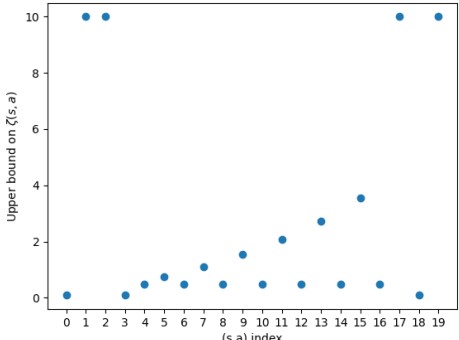

Figure 6: Experiment Results. Visualization of the maximum value that $\zeta$ can assume (using perfect knowledge of the underlying MDP) *(left)*. Visualization of the empirical maximum value that $\zeta$ can assume (the results report the mean over 20 runs) *(right)*.

We refer to these algorithms as $(\epsilon, \delta)$-correct identification strategies. For $(\epsilon, \delta)$-correct strategies, we define their sample complexity as the sum of the number of *unitary* samples gathered from the generative model. In other words, let us denote with $N_t(s, a)$ the number of samples gathered, at step $t$, from the environment, and let $N_t^{(i)}(s)$ denotes the number of samples gathered at step $t$ from the $i$-th expert at state $s$. Then, the sample complexity is given by:

$$\sum_{(s,a)} N_\tau(s, a) + \sum_i \sum_s N_\tau^{(i)}(s).$$

Given this learning formalism, it is straightforward to extend the results of Theorem 4.1 to this setting. More specifically, Theorem E.1 can be used to lower bound $\mathbb{E}\left[\sum_s N_\tau^{(i)}(s)\right]$ for each sub-optimal expert, thus showing a significant increase in the sample complexity (i.e., linear in the number of sub-optimal experts).

## H   Further details on US-IRL-SE

In this section, we provide further details on US-IRL-SE. More precisely, we specify the exact expression of the parameter $m$ that is used to provide $(\epsilon, \delta)$-correc

**Expression of $m$**   From the proof of Theorem 4.2, it is possible to derive an exact expression of $m$ that can be used in US-IRL-SE. More specifically, since $t = N_t(s, a)$,[26] it is sufficient to take $m$ as the minimum $t$ that satisfies Equations (75)-(78) and Equations (55)-(58).

**Computational Complexity of US-IRL-SE**   At each iteration, US-IRL-SE will query the generative model in all state-action pair $(s, a) \in \mathcal{S} \times \mathcal{A}$. Notice that the generative model samples (i) the environment (ii) all the expert's policy. Assuming a unitary cost for generating each sample, a single query to the generative model results in a computational complexity of $\mathcal{O}(n)$. Therefore, the total computational complexity of US-IRL-SE is given $\mathcal{O}(SAmn)$.

## I   Numerical Validation

We designed an experiment that aims at visualizing the reduction of the feasible reward set.

---

[26]This is a direct consequence of the uniform sampling strategy.

**Computing Resources**   The experiments have been run on a laptop with 8 Intel(R) Core(TM) i5-8250U CPU @ 1.60GHz and 8GB of RAM. Total time to gather the results is less then a few hours.

**Experiment Setup**   We considered as environment the forest management scenarios with 10 states and 2 actions that is available in the "pymdptoolbox" library.[27]   We considered a discount factor $\gamma = 0.9$. We have run policy iteration on this domain to recover a set of expert policies (i.e., the policy returned at each iteration by the policy iteration algorithm is a sub-optimal expert), and we have considered as $\xi$ the infinite norm between the value functions of the optimal policy and the sub-optimal ones computed using the true reward function.

**Results**   To appreciate the reduction in the feasible set that is introduced by the sub-optimal experts, we have plotted the maximum value that $\zeta(s, a)$ can achieve according to the theoretical bound of Eq. (7) (notice that this is way easier to visualize rather than an entire set of reward functions). For the sake of visualization, we have flattened the matrix that contained upper bounds on $\zeta$, and we plotted the result in Figure 6 (left). As one can notice, the presence of the sub-optimal experts can significantly limit the value of the advantage function in many state-action pairs (notice that the maximum value for $\zeta$ is given by $(1 - \gamma)^{-1} \approx 10$; this theoretical threshold represents the upper-bound on that was derived in [24] for the single-agent problem).

We have then run our algorithm for 20 times using $\epsilon = 0.1$ and $\delta = 0.1$, and computed the (empirical) theoretical upper bound on $\zeta$. The reviewer can find the empirical value of the upper bounds on in the Figure 6 (right). We have reported only the empirical mean, as the $95\%$ confidence intervals are in the order of 1e-5. As one can verify, the results are almost identical to the exact case.

---

[27]Code can be found at `https://github.com/sawcordwell/pymdptoolbox`. License is BSD 3-Clause "New" or "Revised" License. Master branch at commit 7c96789cc80e280437005c12065cf70266c11636 was used.

