# OpenReview forum: "Sub-optimal Experts mitigate Ambiguity in Inverse Reinforcement Learning"
_NeurIPS.cc/2024/Conference — NeurIPS 2024 poster_

### Official Review · Reviewer_KsdM · 2024-07-08

**Soundness:** 3
**Presentation:** 3
**Contribution:** 3
**Rating:** 6
**Confidence:** 3

**Summary:**

The authors introduce the IRL-SE problem, which which uses multiple experts, including sub-optimal ones, to enhance reward function learning. Given a bounded performance gap between experts, the authors provide a theoretical analysis on shrinking the feasible set of compatible reward functions. Using a PAC framework, the authors provide a lower bound on the sample complexity for estimating the feasible reward set

The authors propose a uniform sampling algorithm with theoretical guarantees, demonstrating its minimax optimality under certain conditions.

**Strengths:**

The authors provide interesting novel insights in a more general sitaution compared to previously cited work

The submission is technically sound and well supported by extremely detailed theoretical analysis. The work is complete and the authors evaluate their strenghts and provide future research avenues. The techincal notation is dense and can be difficult to follow, but the authors did well in writing precisely.

I think the overall proposal in this paper is quite applicable in many RL settings, so this is a strong submission with many potential use cases.

**Weaknesses:**

The submission could be written more clearly. Some results could be expanded instead of just staing the results. Particularly, the use of "it's easy to see" should be removed and expanded upon for ease to the reader. In some cases it may also aid readability to expand out expression instead of writing it in the most concise way possible.

The paper has a strong theoretical focus, which is strong and necessary. However, it would be powerful to see it actually employed in a case study to see how such a technique would be used. Even outside of empirical evidence, it would aid in readability of the paper to see how the notations used in the theoretical sections are actually used in practice. To that end, even a case study with a toy example would go a long way. The authors do a good job in providing technical examples through the paper, but these could be be futher enhanced by connecting them to a toy example of some kind. For a such a dense technical paper, it would make a world of difference in readability.

**Questions:**

Examples 3.1,3.2,3.3. Maybe it's "easy to show", but can you please show where these come from?

Suggestion: The word "Indeed" is used too many times. It is quite repetitive.

Suggestion: For line 109, can you explain why the infinite norm is limited to the particular range?
Operators: $\pi$ is overloaded to be both an operator and a policy

**Limitations:**

Future research directions are provided, but a dicussion on technical limitations would be also be good to have. No discussion on potential negative societal impact.

---

> ### Author Rebuttal · Authors · 2024-08-06
>
> > ### It would be powerful to see it actually employed in a case study to see how such a technique would be used. Even outside of empirical evidence, it would aid in readability of the paper to see how the notations used in the theoretical sections are actually used in practice. To that end, even a case study with a toy example would go a long way.
>
> We thank the Reviewer for raising this point. We have added an experiment to show how the presence of sub-optimal experts limit the feasible reward set in a tabular domain. We refer the reviewer to our reply to Reviewer 6BJu.
>
> Finally, we agree with the Reviewer that an intuitive example can benefit the reader. Given the page limit, however, we preferred to give more space to an appropriate formalism to model the problem. We will make use of the additional page to insert an intuitive example on a 2D goal-based grid-world that shows how the presence of the sub-optimal experts shrinks the feasible reward set.
>
> > ### Examples 3.1,3.2,3.3. Maybe it's "easy to show", but can you please show where these come from?
>
> We thank the reviewer for pointing this out, and we are sorry about this. We agree that the presentation would benefit from formal justifications. Below, the Reviewer can find proofs about these examples. We integrated them in the appendix in a revised version of the paper.
>
> **Proof of Example 3.1**
> Consider two MDPs \ R $\mathcal{M}\_1$ and $\mathcal{M}\_2$ that differs only in the transition function, namely $p_1$ and $p_2$. Suppose that $\pi\_{E\_1}$ and $\pi\_{E\_2}$ are optimal for $\mathcal{M}\_1$ and $\mathcal{M}\_2$ respectively. We are interested in upper-bounding $V\_{\mathcal{M}\_1 \cup r}^{\pi\_{E\_1}}(s) - V\_{\mathcal{M}\_1 \cup r}^{\pi\_{E\_2}}(s)$. Then, for any state $s$, we have that:
>
> $$
> V\_{\mathcal{M}\_1 \cup r}^{\pi\_{E\_1}}(s) - V\_{\mathcal{M}\_1 \cup r}^{\pi\_{E\_2}}(s) \le V\_{\mathcal{M}\_1 \cup r}^{\pi\_{E\_1}}(s) - V\_{\mathcal{M}\_2 \cup r}^{\pi\_{E\_1}}(s) + V\_{\mathcal{M}\_2 \cup r}^{\pi\_{E\_2}}(s) - V\_{\mathcal{M}\_1 \cup r}^{\pi\_{E\_2}}(s)
> $$
>
> (we have added and subtracted $V^{\pi\_{E\_1}}\_{\mathcal{M}\_2 \cup r}$, and then we have used $V^{\pi\_{E\_1}}\_{\mathcal{M}\_1 \cup r} \le V^{\pi\_{E\_2}}\_{\mathcal{M}\_2 \cup r}$ due to the optimality of $\pi\_{E\_2}$ in $\mathcal{M}\_2$). Then, focus on $V\_{\mathcal{M}\_1 \cup r}^{\pi\_{E\_1}}(s) - V\_{\mathcal{M}\_2 \cup r}^{\pi\_{E\_1}}(s)$ (an identical reasoning can be applied for the second difference). This can we written as $$ \gamma \sum\_a \pi\_{E\_1}(a|s) \sum\_{s'} p\_1(s'|s,a) (V^{\pi\_{E\_1}}_{\mathcal{M}\_1}(s') - V^{\pi\_{E\_1}}\_{\mathcal{M}\_2}(s')) + (p\_1(s'|s,a) - p\_2(s'|s,a))V^{\pi\_{E\_1}}\_{\mathcal{M}_2}(s')$$
>
> which, in turn, can be further bounded by:
>
> $$\frac{\gamma}{1-\gamma} ||p\_1 - p\_2 ||\_1 + \gamma \sum\_a \pi\_{E\_1}(a|s) \sum\_{s'} p\_1(s'|s,a) (V^{\pi\_{E\_1}}\_{\mathcal{M}\_1}(s') - V^{\pi\_{E\_1}}\_{\mathcal{M}\_2}(s'))$$
>
> Unrolling the summation to iterate the aforementioned argument, and using the fact that $\sum\_{t=0}^{+\infty} \gamma^t = \frac{1}{1-\gamma}$ concludes the proof.
>
> **Proof of Example 3.2**
> In this proof, we explicit the relationship of the value function $V$ with the discount factor $\gamma$ by writing $V^{\pi, \gamma}\_{\mathcal{M} \cup r}$. Then, we have that
>
> $$V\_{\mathcal{M} \cup r}^{\pi\_{E\_1}, \gamma}(s) - V\_{\mathcal{M} \cup r}^{\pi\_{E\_2}, \gamma}(s) \le V\_{\mathcal{M} \cup r}^{\pi\_{E\_1}, \gamma}(s) - V\_{\mathcal{M} \cup r}^{\pi\_{E\_1}, \gamma'}(s) + V\_{\mathcal{M} \cup r}^{\pi\_{E\_2}, \gamma'}(s) - V\_{\mathcal{M} \cup r}^{\pi\_{E\_2}, \gamma}(s)$$
>
> (we have added and subtracted $V\_{\mathcal{M} \cup r}^{\pi\_{E\_1}, \gamma'}(s)$, and then we have used $V\_{\mathcal{M} \cup r}^{\pi\_{E\_1}, \gamma'}(s) \le V\_{\mathcal{M} \cup r}^{\pi\_{E\_2}, \gamma'}(s)$ due to the optimality of $\pi\_{E\_2}$ for the discount factor $\gamma'$).
>
> At this point, focus on $V\_{\mathcal{M} \cup r}^{\pi\_{E\_1}, \gamma}(s) - V\_{\mathcal{M} \cup r}^{\pi\_{E\_1}, \gamma'}(s)$. Using the definition of the value function, together with the fact that rewards are bounded in $[0,1]$, we can rewrite this difference as $\mathbb{E}[\sum\_{t=0}^{+\infty} (\gamma^t - {\gamma'}^{t}) r(s\_t, a\_t)] \le \sum\_{t=0}^{+\infty} \gamma^t - {\gamma'}^{t} = \frac{\gamma - \gamma'}{(1-\gamma)(1-\gamma')}$. An identical argument holds for the second difference, thus concluding the proof.
>
> **Proof of Example 3.3**
>
> By using the definition of value function, we can rewrite $V\_{\mathcal{M} \cup r}^{\pi\_{E\_1}}(s) - V\_{\mathcal{M} \cup r}^{\pi\_{E\_2}}(s)$ as
> $\sum\_{a} (\pi\_{E\_1}(a|s) - \pi\_{E\_2}(a|s) )r(s,a) + \gamma \sum\_{s'} p(s'|s,a) (V\_{\mathcal{M} \cup r}^{\pi\_{E\_1}}(s') - V\_{\mathcal{M} \cup r}^{\pi\_{E\_2}}(s')))$.
> At this point, since rewards are bounded in $[0,1]$ and by the policy similarity assumption, we have that the difference in value functions can be upper bounded by $\epsilon + \gamma \sum\_{s'} p(s'|s,a) (V\_{\mathcal{M} \cup r}^{\pi\_{E\_1}}(s') - V\_{\mathcal{M} \cup r}^{\pi\_{E\_2}}(s'))$. Unrolling the summation to iterate the aforementioned argument, and using the fact that $\sum\_{t=0}^{+\infty} \gamma^t = \frac{1}{1-\gamma}$ concludes the proof.
>
>
> > ### Writing suggestions
>
> We thank the Reviewer for the suggestions. In a revised version, we limit the use of word "Indeed" and we removed the use of "it's easy to see" and we replaced with formal justifications (see point above).
> On the use of the symbol $\pi$, we replaced the operator $\pi$ with $\bar{\pi}$ to distinguish it from the policy symbol $\pi$.

---

> > ### Comment · Reviewer_KsdM · 2024-08-12
> >
> > Thanks for your response! I will keep my score as is

---

### Official Review · Reviewer_6BJu · 2024-07-08

**Soundness:** 3
**Presentation:** 4
**Contribution:** 3
**Rating:** 6
**Confidence:** 3

**Summary:**

This paper aims to mitigate the intrinsic reward ambiguity in IRL problems.  The authors propose that incorporating sub-optimal expert demonstrations can lead to a more accurate estimate of the feasible reward set. Initially, they formulate the IRL with Sub-Optimal Experts (IRL-SE) problem and explore the theoretical properties of the resulting feasible reward set, offering both implicit and explicit descriptions. Additionally, they establish the lower bound of the statistical complexity for IRL-SE problems within the Probably Approximately Correct (PAC) framework. Finally, the authors introduce a uniform sampling algorithm that achieves minimax optimality for IRL-SE.

**Strengths:**

- This paper is well-written and easy to follow. The examples provided in the paper effectively illustrate the role of sub-optimal experts.
- This paper provides a clear and comprehensive theoretical analysis of the IRL-SE problem. The approach of identifying, modeling, evaluating, and solving the problem is highly commendable and worth emulating. First, the authors formalize the problem and demonstrate the theoretical feasibility of resolving reward ambiguity. Then, they determine the difficulty of the problem by analyzing the lower bound of statistical complexity within the PAC framework. Finally, the authors propose an algorithm that achieves minimax optimality to solve the IRL-SE problem.

**Weaknesses:**

- The paper does not provide a method for determining a suitable performance gap $\zeta$ for sub-optimal experts. Additionally, in examples such as 3.1 and 3.3, when $\gamma=0.99$ (a commonly used value), the estimated performance gap $\zeta$ tends to be quite large. This may limit the effectiveness of sub-optimal experts in resolving ambiguity in reward sets, which could be a significant obstacle for algorithm deployment.
- The paper lacks experimental results to validate the role of sub-optimal experts in resolving reward ambiguity.

**Questions:**

Based on the weaknesses, my questions are as follows.

- How can we utilize the feasible reward set in a real-world scenario? I understand that IRL enables learning of reward models from expert demonstrations, which can then optimize policies through RL. However, if we obtain a reward set instead, how should we make use of it?
- Can you provide more insights into determining the performance gap of sub-optimal policies?
- Can you experimentally verify if sub-optimal experts can mitigate reward ambiguity? For example, in certain MDPs, as more sub-optimal experts are provided, the cardinality of the feasible reward set tends to decrease.

**Limitations:**

This paper does not discuss the limitations of the work.

---

> ### Author Rebuttal · Authors · 2024-08-06
>
> > ### How can we utilize the feasible reward set in a real-world scenario?
>
> We thank the Reviewer for raising this point. How to make the best use of the feasible reward set to learn a policy in RL is currently an open problem, even for the single-agent IRL formulation of the feasible reward set (see Section 8 in Metelli et al., ICML 2023).
>
> Nevertheless, suppose that the algorithm has terminated, and, consequently, the estimated policies $\\{ \hat{\pi}\_{E\_i} \\}\_{i=1}^{n+1}$ and the empirical model $\hat{P}$ are available to the agent.  At this point, given any $V$ and $\zeta$ that satisfies Eq. (4) and (5) for $\\{ \hat{\pi}\_{E\_i} \\}\_{i=1}^{n+1}$  and $\hat{P}$, let us write $\hat{r}\_{v,\zeta}$ for its corresponding reward function.
>
> Now, suppose we are given a function $f: \mathcal{S} \times \mathcal{A} \rightarrow \mathbb{R}$ that evaluates the quality of a compatible reward function $\hat{r}\_{V, \zeta}$, i.e., $f(\hat{r}\_{V, \zeta})$, so that we are interested in learning a policy for the reward function that maximizes $f$. Then, we can search over the feasible reward set for the function that maximizes $f$. This problem is simply a constrained optimization problem, where the constraints are the ones on $V$ and $\zeta$ specified by the empirical versions Eq. (5) and (6) (i.e., $P$ is replaced with $\hat{P}$ and $\pi_{E_i}$ is replaced with $\hat{\pi}_{E_i}$). Since all these constraints are linear in $V$ and $\zeta$, the complexity of the optimization is entirely dominated by the complexity of evaluating the objective function $f$. If evaluating $f$ is computationally efficient, then, searching for a reward function that maximizes $f$ is computationally efficient.
>
> Once the reward function that maximizes $f$ (or an approximation) is available, we can use it to train the RL agent. Notice that, in this sense, we can appreciate that this approach does not force us to select a criterion $f$ beforehand, and therefore, we might adopt the aforementioned strategy for several criteria (e.g., the max-margin approach of "Algorithms for inverse reinforcement learning", Ng and Russel, ICML 2000) and train several RL agents.
>
> Finally, it has to be remarked that learning a policy in RL is not the only purpose of IRL. Another typical application is the interpretability of the behavior of the expert. In this sense, the feasible reward set approach offers broader perspectives w.r.t. standard IRL methods that adopt specific criteria to select a single reward function.
>
> > ### Can you provide more insights into determining the performance gap of sub-optimal policies?
>
> Determining the performance gap $\xi_i$ for sub-optimal experts is a relevant issue. First of all, in the paper we have provided three scenarios (Examples 3.1-3.3) in which the values of $\xi_i$ can be computed, with no assumption of no knowledge of the (possible) reward function optimized by the expert policy. We remark that $\xi_i$ is to be intended as an upper bound of the degree of sub-optimality of expert $i$ and, for this reason, even a rought estimate (provided that it is an overestimate) is acceptable. Furthermore, the presence of a human domain expert is able to evaluate the sub-optimality of several agents, providing estimates of $\xi_i$, is a more realistic scenario than requesting the human expert to explicitly design a reward function.
>
> >  ### Can you experimentally verify if sub-optimal experts can mitigate reward ambiguity?
>
> We thank the Reviewer for raising this point. We have designed an experiment that aims at visualizing the reduction of the feasible reward set. We have considered as environment the forest management scenarios with $10$ states and $2$ actions that is available in the "pymdptoolbox" library. We considered a discount factor $\gamma = 0.9$. We have run policy iteration on this domain to recover a set of expert policies, and we have considered as $\xi_i$ the infinite norm between the value functions of the optimal policy and the sub-optimal ones computed using the true reward function.
>
> To appreciate the reduction in the feasible set that is introduced by the sub-optimal experts, we have plotted the maximum value that $\zeta(s,a)$ can achieve according to the theoretical bound of Eq. (7) (notice, indeed, that this is way easier to visualize rather than an entire set of reward functions). For the sake of visualization, we have flattened the matrix that contained upper bounds on $\zeta$, and the reviewer can find the result in the right figure in the pdf that we attached to the general answer. As the Reviewer can notice, the presence of the sub-optimal experts can significantly limit the value of the advantage function in many state-action pairs (notice that the maximum value for $\zeta$ is given by $1 / (1-\gamma) \approx 10$; this theoretical threshold represents the upper-bound on $\zeta$ that was derived in Metelli et al., 2021 for the single-agent problem).
>
> We have then run our algorithm for $20$ times using $\epsilon = 0.1$ and $\delta = 0.1$, and computed the again the theoretical upper bound on $\zeta(s,a)$. The reviewer can find the empirical value of the upper bounds on $\zeta$ in the left figure. We have reported only the empirical mean, as the $95\%$ confidence intervals are in the order of $1^{-5}$. As one can verify, the results are almost identical to the exact case.
>
>
> We integrated this result in the appendix in a revised version of our manuscript.

---

> > ### Comment · Reviewer_6BJu · 2024-08-09
> >
> > Thank you very much for your detailed response. I appreciate the contributions of this work and will remain my positive score.

---

### Official Review · Reviewer_zNGu · 2024-07-09

**Soundness:** 4
**Presentation:** 3
**Contribution:** 3
**Rating:** 7
**Confidence:** 4

**Summary:**

This paper studies unregularized IRL involving one optimal expert and $n$ suboptimal experts. The authors show that the additional suboptimal experts can help mitigate the reward ambiguity in IRL that arises due to the unknown suboptimality of unvisited state-action pairs. They present an analytical expression for the feasible reward set in this setting and demonstrate that the reward ambiguity can be reduced when learning from not-too-suboptimal experts yet playing differently from the optimal expert. Furthermore, assuming a generative model and sample-based access to experts' policies, the authors provide a PAC lower bound for the sample complexity of IRL and propose an algorithm that is minimax optimal for small suboptimalities of the experts.

**Strengths:**

1. The paper makes a notable contribution to IRL by showing that leveraging suboptimal experts can effectively reduce reward ambiguity.
2. Theorems 4.1 and 4.2, presenting lower and upper bounds, are particularly appreciated.
3. The mathematical notation is clearly introduced, and full proofs are available in the appendix.

**Weaknesses:**

1. The paper assumes access to a generative model of the experts' policies. However, in practical IRL a finite data set of precollected expert demonstrations is the norm. This makes Algorithm 1 difficult to implement in practice.
2. Minor points: a) In my opinion the claims made in line 109 and in Examples 3.1-3.3 need a reference or a brief proof. b) I believe there is a typo in Eq. (5): Since we are taking the expectation with respect to the state-action occupancy measure, we don't need the right-hand side should be a scalar, right? Moreover, in Eq. (10), I assume it should be $\min$ instead of $\max$.

**Questions:**

1. As pointed out, we would generally like the suboptimalities $\xi_i$ to be small and the $\pi_{\min}$ to be large. However, these may be conflicting goals, given that we also want the suboptimal experts to deviate from $\pi_{E_1}$. Could this problem be mitigated by having experts that are more suboptimal (e.g. a completely uniform expert) but with lower and upper bounds on their suboptimality?
2. What if we were to drop the optimal expert $\pi_{E_1}$? Can we say anything about the feasible reward set in that setting?

**Limitations:**

Some limitations are discussed in the conclusion section. However, it would be good to be more clear about the practical limitations of the provided results.

---

> ### Author Rebuttal · Authors · 2024-08-06
>
> > ### The paper assumes access to a generative model of the experts' policies. However, in practical IRL a finite data set of precollected expert demonstrations is the norm
>
> We thank the reviewer for raising this point. We agree with the reviewer that the generative model represents a limitation. Extending this work to remove the assumption on the generative model and working directly on a dataset of demonstrations is an intriguing avenue for future research. In this sense, one could draw inspiration from the recent paper "Offline Inverse RL: New Solution Concepts and Provably Efficient Algorithms", Lazzati et al., 2024, where the authors analyze solution concepts and algorithms for recovering the feasible reward set in the single-agent IRL setting.
>
> > ### Minor points
>
> We thank the Reviewer for these notes.
>
> We have inserted proofs for Example 3.1-3.3 in a revised version of the manuscript and also in this Rebuttal (see reply to Reviewer KsdM).
>
> On Equation (5). Having fixed a sub-optimal expert $i$, Equation (5) represents a set of $S$ inequalities. This is visible in Equation (6) that makes explicit *one* of such inequalities having fixed $s'$ (the initial state of the occupancy measure). By varying $s'\in \mathcal{S}$, we obstain $S$ inequalities, and for this reason, the right hand side of Equation (5) is an $S$-dimensional vector.
>
> On Equation (10). Yes, thanks for the typo. Equation (10) should be changed to a $\min_{i \in \{2, \dots n \}} \min_{(s,a): \pi_{E_i} > 0} \pi_{E_i}(s,a)$.
>
> > ### As pointed out, we would generally like the suboptimalities to be small and the $\xi$ to be large. However, these may be conflicting goals, given that we also want the suboptimal experts to deviate from $\pi_{E_1}$. Could this problem be mitigated by having experts that are more suboptimal (e.g. a completely uniform expert) but with lower and upper bounds on their suboptimality?
>
> We thank the Reviewer for the interesting question. We agree that these may be conflicting goals, at least when no additional structure is enforced (consider, for example, a kernelized and continuous state-action space which has been discretized; here, taking similar but distinct actions leads to similar effects on the underlying MDP; in this case, the goals are not necessarly conflicting).
>
> Nevertheless, as the Reviewer suggests, we agree that one way to mitigate this problem would be having experts with lower and upper bounds on their sub-optimality. In this case, the lower and upper bounds will delimit the set of reward functions with potentially more information, thus avoiding the "need" for "precise" experts with small sub-optimality gaps.
>
> >  ### What if we were to drop the optimal expert ? Can we say anything about the feasible reward set in that setting?
>
> We thank the Reviewer for raising the interesting question. If we drop the optimal expert, we are still be able to make use of this knowledge to reduce the feasible reward set. Consider a set of experts for which we know that $0 \le V^{E_1} - V^{E_i} \le \xi_i$ holds. Then, for each pair $(i,j) \in \{2, \dots, n+1 \}$, simple algebraic manipulations leads to the following inequalities: $V^{E_j} - V^{E_i} \le \xi_i$, which does not depend on the optimal policy $\pi_{E_1}$. As a remark, we note that, by considering only these inequalities (i.e., by neglecting the presence of $\pi_{E_1}$), one can still obtain sufficient conditions for describing the feasible reward set.
>
> As an illustrative example, consider a simple bandit with two sub-optimal experts $i$ and $j$. Then, suppose that $\pi_{E_i}(a_i) = 1$ and $\pi_{E_j}(a_j) = 1$ for some actions $a_i, a_j$ such that $a_i \ne a_j$. Then, the previous set of equations leads to $r(a_j) - r(a_i) \le \xi_i$ and $r(a_i) - r(a_j) \le \xi_j$, thus introducing constraints on the values that the reward functions can assume in these actions.

---

> > ### Comment · Reviewer_zNGu · 2024-08-12
> >
> > Thank you for the thorough response to my concerns and questions. I appreciate the paper's contributions and will keep my positive score.

---

### Official Review · Reviewer_Euvz · 2024-07-30

**Soundness:** 3
**Presentation:** 3
**Contribution:** 3
**Rating:** 7
**Confidence:** 2

**Summary:**

The paper develops a theory to address inverse reinforcement learning (IRL) from sub-optimal expert demonstrations. The authors assume a set of experts with known degrees of sub-optimality. Rather than learning a single reward model, as is often done in standard IRL, they provide an explicit characterization of all plausible reward models compatible with the experts. Moreover, they give lower bounds on the number of generated samples from the experts for a PAC guarantee on the set of reward models. Finally, they propose a uniform sampling procedure that can result in near-optimal estimation of the reward models in terms of sample complexity.

**Strengths:**

1. Although this is not the first work for IRL from sub-optimal experts, I haven't seen characterizing all plausible reward models for such a setting, so I would call the work original in this sense.
2. The authors provide good intuition on why having more sub-optimal experts can shrink the set of feasible reward models in Figures 1 and 2.
3. I haven't fully validated the proofs for the theory. But it seems both Theorem 4.1 (the lower bound on the required sample size for the $(\epsilon,\delta)$-correct identification of the reward set) and Theorem 4.2 (the upper bound for the uniform sampling algorithm) are mainly based on the theoretical results in Metelli et al., 2023. Moreover, the characterization of plausible reward models in Theorem 3.3 is mainly developed by Metelli et al., 2021 (especially the eq. (5), which characterizes the feasible rewards for the case of optimal experts). In that sense, the novelty of this work is introducing more constraints based on the sub-optimality index of additional demonstrators, which is characterized in eq. (3) and point (iii) in line 162 of page 3. Having said that, those extensions based on the extra constraints seem non-trivial. So, I'd call the theoretical contribution novel.

**Weaknesses:**

The main weakness of the setting is that it assumes having access to a generative model for the optimal and sub-optimal experts during sampling and that there is a performance gap between the sub-optimal and optimal experts. However, given that the main contribution of the paper is theoretical, I still vote for acceptance.

**Questions:**

N/A

---

> ### Author Rebuttal · Authors · 2024-08-06
>
> > ### The main weakness of the setting is that it assumes having access to a generative model for the optimal and sub-optimal experts during sampling and that there is a performance gap between the sub-optimal and optimal experts
>
> We thank the Reviewer for raising this point. Extending this work to remove the assumption on the generative model and working directly on a dataset of demonstrations is an intriguing avenue for future research. In this sense, one could draw inspiration from the recent paper "Offline Inverse RL: New Solution Concepts and Provably Efficient Algorithms", Lazzati et al., 2024, where the authors analyze solution concepts and algorithms for recovering the feasible reward set in the single-agent IRL setting. Concerning the assumption of the performance gap, we take the chance to note that our formulation still generalizes existing theoretical works on IRL, which usually assume to have access to possibly multiple **optimal** experts (see, e.g., "Identifiability and generalizability from multiple experts in inverse reinforcement learning.", Rolland et al., NeurIPS 2022, or additional works that we discuss in Section 5).

---

> > ### Comment · Reviewer_Euvz · 2024-08-09
> >
> > Thank you for your response! I'll keep my score as it is.

---

### Author Rebuttal · Authors · 2024-08-06

We thank the reviewers for the time they spent reviewing our paper. Specifically, we are happy that the reviewers considered our work "original" and with "novel technical contribution" (Reviewer Euvz), with "notable contribution to IRL" (Reviewer zNGu), and a "clear and comprehensive theoretical analysis of the IRL-SE problem" (Reviewer 6BJu), and that it models "more general situation compared to previously cited work" (Reviewer KsdM).

In the following, we answer the remaining questions. We are also attaching a PDF that contains figures on an experiment that was asked by reviewers 6BJu and KsdM.

---

### Decision · Program_Chairs · 2024-09-25

**Decision:**

Accept (poster)

**Comment:**

The paper examines an instance of the inverse RL problem, where the goal is to reconstruct the feasible reward set from demonstrations provided by both an optimal expert and multiple sub-optimal experts. All reviewers were positive about this work, appreciating its theoretical contributions to the literature on IRL and the soundness of its main technical claims. Although the paper lacks experimental results to validate its main findings, its theoretical contributions appear strong enough on their own. Therefore, I recommend acceptance.